# Macroscopic Length Correlations in Non-Equilibrium Systems and Their Possible Realizations

Zohar Nussinov[1, 2, ∗]

[1]*Department of Physics, Washington University, St. Louis, MO 63160, USA*
[2]*Kavli Institute for Theoretical Physics, University of California, Santa Barbara, CA 93106, USA*
(Dated: August 29, 2019)

We consider general systems that start from and/or end in thermodynamic equilibrium while experiencing a finite rate of change of their energy density or of other intensive quantities $q$ at intermediate times. We demonstrate that at these times, during which the global intensive quantities $q$ vary at a finite rate, the associated covariance, the connected pair correlator $G_{ij} = \langle q_i q_j \rangle - \langle q_i \rangle \langle q_j \rangle$, between any two (far separated) sites $i$ and $j$ in a macroscopic system may, on average, become finite. Such non-vanishing connected correlations between distant sites are general and may also appear in quantum and classical theories that only have local interactions. If in an initial equilibrium state, the intensive quantities $q$ are static then a minimal time scale $t_{\min}$ may need to be exceeded in order for an external drive to create a finite expectation value of $dq/dt$; concomitantly, in such driven systems, a finite average of $G_{ij}$ over all site pairs can appear at times $t > t_{\min}$. For systems of linear scale $L$ with a maximal (Lieb-Robinson or other) speed $v$, this minimal time $t_{\min} = \mathcal{O}(L/v)$. Once the global mean $q$ no longer changes, the average of $G_{ij}$ over all site pairs $i$ and $j$ may tend to zero. However, when the equilibration times are significant (e.g., as in a glass that is not in true thermodynamic equilibrium yet in which the energy density (or temperature) reaches a final steady state value), these long range correlations may persist also long after $q$ ceases to change. We explore viable experimental implications of our findings and speculate on their potential realization in glasses (where a prediction of a theory based on the effect that we describe here suggests a universal collapse of the viscosity that agrees with all published viscosity measurements over sixteen decades) and non-Fermi liquids. We derive uncertainty relation based inequalities that connect the heat capacity to the dynamics in general open thermal systems. These rigorous inequalities suggest the shortest possible fluctuation times scales in open equilibrated systems at a temperature $T$ are typically "Planckian" (i.e., $\mathcal{O}(\hbar/(k_B T))$). We briefly comment on parallels between quantum measurements, unitary quantum evolution, and thermalization and on how Gaussian distributions of intensive quantities may generically emerge at long times after the system is no longer driven.

PACS numbers: 05.50.+q, 64.60.De, 75.10.Hk

## I. INTRODUCTION

In theories with local interactions, the connected correlations between two different sites $i$ and $j$ often decay with their spatial separation $|i - j|$. Indeed, connected correlations decay exponentially with distance in systems with finite correlation lengths. In massless (or critical) theories, this exponential decay is typically replaced by an algebraic drop. The detailed understanding of these decays was achieved via numerous investigations that primarily focused on venerable equilibrium and other systems with fixed control parameters, e.g., [1–12]. Further pioneering studies examined work-free energy relations in irreversible systems [13–16]. We wish to build on these notions and ask what occurs in a general (quantum or classical) non-relativistic system, when an intensive parameter such as the average energy density (set, in all but the phase coexistence region where latent heat appears, by the temperature) or external field is varied so that, during transient times, the system is forcefully kept *out of thermal equilibrium*. We will illustrate that, un-

der these circumstances, extensive fluctuations will generally appear. These large fluctuations will imply the existence of connected two point correlation functions that will, on average, *remain finite for all spatial separations*. If the system returns to equilibrium, these long range correlations may be lost. In focusing on driven non-equilibrium systems, the quantum facets of our work complement investigations on nontrivial aspects of the interplay between entanglement and thermalization that have witnessed a flurry of activity in recent years in, e.g., studies of operator scrambling [17, 18] and entanglement growth [19]. Earlier celebrated analysis also suggested fundamental quantum mechanical bounds on the rate on which general thermal systems may become chaotic [20]. In the current work, we will largely focus on the more precise quantum descriptions. Nonetheless, much of our analysis can be replicated for the classical limit of these systems.

Although our considerations are general, we will largely couch these for theories residing on $d-$dimensional hypercubic lattices of $N = L^d$ sites; the average energy density $\epsilon \equiv E/N$ with $E$ the total energy. In theories with local interactions, we may express (in a variety of ways) the Hamiltonian $H$ as a sum of $N' = \mathcal{O}(N)$ terms ($\{\mathcal{H}_i\}_{i=1}^{N'}$) that are each of

∗ zohar@wustl.edu

finite range and bounded operator norm,

$$H = \sum_{i=1}^{N'} \mathcal{H}_i. \qquad (1)$$

Our principal interest lies in the thermodynamic ($N \gg 1$) limit. Since our focus is on general non-equilibrium systems, the (general time dependent) Schrodinger picture probability density matrix $\rho(t)$ need not be equal to the any of the standard density matrices describing equilibrium systems.

## II. SKETCH OF MAIN RESULT

In a nutshell, in order to establish the existence of long range correlations we will show the following:

> • If the expectation value of the Hamiltonian $H$ of the original (undriven) system varies in the time evolved (driven) state such that $\frac{d\epsilon}{dt} \equiv \frac{d}{dt} Tr\left(\rho(t) \frac{H}{N}\right) \neq 0$ then the energy density fluctuations $\sigma_\epsilon \equiv \sigma_{\frac{H}{N}}$ as computed with $\rho(t)$ will, generally, also be finite. Similar results apply to all other intensive quantities.

While we will largely employ the more general quantum formalism, our central result holds for *both quantum and classical systems*. The central function that we will focus on to further quantify these fluctuations is the probability density of global energy density,

$$P(\epsilon') \equiv Tr\left[\rho(t)\ \delta(\epsilon' - \frac{H}{N})\right]. \qquad (2)$$

To avoid cumbersome notation, in Eq. (2) and what follows, the time dependence of $P(\epsilon')$ is not made explicit; the reader should bear in mind that, throughout the current work, $P(\epsilon')$ is time dependent. In equilibrium, the energy density (similar to all other intensive thermodynamic variables) is sharply defined; regardless of the specific equilibrium ensemble employed, the distribution of Eq. (2) is a Dirac delta-function, $P(\epsilon') = \delta(\epsilon' - \epsilon)$. This is schematically illustrated in the left and righthand sides of Figure 1. As we highlighted above, the chief goal of the current article is to demonstrate that when a system that was initially in equilibrium is driven at intermediate times (by, e.g., rapid cooling) such that its energy density varies at a finite rate as a function of time, the distribution $P(\epsilon')$ will need not remain a delta-function. A caricature of this feature is provided in the central panel of Figure 1 [21]. Because the final state displays a broad distribution of energy densities, our result implies that the "work" per site, in the context of its quantum mechanical definitions as energy differences between final and initial states [13–16, 22] is not necessarily sharp (even in the $N \to \infty$ limit). Since the variance of $P(\epsilon')$ is a sum of pair correlators $G_{ij} \equiv \langle \mathcal{H}_i \mathcal{H}_j \rangle - \langle \mathcal{H}_i \rangle \langle \mathcal{H}_j \rangle$, this latter finite width of $P(\epsilon')$ of the system when it is driven implies (as we will explain in depth) that the correlations

$G_{ij}$ extend over macroscopic length scales that are of the order of the system size. (Here, $\langle \cdot \rangle$ denotes the average as computed with $\rho(t)$.)

Whenever the formerly driven system re-equilibrates, $P(\epsilon')$ becomes a delta-function once again (right panel of Figure 1). We will investigate driving implemented by either one of two possibilities:

(1) Endowing the Hamiltonian with a non-adiabatic transient time dependence leading to a deviation from $H$ only during a short time interval during which the system is driven (Sections (V, VI,VII,VIII, and XI)). In this case, between an initial and a final time, the Schrodinger picture Hamiltonian differs from $H$, i.e., $H(t_i = 0 < t' < t_f) \neq H$.

(2) Including a coupling to an external bath yet allow for no explicit time dependence in the fundamental terms forming the Hamiltonian (this approach is invoked in Section IV (in particular, in its second half describing Eq. (4), Section IX), and Appendices B, C, and D)). By comparison to procedure (1) above, this approach is more faithful to the real physical system in which the form of all fundamental interactions is time independent.

In procedure (1), the density matrix of the system evolves unitarily $\rho \to \rho(t) = \mathcal{U}(t)\rho\mathcal{U}^\dagger(t)$. In the more realistic approach (2), the evolution of the density matrix of the system $\rho_S(t)$ (now a reduced density matrix after a trace over the environment is performed) is described by a general (non-unitary [23]) dynamic map $\rho_S(t) = \Phi_t(\rho_S(0))$; a cartoon is provided in Figure 2.

In procedure (2), we will examine the probability distribution $P(\epsilon')$ of Eq. (2) with the replacement of $\rho(t)$ by $\rho_S(t)$.

The divide between these unitary and non-unitary evolutions with and without an external environment is a feature that is not always of great pertinence; indeed though common physical systems are not truly closed they are described to an excellent approximation by the standard unitary evolution of the Schrodinger equation. Complementing the standard distinction between unitary and non-unitary evolutions, there is another issue that we will highlight in the current work. As we will elaborate in Appendix B, there are physical constraints on the possible transient time variations of the effective Hamiltonian (that are captured by analysis including the effect of the environment). Notably, in a theory with interactions that are of finite range and strength, due to causality, the allowed changes in the transient time Hamiltonian that captures the effects of the environment cannot be made to instantaneously vary over arbitrarily large distances. That is, the environment cannot couple (nor decouple) to a finite fraction of a macroscopic system instantaneously. Keeping in mind this constraint on the form of the possible variations of the effective Hamilto-

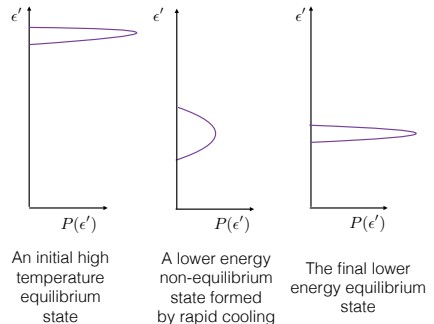

FIG. 1. A schematic of the probability distribution $P(\epsilon')$ of the energy density (Eq. (2)) for a rapid cooling process. Left: An initially equilibrated system at high temperatures where the energy density is sharply defined (in the thermodynamic limit, the distribution is a delta-function). Center: The system is rapidly cooled to a final state such that its energy density drops down at a finite rate as a function of time. During this cooling process, as it is being driven, $P(\epsilon')$ obtains a finite standard deviation (even for macroscopic systems). The demonstration of such a generic widening of the distribution is a principal objective of this paper. A finite standard deviation of $P(\epsilon')$ implies correlations that extend over length scales comparable to the system size. Right: After the cooling ceases, (if and) when the formerly driven system re-equilibrates, the distribution $P(\epsilon')$ becomes a delta-function once again (yet now at the lower temperature (smaller average energy density $\epsilon$) to which the system was cooled). Similar broadening may occur for general intensive quantities $q$.

nian of approach (1), we will often use these two descriptions interchangeably. Our inequalities will bound, from below, (a) the variance of the distribution $P(\epsilon')$ and (b) the magnitude of the pair correlator $G_{ij}$ for sites $i$ and $j$ that are separated by a distance that is of the order of the system size [24]. A similar broadening of the distribution $P(q')$ (and ensuing lower bounds on the associated pair correlators) may arise for general intensive quantities $q \equiv \langle Q \rangle / N$ (that include the energy density $\epsilon$ only as a special case).

## III. OUTLINE

A large fraction of the current work (Sections V - XI) establishing the central result of Section II and related effects will be somewhat mathematical in spirit. Towards the end of this paper (Sections XII,XIII, and Section XIV), touching on possible measurable quantities, our discussion will become more speculative.

We now briefly summarize the central contents of the various Sections. In Section IV, we explain why, in spite of its seemingly striking nature, our main finding of large variances (even in systems with local interactions) and the macroscopic range correlations that they imply is quite natural. By *macroscopic range*, we refer, in any macroscopic $N \gg 1$ site system, to correlations that span the entire system size. As we explain in Section IV (and in Appendices A, B, C, and D), in various physical settings, finite rates of change of the energy (and other) densities and concomitant long range correlations may appear only at sufficiently long time after coupling the system to an external drive. Next, in Section V, we discuss special situations in which our results do not hold—those of product states. This will prompt us to explore

systems that do not have a probability density that is of the simple local product form and to further discuss various aspects of entanglement. Notwithstanding their simplicity and appeal, product states do not generally describe systems above their ground state energy density. Similarly, the finite temperature probability densities of interacting classical systems do not have a product state form. In Section VI, we turn to more generic situations such as those appearing in rather natural dual models on lattices in an arbitrary number of spatial dimensions for which a class of finite energy density eigenstates can be exactly constructed. These theories principally include (1) general rotationally symmetric spin models (both quantum and classical) in an external magnetic field and (2) systems of itinerant hard-core bosons with attractive interactions. We investigate the effects of "cooling/heating" and "doping" protocols on these systems and illustrate that, regardless of the system size, after a finite amount of time, notable energy or carrier density fluctuations will appear. Armed with these proof of principle demonstrations of the energy density and number density fluctuations, we examine in Section VII the anatomy of a Dyson type expansion to see how generic these large fluctuations may be. Straightforward calculations illustrate that although there exist fine tuned situations in which the variance of intensive quantities such as the energy density remain zero (e.g., the product states of Section V) in rapidly driven systems, such circumstances are exceedingly rare. General non-adiabatic evolutions that change the expectation values of various intensive quantities may, concomitantly, lead to substantial standard deviations. In Section IX, we go one step further and establish that under a rather mild set of constraints, macroscopic range connected fluctuations are all but inevitable. (Yet another proof of these long range correla-

tions will be provided in Appendices C and D). In Section IX B, we derive bounds on the fastest fluctuation rates in open thermal system by linking a generalized variant of the quantum standard time-energy uncertainty relations to the heat capacity. Our new thermalization bounds suggest that, under typical circumstances, up to factors of order unity, the smallest fluctuation times for thermal systems cannot be shorter than "Planckian" times $\mathcal{O}(\hbar/(k_B T))$. We next illustrate (Section X) how general expectation values in these systems relate to equilibrium averages. Our effect has broad experimental implications: common systems undergoing heating/cooling and/or other evolutions of their intensive quantities may exhibit long range correlations. In Section XI, we demonstrate that the non-equilibrium system displays an effective equilibrium relative to a time evolved Hamiltonian. The remainder of the paper, largely focusing on candidate experimental and *in silico* realizations of our effect, is more speculative than the detailed exact solutions and derivations presented in its earlier Sections. In Sections XII and XIII, we turn to two prototypical systems and ask whether our findings may rationalize experimental (and numerical) results. In particular, in Section XII, we discuss glasses and show a universal collapse of the viscosity data that was inspired by considerations similar to those that we describe in the current work. In Section XIII, we ask whether the broadened distributions that we find may lead to "non-Fermi" liquid type behavior in various electronic systems. In Section XIV, we discuss adiabatic quantum processes and demonstrate how these may maintain thermal equilibration. We further speculate on possible offshoots of this result that suggest certain similarities between quantum measurements and thermalization. We conclude in Section XV with a synopsis of our results.

Various details (including an alternate proof of our central result, typical order of magnitude estimates, and further analysis) have been relegated to the appendices.

Appendix A provides simple estimates of the minimal time scale $t_{\min}$ that must be exceeded in order to establish finite rate of variation of the energy density (and concomitant long range correlations amongst the local contributions $\{\mathcal{H}_i\}$ to the Hamiltonian). In Appendix B, we prove that in typical non-relativistic systems with local interactions (where the Lieb-Robinson bounds apply), a finite rate of change of the energy density (and, similarly, that of other intensive quantities) is only possible at sufficiently long times $t > t_{\min}$.

As we briefly noted above, Appendices C and D will provide a complementary proof of our central result. In Appendix C, we demonstrate that a finite a rate of variation of the energy density implies long range connected correlations between the environment driving the system and the system itself. Appendix D then employs "classical" probability arguments to illustrate that the latter long range correlation between different sites in the system and its surrounding environment may lead to correlations between the sites in the system bulk even if these

sites are far separated.

In Appendix E, we show that using entangled states (similar to those analyzed in Section VI) reproduces the finite temperature correlators of an Ising chain. In Appendix F, we demonstrate that the entanglement entropy of symmetric entangled states is logarithmic in the system size; this latter calculation will further illustrate that the entangled spin states studied in Section VI display such macroscopic entanglement. These examples are meant to underscore that, even in closed systems, eigenstates of an energy density larger than that of the ground state can very naturally exhibit a macroscopic entanglement.

In Appendices G, H, and I, we discuss aspects related to the spin model example of Section VI (and, by extension, to some of the models dual to this spin model that are further studied in Section VI). Appendix G details what occurs when adding a general number of $S = 1/2$ spins. We connect the result in the limit of a large number of spins to the Gaussian distribution resulting from random walks (in the limit of large spins, the addition of spins naturally relates to the addition of classical vectors). Appendices H and I underscore the correlations in the initial state of this spin model system. Appendix H 1 explicitly introduces these correlations. In Appendix H 2, we explain why such correlations are inevitable in various cases. (The discussion in these appendices augment a more general result concerning correlations in the initial state of various driven systems that is described in the text following Eqs. (5, 6) regarding generally more complex correlations.) The central aim of Section VI was to provide the reader with a simple solvable spin model and its duals where a finite $\sigma_\epsilon$ and associated long range correlations between $\mathcal{H}_i$ appear hand in hand with a finite rate of change of the energy density. The exact solvability of the spin model of Section VI hints that the correlations that its initial simple correlations exhibit are not necessarily generic. In Appendix I, we outline a gedanken experiment in which the initial state of Section VI may be realized.

Appendix J provides intuitive arguments for the appearance of long time Gaussian distributions. Such long time Gaussian distributions were (a) invoked in our derivation of the 16 decade viscosity collapse of supercooled liquids and glasses and also appear (b) in standard textbook systems that have equilibrated at long times at general temperatures $T$ where (with $C_v$ denoting the heat capacity at constant volume), the width of the Gaussian distribution is given by $\sigma_\epsilon = \sqrt{k_B T^2 C_v}/N \sim \mathcal{O}(N^{-1/2})$.

Lastly, in Appendix K, we explain that, generally, the entanglement entropy may be higher than of the states studied in Appendix F.

## IV. INTUITIVE ARGUMENTS

To make our more abstract discussions clear, we first try to motivate why our central claim might not, at all,

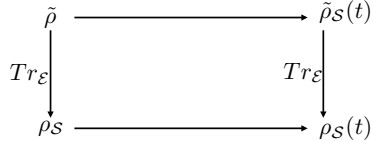

FIG. 2. The evolution of the density matrix $\tilde{\rho} \to \tilde{\rho}(t) = \tilde{\mathcal{U}}\tilde{\rho}\tilde{\mathcal{U}}$ describing the system $\mathcal{S}$ and its environment $\mathcal{E}$ (when, together, they form a larger closed hybrid system $\mathcal{I} = \mathcal{S} \cup \mathcal{E}$) is unitary. After tracing over the environment (Eq. (4)), the resultant dynamical mapping $\rho_{\mathcal{S}}(t) = \Phi_t(\rho\mathcal{S}(0))$ describing the reduced density matrix of the system alone is, generally, not a unitary transformation. In many situations of physical interest, the environment may, however, still be effectively captured by a modification of a system Hamiltonian. As we will explain in Appendix B, causality constrains this effective system Hamiltonian. In systems with local interactions, the rate of energy density of the system cannot be made to change instantaneously from zero to a finite value. A minimal time (linear in the system size) must elapse before the environment (and effective interactions borne by the presence of the environment) can couple to a finite fraction of the system.

be surprising and expand on the basic premise outlined in Section II. Consider a system that is, initially, in thermodynamic equilibrium with a sharp energy density $\epsilon$. For an initial closed equilibrium system (described by the microcanonical ensemble), the standard deviation of $\epsilon$ scales as $1/N$ while in open systems connected to a heat bath, the standard deviation of $\epsilon$ is $\mathcal{O}(1/\sqrt{N})$. In either of these two cases, the standard deviation of $\epsilon$ vanishes in the thermodynamic limit (similar results apply to any intensive thermodynamic variable), see, e.g., the right-hand panel of Figure 1. Now imagine cooling the system. As the system is cooled, its energy density $\epsilon$ drops. Various arguments hint that as $\epsilon$ drifts (or is "translated") downwards in value, its associated standard deviation also increases (see the central panel of Figure 1). This is analogous to the increase in width of an initially localized "wave packet" with a non-trivial evolution (with the energy density itself playing the role of the packet location). This argument applies to *both quantum and classical systems* (with the classical probability distribution obeying a Liouville or Fokker-Planck type equations instead of the von Neumann equation obeyed by the quantum probability density matrices). Thus, on a rudimentary level, it might be hardly surprising that the energy density obtains a finite standard deviation when it continuously varies in time. A finite standard deviation of the energy density implies long range correlations of the local energy terms. This is so since the variance of the energy density

$$0 < \sigma_\epsilon^2 = \frac{1}{N^2}\sum_{i,j}(\langle\mathcal{H}_i\mathcal{H}_j\rangle - \langle\mathcal{H}_i\rangle\langle\mathcal{H}_j\rangle)$$

$$= \frac{1}{N^2}\sum_{i,j}G_{ij} \equiv \overline{G}. \tag{3}$$

Thus, if $\sigma_\epsilon$ is finite then the average $\overline{G}$ of $G_{ij}$ over all separations $|i - j|$ will be non-vanishing. More broadly, similar considerations apply to intensive quantities of the form $q = \frac{1}{N}\sum_i q_i$ that must have a sharp value in thermodynamic equilibrium. Thus, generally, if $q$ broadens as some parameters are varied, there must be finite connected correlations $(\langle q_i q_j\rangle - \langle q_i\rangle\langle q_j\rangle)$ even when $|i - j'|$

is the order of the linear dimension of a macroscopic system. Identical conclusions to the ones presented above may be drawn for systems that end in thermodynamic equilibrium (instead of starting from equilibrium) while experiencing a finite rate of change of their energy density at earlier times at which Eq. (3) will hold. This effect may appear for quantum as well as classical systems. Generally, there are "classical" and "quantum" contributions [25] to the variance $\sigma_\epsilon^2$.

Empirically, in cases of experimental relevance, as in, e.g., cooling or heating a material, if the rate of change of its temperature (or energy density) is finite then Eq. (3) will hold. Although heat (and other) currents associated with various intensive quantities $q$ traverse material surfaces, experimentally, even for thermodynamically large systems, the rate of change of energy density $\epsilon$, and other intensive quantities $q$ can be readily made finite, i.e., $dq/dt = \mathcal{O}(1)$. This common experimentally relevant situation of finite heat or other rate of change $dq/dt$ in macroscopic finite size ($N \gg 1$) samples is the focus of our attention (see Appendix A). We nonetheless remark that if the energy density (or other intensive parameter) exchange rate are dominated by contributions in Eq. (3) with $i$ and $j$ close to the surface then $dq/dt = \mathcal{O}(1/L)$ and the average connected correlator associated with $q$ for arbitrarily far separated sites $i$ and $j$ will be bounded by $\overline{G_q} \geq \mathcal{O}(N^{-2/d})$ [26]. As we will emphasize in Section IX, in order to achieve a finite rate of change of any intensive quantity (including that of the energy density $d\epsilon/dt$ (or, equivalently, of the measured temperature $dT/dt$)), the coupling (and correlations) between the system and its surroundings must be extensive and involve minimal time scales (see Appendices A, B, and C). In reality, due to the surface flow of the heat current from the surrounding environment to the system during periods of heating or cooling, the local energy density in the system is generally spatially non–uniform and may depend on the distance to the surrounding external bath from which heat flows to the system.

The physical origin of the long range correlations of Eq. (3) in general systems (either quantum or classical) is rather transparent and is symbolically depicted

in Figure 3. As noted above, in order to achieve a finite rate of cooling/heating in a system with bounded interaction strengths, a finite fraction of the fields/sites in the system must couple to the surrounding heat bath (see also Appendix C for a simple brief demonstration of macroscopic length correlations between the surrounding environment and the system bulk in systems with time dependent $\tilde{H}$). If such a single bath/external drive couples to a finite fraction of all sites/fields in the system $\mathcal{S}$ so as to lower the average energy density then even fields that are spatially far apart become correlated by virtue of their non-local interaction with the common environment $\mathcal{E}$ (their shared bath or external drive). The full Hamiltonian $\tilde{H}$ describing the system $\mathcal{S}$ and its environment $\mathcal{E}$ (including the coupling between $\mathcal{S}$ and $\mathcal{E}$) provides the full time evolution $\tilde{\mathcal{U}}(t)$ for the initial density matrix $\tilde{\rho}$ on $\mathcal{I} = \mathcal{S} \cup \mathcal{E}$. We may trace or "integrate" over the bath/drive degrees of freedom in $\mathcal{E}$ (accounting for the driving (as well as dissipation) due to coupling to the environment) to arrive, for quantum systems, at the Schrodinger picture reduced density matrix $\rho_{\mathcal{S}}$ depending only on the degrees of freedom in $\mathcal{S}$. Thus, we consider

$$
\rho_{\mathcal{S}}(t) \equiv Tr_{\mathcal{E}}(\tilde{\mathcal{U}}(t)\tilde{\rho}\tilde{\mathcal{U}}^{\dagger}(t)),
$$

$$
\tilde{\mathcal{U}}(t) = \mathcal{T} \exp(-\frac{i}{\hbar} \int_0^t \tilde{H}(t')dt') \tag{4}
$$

$$
\equiv \mathcal{T} \exp(-\frac{i}{\hbar} \int_0^t (H_{\mathcal{S}}(t') + H_{\mathcal{E}}(t') + H_{\mathcal{S}-\mathcal{E}}(t'))dt').
$$

Here, $\mathcal{T}$ denotes time ordering, and three Hamiltonians (i) $H_{\mathcal{S}}$, (ii) $H_{\mathcal{E}}$, and (iii) $H_{\mathcal{S}-\mathcal{E}}$ describe, respectively, (i) the Hamiltonian involving only the degrees of freedom in $\mathcal{S}$, (ii) the Hamiltonian involving degrees of freedom in $\mathcal{E}$ alone, and (iii) the interaction between the system and its environment. $H_{\mathcal{S}-\mathcal{E}}$ may capture the coupling between different, far separated, fields (say at sites $i$ and $j$) in the system $\mathcal{S}$ to the same external drive/environment $\mathcal{E}$. The trace in the first line of Eq. (4) over the refrigerator/heater or other external drive degrees of freedom $\mathcal{E}$ may generate a correlation between these two fields at $i$ and $j$ irrespective of their spatial separation. This correlation in $\rho_{\mathcal{S}}(t)$ between spatially distant fields may arise, rather universally, if in $H_{\mathcal{S}-\mathcal{E}}$ the latter two fields couple to the very same external drive or environment $\mathcal{E}$. For a uniform external drive, the coupling between all fields in $\mathcal{S}$ to those in $\mathcal{E}$ is of typical comparable strength. Thus, the resulting correlation in $\rho_{\mathcal{S}}(t)$ may be *non-local* even at short times $t$ (so long as at that these (or earlier) times, a finite fraction of the fields in $\mathcal{S}$ couple to the external drive/bath $\mathcal{E}$). A semi-classical motivation for this effect is sketched in Appendix D. As alluded to in procedure (ii) of Section II, in real physical systems the form of the microscopic interactions is time independent (corresponding to a time independent $\tilde{H}$ in Eq. (4).

In relativistic theories, a strict minimal cutoff time $t_{\min}$ for a finite fraction of the fields in $\mathcal{S}$ to become coupled to an external drive/bath $\mathcal{E}$ is set by $t_{\min} = \ell_{\min}/c$. Here,

$\ell_{\min}$ is the minimal linear distance between the "center of mass" of $\mathcal{S}$ and the nearest point in $\mathcal{E}$ and $c$ is the speed of light for bona fide radiative coupling that changes the energy density $\epsilon$ (or temperature) of the system. Thus, since $\ell_{\min} = \mathcal{O}(L)$ for, e.g., radiative coupling to the environment, this minimal time $t_{\min} = \mathcal{O}(L/c)$ (as further discussed in Appendix A while paying attention to absorption lengths). For generic spin models and other non-relativistic local theories, a similar bound on $t_{\min}$ on the time required for the environment to couple with a typical uniform strength or become entangled with a finite fraction of the sites in $\mathcal{S}$ is set by the effective (Lieb-Robinson (LR)) speed $v_{\mathsf{LR}}$ [8–11, 27] ($t_{\min} = t_{\mathsf{LR}} = \mathcal{O}(L/v_{\mathsf{LR}})$). In all cases (relativistic or non-relativistic) $t_{\min} = \mathcal{O}(L/v)$ with $v$ a finite relevant speed. Thus, no long-range correlations violating causality (either relativistic or non-relativistic Lieb-Robinson type) appear. Rather, our results concerning long-range correlations pertain to times $t > t_{\min}$. At such times, the relativistic or Lieb-Robinson light-cones (respectively given by $(ct)$ or $(v_{\mathsf{LR}}t)$) already *span most of the system* $\mathcal{S}$. Indeed, as seen from Eq. (4), long range correlations may be generated from the coupling of the environment $\mathcal{E}$ to the bulk of $\mathcal{S}$. At sufficiently short times, no such coupling exists and, in tandem, the total energy of the system cannot change at a rate proportional to its volume (i.e., at these short times, the rate of change of the energy density vanishes, $d\epsilon/dt = 0$). A system that starts off with only local $G_{ij}$ will require a time $t > t_{\min}$ to develop long range correlations [8, 9] consistent with our new results concerning (i) a required minimal time scale for changing the energy density of the system at a finite rate (Appendix B) and (ii) the appearance of nontrivial correlations once the energy density varies (the central result of this paper). The above also applies to general intensive quantities $q$ different from the energy density. In Section IX, we will sharpen other considerations related to $\tilde{\mathcal{U}}(t)$ to arrive at exact inequalities.

It has long been known that algebraic power law correlations appear in nonuniformly driven systems [28]. The existence of a spatially non-uniform profile of the local energy density may enhance the large fluctuations that we find in the current work. We will briefly touch on related aspects towards the end of Section XII. In classical systems with local interactions, broad distributions of various observables may also occur in the thermodynamic limit when these systems are disordered. This phenomenon is known as "non-self-averaging", e.g., [29–32]. In these disordered systems, an ensemble average of a physical observable computed over different disorder realizations may differ significantly from the expectation value of the same quantity in any single member of the ensemble. The systems that we will focus on in the current work need not be disordered nor critical. However, given the absence of self-averaging in such disordered classical systems, we remark that the broadening that we find will also apply to various systems when the ("ensemble of") eigenstates of the density matrix effectively

describe these different disorder realizations of classical critical systems. This is so since, in such cases, an average computed with the probability density matrix $\rho$ will reproduce the average associated with an ensemble of disordered classical states.

In the driven system, the non-local correlators $G_{ij}$ of Eq. (3) may be finite. This does not imply that other non-local correlators different from those appearing in Eq. (3) cannot be finite in a more general system. By evolving (forward and backwards) in time, one can examine the correlations of general quantities associated $G_{ij}$ in the driven system. Taken together, Eqs. (3, 4) allow for other non-local covariances to be finite. Specifically, whenever Eq. (3) holds, regarded as a formal operator, the Heisenberg picture Hamiltonian $H^H(t) = \tilde{\mathcal{U}}^\dagger(t) H \tilde{\mathcal{U}}(t)$, evaluated for times $t$ at which Eq. (3) applies, will trivially, exhibit a standard deviation that is $\mathcal{O}(N)$ when computed with the initial density matrix $\tilde{\rho}$ at (i.e., prior to driving the system). The proof of this assertion is straightforward. If $\langle H^H(t)\rangle = Tr(\tilde{\rho} H^H(t))$ then,

$$Tr\Big[\tilde{\rho}(H^H(t) - \langle H^H(t)\rangle)^2\Big] =$$
$$Tr\Big[\tilde{\mathcal{U}}(t)\tilde{\rho}\tilde{\mathcal{U}}^\dagger(t)(H - \langle H^H(t)\rangle)^2\Big]. \qquad (5)$$

Whenever Eq. (3) holds,

$$Tr\Big[\tilde{\rho}(H^H(t) - \langle H^H(t)\rangle)^2\Big] = \mathcal{O}(N^2). \qquad (6)$$

Thus, rather trivially, when evaluated with the initial probability density matrix $\tilde{\rho}$, the operator $(H^H(t) - \langle H^H(t)\rangle)$ exhibits an $\mathcal{O}(N^2)$ variance. This allows for non-local correlations similar to those in Eq. (3) for operators different from $H$ also at initial times before the system is driven. In special cases, when $H^H(t)$ will remain a sum of local terms similar to those in Eqs. (1), the simple derivation of Eq. (3) may imply non-local correlations for operators do not appear in the Hamiltonian $H$ at time $t = 0$. We will indeed precisely encounter such correlations and further elaborate on viable preparation of non-product form type states with these correlations in the example of Section VI (discussed in some detail in Appendices H and I) where the correlations in the initial state assume a particularly simple form.

It should be stressed that in the current work we explain how long range correlations of the particular form of Eq. (3) for the local energetic terms $\{\mathcal{H}_i\}$ may arise when the corresponding energy density $(\frac{1}{N}\sum_i \langle\mathcal{H}_i\rangle)$ changes at a finite rate (and also explain how similar correlations appear when other intensive quantities vary at a finite rate). We do not derive results concerning macroscopic range correlations that are different from Eq. (3). That is, in this paper we only analyze the correlations between the driven observables. For completeness, we must remark that many other nontrivial correlations may appear between quantities that are not driven. Indeed, long range correlations may even appear in equilibrated systems. As has been long known, systems such as the celebrated

AKLT spin chains [33–37] may indeed display nontrivial long range correlations. The AKLT spin chains exhibit non-trivial long range string correlations in their ground states [34–37] in addition to more mundane conventional short range nematic type correlations [38, 39]. In [38, 39], a general algorithm was provided for the construction of non-vanishing string type and other correlators for general entangled ground states.

In what follows, we first turn to product states where no broad distributions of intensive quantities arise. For product states undergoing an evolution with a locally separable Hamiltonian, the system degrees of freedom cannot couple to a common environment in Eq. (4). In the sections thereafter, we will demonstrate that in general quantum systems (not constrained to a product state structure), broadening may be quite prevalent. Prevalent non-factorizable states generally allow for a coupling to a common environment.

## V. PRODUCT STATES AND BOUNDED SEPARABLE HAMILTONIANS

Prior to demonstrating that energy density broadening naturally accompanies a cooling or heating of the system, we first discuss (within the framework of procedure (1) of Section II for which the detailed considerations of Eq. (4) (Figure 3) do not apply) states associated with individually decoupled local subsystems. Our focus is on systems with separable bounded local interactions. For a density matrix $\rho(t)$ that, at a time $t$, is a direct tensor product of local density matrices $\{\rho_l(t)\}_{l=1}^M$ acting on disjoint spaces, with $M = \mathcal{O}(N)$,

$$\rho(t) = \rho_1(t) \otimes \rho_2(t) \otimes \cdots \otimes \rho_M(t), \qquad (7)$$

the standard deviation $\sigma_H$ of the Hamiltonian of Eq. (1) at this time will, in accord with the central limit theorem, generally be $\mathcal{O}(\sqrt{N})$ even when the rate of change of the energy $dE/dt$ may be extensive (i.e., $\propto N$). This result applies to both quantum and classical systems. In classical theories, $\{\rho_l\}$ portray the probability distributions of decoupled local degrees of freedom and Eq. (7) describes independent degrees of freedom $l$. In the quantum arena, Eq. (7) also describes states in which no entanglement exists.

As a case in point, we may consider the initial (spin $S = 1/2$) state $|\psi^0_{Ising}\rangle = |s_1^0 s_2^0 \cdots s_N^0\rangle$ to be a low energy eigenstate of an Ising model $H_I = -\sum_{\langle ij\rangle} J_{ij} S_i^z S_j^z$ that is acted on during intermediate times by a transverse magnetic field Hamiltonian ($H_{tr} = -B_y(t)\sum_i S_i^y$) that causes a precession around the $S^y$ axis and thus alters the energy as measured by $H_I$ (thereby heating or cooling the system). Here, $s_i = \pm 1$ denote the scaled eigenvalues of the local spin operators $S_i^z$. The transverse field Hamiltonian $H_{tr}$ may be explicitly written a sum of decoupled terms each of which acts on a separate local subspace, $H_{tr} \equiv \sum_{i=1}^M \mathcal{H}_i$, with $M = N$. The initial state $|\psi^0_{Ising}\rangle$ (and its associated density matrix)

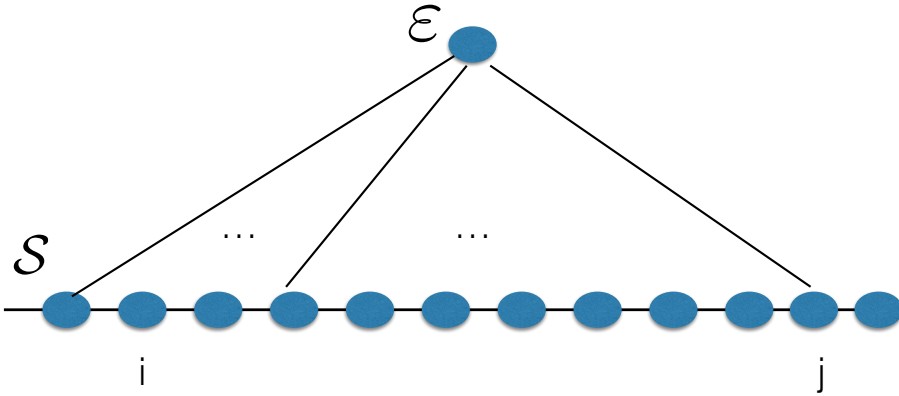

FIG. 3. An intuitive representation of the effect. In order to drive the system $\mathcal{S}$ and vary its energy density at a finite rate, the environment ($\mathcal{E}$) must couple to a finite fraction of the number of sites in $\mathcal{S}$ (e.g., sites $i$ and $j$). The energy fluctuations at both $i$ and $j$ are correlated with $\mathcal{S}$. This, consequently, allows for non-trivial correlations between the local energy fluctuations (those of $\mathcal{H}_i$ and $\mathcal{H}_j$ of Eq. (1)) even when $i$ and $j$ are far apart. As we will explain in Appendix B, in non-relativistic systems with local interactions, causality in the form of the Lieb-Robinson bounds [27] mandates that a minimal time must elapse before an external drive may couple to sites in the bulk of the system $\mathcal{S}$. Physical estimates on lower bounds on minimal times are further briefly discussed in Appendix A.

can be written as an outer product of $M = N$ single spin states (density matrices) defined on the same $M$ decoupled separate spaces. Thus an evolution, from an initial product state, with $H_{tr}$ will trivially lead to a final state which still is of the product state form. All product states $|\psi\rangle = |s_1 s_2 \cdots s_N\rangle$ are eigenstates of $H_I$. A uniform rotation, between an initial time ($t = 0$) and a final time $t_f$, of all of the $N$ spins around the $y$ spin axis by the transverse field Hamiltonian $H_{tr}$ by an angle of $\pi/2$ will transform $|\psi_{Ising}^0\rangle$ to a final state $|\chi\rangle$ that is an equal modulus superposition of all Ising product states (all eigenstates of $H_I$), viz.,

$$|\chi\rangle = 2^{-N/2} \sum_{s_1 s_2 \cdots s_N} (-1)^{\sum_{i=1}^{N}(\delta_{s_i^0,-1}\delta_{s_i,-1})}|s_1 s_2 \cdots s_N\rangle,$$

with $\delta_{\sigma_i,\sigma_j}$ the Kronecker delta. We next discuss what occurs when the exchange constants $J_{ij}$ are of finite range but are otherwise arbitrary. The standard deviation of the energy (i.e., the standard deviation of $H_I$) associated with this final rotated state (and any other state during the evolution) of the initial Ising product state scales as $\mathcal{O}(\sqrt{N})$ while the energy change can be extensive [40]. The state $|\chi\rangle$ corresponds to the infinite temperature limit of the classical Ising model of $H_I$ (its energy density is equal to that of the system at infinite temperature and similarly all correlation functions vanish). A key point is that generic finite temperature states are *not* of the type of Eq. (7). In fact, general thermal states (i.e., eigenstates of either local or nonlocal Hamiltonians that are elevated by a finite energy density difference relative to the ground state) typically display volume law entanglement entropy [41–44] in agreement with the Eigenstate Thermalization Hypothesis [45–53] while ground states and many body localized states of arbitrarily high energy [54–62] may exhibit area law entropies [63]. The en-

tanglement entropy of individual quantum "thermalized" states imitates the conventional thermodynamic entropy of the macroscopic system that they describe [64]. In order to further elucidate these notions, in Appendix E, we illustrate that correlations in finite energy density eigenstates of the Ising chain mirror those in equilibrated Ising chains at positive temperatures. In the one dimensional Ising model and other equilibrium systems at temperatures $T > 0$, the high degree of entanglement and mixing between individual product states leads to contributions to the two point correlation functions that alternate in sign and ultimately lead to the usual decay of correlations with distance. Our central thesis is that an external driving Hamiltonian (such as that present in cooling/heating of a system) may lead to large extensive fluctuations. While the appearance of such extensive fluctuations may seem natural for non-local operators (such as (Heisenberg picture) time evolved local Hamiltonian terms in various examples), these generic fluctuations may also appear for local quantities (e.g., the local operators $\{\mathcal{H}_i\}$ in Eq. (3)). In Section VI, we will study systems for which the relevant (Heisenberg picture) operators $\{\mathcal{H}_i\}$ are, indeed, local.

When all of the eigenvectors of the density matrix are trivial local product states that do not exhibit entanglement, the system described by $\rho$ is a classical system (with different classical realizations having disparate probabilities). In the next sections, we will demonstrate that large fluctuations of any observable may naturally arise for all system sizes (including systems in their thermodynamic limit). The calculations in the studied examples will be for single quantum mechanical states. Any density matrix (also that capturing a system having a mixed state in any region $\mathcal{S}$) may be expressed as $\rho = |\psi\rangle\langle\psi|$ with a pure state $|\psi\rangle$ that extends over a

volume $\mathcal{I}' \supset \mathcal{S}$ [65, 66].

As suggested in Section IV, the physics underlying our effect may be realized in both quantum and classical systems. As demonstrated above, in the quantum arena, entanglement is mandatory for a non-vanishing $\sigma_\epsilon$. Accordingly, our analysis will naturally allow for entangled quantum states. Physically, these entangled states effectively describe situations wherein the generic evolution operator or the environment $\mathcal{E}$ in Eq. (4) are non-factorizable. Thus, for generic entangled states, ensuing long-range coupling/correlations between the sites in $\mathcal{S}$ may result.

## VI. DUAL EXAMPLES

The existence of finite connected correlations $|G_{ij}|$ (Eq. (3)) for far separated sites $|i - j| \to \infty$ is at odds with common lore. Before turning to more formal general aspects, we illustrate how this occurs in two classes of archetypical systems- (i) any globally $SU(2)$ symmetric (arbitrary graph or lattice) spin $S = 1/2$ model in an external magnetic field (discussed next in Section VI A) and (ii) dual hard core Bose systems on the same graphs or latices (Section VI B). Although (i) and (ii) constitute two well known (and very general) intractable many-body theories, as we will demonstrate, the analysis of the fluctuations becomes identical to that associated with an integrable one body problem. In the context of example (i), this effective single body problem will be associated with the total system spin $\vec{S}_{tot}$. This simplification will enable us to arrive at exact results. Similar to Section V, the analysis below will be performed within the framework of procedure (1) of Section II- that of an explicitly time varying Hamiltonian in a closed system having no environment.

### A. Rotationally invariant spin models on all graphs (including lattices in general dimensions)

In what follows, we consider a general rotationally symmetric spin model ($H_{symm}$) of local spin-$S$ moments augmented by a uniform magnetic field.

$$H_{spin} = H_{symm} - B_z \sum_i S_i^z. \qquad (8)$$

Amongst many other possibilities, the general rotationally symmetric Hamiltonian $H_{symm}$ may be a conventional spin interaction of the type

$$\begin{aligned} H_{Heisenberg} = &-\sum_{ij} J_{ij}\vec{S}_i \cdot \vec{S}_j \\ &- \sum_{ijkl} W_{ijkl}(\vec{S}_i \cdot \vec{S}_j)(\vec{S}_k \cdot \vec{S}_l) + \cdots, \end{aligned} \qquad (9)$$

with arbitrary Heisenberg spin exchange couplings $\{J_{ij}\}$ augmented by conventional higher order rotationally

symmetric terms. We reiterate that the model of Eq. (8) is defined on any graph (including lattices in *any number of spatial dimensions*).

#### 1. Quantum Spin System

In the upcoming analysis, we will label the eigenstates of $H_{spin}$ (and their energies) by $\{|\phi_\alpha\rangle\}$ (having, respectively, energies $\{E_\alpha\}$). We will employ the total spin operator $\vec{S}_{tot} = \sum_{i=1}^N \vec{S}_i$. Since $[\vec{S}_{tot}, H_{spin}] = 0$, all eigenstates of $H_{spin}$ may be simultaneously diagonalized with $S_{tot}^z$ (with eigenvalue $m\hbar$) and $\vec{S}_{tot}^2$ (with eigenvalue $S_{tot}(S_{tot} + 1)\hbar^2$). Thus, any eigenstate of Eq. (8) may be written as $|\phi_\alpha\rangle = |v_\alpha; S_{tot}, S_{tot}^z\rangle$ with $v_\alpha$ denoting all additional quantum numbers labeling the eigenstates of $H_{spin}$ in a given sector of $S_{tot}$ and $S_{tot}^z$ [67]. Although our results apply for local spins of any size $S$, in order to elucidate certain aspects, we will often allude to spin $S = 1/2$ systems. For any eigenstate having a general $S_{tot}^z \neq \pm S_{max} = \pm NS$, the associated density matrix is not of the local tensor product form of Eq. (7). Rather, any such eigenstate is a particular superposition of spin $S = 1/2$ product states having a total fixed value of $S_{tot}^z$. The state of maximal total spin $S_{tot} = S_{max}$ (which can be trivially shown to be a non-degenerate eigenstate for any value of $S_{tot}^z$, see Appendix G) corresponds to a symmetric equal amplitude superposition of all such product states of a given $S_{tot}^z$ (i.e, such a sum of all product states of the type $|\uparrow_1\uparrow_2\downarrow_3\uparrow_4\downarrow_5\uparrow_6 \cdots \uparrow_{N-1}\downarrow_N\rangle$ in which there are a total of $(N/2 \pm S_{tot}^z/\hbar)$ single spin of up/down polarizations along the $z$ axis). We set an arbitrary eigenstate $|\phi_\alpha\rangle$ to be the initial state (at time $t = 0$) of the system $|\psi_{Spin}^0\rangle$. The energy density (and the global energy itself) will have a vanishing standard deviation in any such initially chosen eigenstate, $\sigma_\epsilon = 0$. We next evolve this initial ($t = 0$) state via a "cooling/heating process" wherein the energy (as measured by $H_{spin}$) is varied by replacing, during the period of time in which the system is cooled or heated, the Hamiltonian of Eq. (8) by a time dependent transverse field Hamiltonian (see Section VI A 3 for restrictions imposed by causality)

$$H_{tr}(t') = -B_y(t') \sum_i S_i^y. \qquad (10)$$

At $t = 0$, the system Hamiltonian varies instantaneously (a particular realization of procedure (1) of Section II) from $H_{spin}$ to $H_{tr}$. Once the "cooling/heating process" terminates at a final time ($t = t_f$), the system Hamiltonian becomes, once again, the original Hamiltonian of Eq. (8). Once again, in this case, the change of the Hamiltonian at the final time $t_f$ is instantaneous. In accord with the discussion in Section IV, in Eq. (10), a finite fraction (in this case all) of the system degrees of freedom (i.e., the spins) couple to the external drive/bath (the external transverse field). Such a global coupling is necessary to achieve a finite $d\epsilon/dt$. During the evolution with $H_{tr}$, the spins globally precess about

the $y$ axis. Thus, after a time $t$, the energy per lattice site is changed (relative to its initial value $\epsilon_0$) by an amount $\epsilon(t_f) - \epsilon_0 = B_z \frac{S_{tot}^z}{N}(1 - \cos\theta(t_f))$. Here, $\theta(t) \equiv \int_0^t B_y(t') \, dt'$. In the terminology of [13–16, 22], this energy density shift represents the work done per site. When $B_z S_{tot}^z > 0$, the energy density of the system is generally increased relative to its initial value while for negative $B_z S_{tot}^z$, the system is "cooled" relative to its initial energy density. For all $S_{tot}^z$, the energy density $\epsilon(t)$ exhibits consecutive cooling and heating periods. Employing the shorthand $w \equiv S_{tot}^z/(\hbar S_{tot})$, the standard deviation of the energy density of $\frac{H_{spin}}{N}$ is [68]

$$\sigma_\epsilon(t_f) = \frac{S_{tot}\hbar|B_z \sin\theta(t_f)|}{N\sqrt{2}}\sqrt{1 + \frac{1}{S_{tot}} - w^2}. \quad (11)$$

A comparable standard deviation appears not only for a single eigenstate of $H_{spin}$ but also for any other initial states having an uncertainty in the total energy that is not extensive. When $w = 1$ (or $-1$) with the total spin being maximal, $S_{tot} = S_{max}$, the initial state $|\psi_{Spin}^0\rangle$ is a product state of all spins being maximally up (or all spins pointing maximally down) [69]. Away from this singular $S_{tot}^z = \pm\hbar S_{max}$ limit, spatial long range entanglement develops. When $(1 - |w|) = \mathcal{O}(1)$, the scaled standard deviation of the energy density is, for general times, $(\frac{1}{\hbar B_z})\sigma_\epsilon = \mathcal{O}(1)$ and, as we will elucidate in Appendix F 1, a macroscopic (logarithmic in system size) entanglement entropy appears. A comparable standard deviation $\sigma_\epsilon$ appears not only for the eigenstate but also for states initial having an energy uncertainty of order $\mathcal{O}(1)$ (in units of $B_z\hbar$) (e.g., $c_1|S_{tot}, S_{tot}^z\rangle + c_2|S_{tot}, S_{tot}^z - \hbar\rangle$ with $c_{1,2} = \mathcal{O}(1)$). In the following, we briefly remark on the simplest case of a constant (time indeodent) $B_y$. Here, the time required to first achieve $\frac{1}{B_z\hbar}\sigma_\epsilon = \mathcal{O}(1)$ starting from an eigenstate of $H_{spin}$ is $\mathcal{O}(1/B_y)$. This requisite waiting time is independent of the system size (as it must be in this model where a finite $\sigma_\epsilon$ is brought about by the sum of local decoupled transverse magnetic field terms in $H_{tr}$). The large standard deviation implies (Eq. (3)) that long range connected correlations of $S_i^z$ emerge once the state is rotated under the evolution with $H_{tr}$. This large standard deviation of $\frac{1}{N}\sum_{i=1}^N S_i^z$ appears in the rotated state displaying (at all sites $i$) a uniform value of $\langle S_i^z\rangle$. Even though there are no connected correlations of the energy densities themselves in the initial state, the non-local entanglement enables long range correlations of the local energy densities once the system is evolved with a transverse field. The variance $\sigma_\epsilon$ should not, of course, be confused with the spread of energy densities that the system assumes as it evolves (e.g., for the $S_{tot}^z = 0$ state, $\sigma_\epsilon = \mathcal{O}(1)$ while the energy density $\epsilon(t)$ does not vary with time). We nonetheless remark that the standard deviation $\sigma_\epsilon$ vanishes at the discrete times $t_k = k\pi/B_y$ (with $k$ an integer)- the very same times where the rate of change of the energy density $\epsilon(t)$ is zero. Indeed, in our model system, up to important time independent multiplicative factors, $\sigma_\epsilon \propto |\frac{d\epsilon(t)}{dt}|$.

Before performing further analysis, we make explicit one comment (briefly mentioned in Section III):

In line with the discussion following Eqs. (5, 6), non-product like eigenstates of $H_{spin}$ (the initial ($t = 0$) states of our example) having a total ferromagnetic moment ($S_{tot}^z = \mathcal{O}(N)$) and $|w| < 1$ may already exhibit nonlocal correlations. In Appendix H, we detail these correlations and further explain that *such correlations are unavoidable* if (in the spirit of the current work) general averages of derivatives of the energy density in the driven system need to be finite. Specifically, if either the second order derivatives of the energy density and/or higher order cumulants of the energy density derivatives are finite then these nonlocal correlations must appear. In Appendix I, we briefly discuss a Gedanken experiment for constructing such states.

We now turn to the higher order moments of the fluctuations of the $t > 0$ states evolved with Eq. (10), $\langle(\Delta\epsilon)^p\rangle \equiv \frac{1}{N^p}\langle(H_{spin}^H - \langle H_{spin}^H\rangle)^p\rangle$ with $p > 2$. (The standard deviation of Eq. (11) corresponds to $p = 2$.) Here, $H_{spin}^H(t) = (\mathcal{T}e^{\frac{i}{\hbar}\int_0^t iH_{tr}(t')dt'})H_{spin}\mathcal{T}(e^{-\frac{i}{\hbar}\int_0^t H_{tr}(t')dt'})$ is the Heisenberg picture Hamiltonian and the expectation value is taken in the initial state $|\psi_{Spin}^0\rangle$. If $N \gg 1$ and $1 > |w|$ then $S_\pm^{tot}|S_{tot}, S_{tot}^z\rangle = \hbar\sqrt{S_{tot}(S_{tot}+1) - m(m\pm1)}|S_{tot}, m \pm 1\rangle \sim S_{tot}\hbar\sqrt{1-w^2}|S_{tot}, m \pm 1\rangle$ where $S_{tot}^z = m\hbar$. Trivially, for all $m$ and $m'$, the matrix element of $\delta S_{tot}^z \equiv S_{tot}^z - \langle S_{tot}^z\rangle$ between any two eigenstates, $\langle S_{tot}, m|\delta S_{tot}^z|S_{tot}, m'\rangle = 0$. Thus, the only non-vanishing contributions to $\langle(\Delta\epsilon)^p\rangle$ stem from $\langle(S_{tot}^x)^p\rangle$. This expectation value may be finite only for even $p$. Thus, in what follows, we set $p = 2g$ with $g$ being a natural number. For $S_{tot} = \mathcal{O}(N)$, when expressing the expectation value of $\langle(\Delta\epsilon)^{2g}\rangle$ longhand in terms of spin raising and lowering operators, one notices that, in this large $N$ limit, each individual term containing an equal number of raising and lowering operators yields an identical contribution (proportional to $(S_{tot}\hbar\sqrt{1-w^2})^{2g}$) to the expectation value $\langle(\Delta\epsilon)^{2g}\rangle$. Since there are $\binom{2g}{g}$ such contributions, for all $g \ll N$ in the thermodynamic ($N \to \infty$) limit, the expectation value $\langle(\Delta\epsilon)^{2g}\rangle = \binom{2g}{g}(\frac{\sigma_\epsilon^2}{2})^g$. We write the final (Schrodinger picture) state at time $t = t_f$ as $|\psi_{Spin}\rangle = \sum_\alpha c_\alpha|\phi_\alpha\rangle$. The probability distribution of the energy density of Eq. (2) reads

$$P(\epsilon') = \sum_\alpha |c_\alpha|^2\delta(\epsilon' - \frac{E_\alpha}{N}). \quad (12)$$

In this example, the Heisenberg picture Hamiltonian $H_{spin}^H$ (and the associated operators $\mathcal{H}_i$) remains local for all times. In general systems, the time evolved Heisenberg picture Hamiltonian need not be spatially local. Eq. (12) describes the probability distribution associated with the "wave packet" intuitively discussed in Section IV (a "packet" that is now given by the amplitudes $\{c_\alpha\}$ in our eigenvalue decomposition of the final state $|\psi_{Spin}\rangle$). The averaged moments of $\Delta\epsilon' \equiv (\epsilon' - \epsilon)$ are $\langle(\Delta\epsilon)^{2g}\rangle = \int d\epsilon' P(\epsilon')(\epsilon'-\epsilon)^{2g}$. Here, as throughout, $\epsilon = $

$\frac{1}{N}\langle\psi_{Spin}|H_{spin}|\psi_{Spin}\rangle = -(\sum_{ij}J_{ij}+B_z S_{tot}^z\cos\theta(t_f))/N$ is the energy density in the final state. More generally, the expectation value of a general function $f(\frac{H_{spin}^H}{N})$ in the state $|\psi_{Spin}^0\rangle$ (or, equivalently, of $f(\frac{H_{spin}}{N})$ in the above defined final Schrodinger picture state $|\psi_{Spin}\rangle$) is given by $\langle f(\frac{H_{spin}^H}{N})\rangle = \int d\epsilon' f(\epsilon')P(\epsilon')$. The mean value of each Fourier component $e^{iq(\Delta\epsilon')}$ when evaluated with $P(\epsilon')$ is [70]

$$\langle e^{iq(\Delta\epsilon')}\rangle = J_0(q\sigma_\epsilon\sqrt{2}), \tag{13}$$

where $J_0$ is a Bessel function. An inverse Fourier transformation then yields

$$P(\epsilon') = \frac{\theta(\sigma_\epsilon\sqrt{2}-|\Delta\epsilon'|)}{\pi\sigma_\epsilon\sqrt{2-\frac{(\Delta\epsilon')^2}{(\sigma_\epsilon)^2}}}. \tag{14}$$

Here, as earlier, $\Delta\epsilon'$ denotes the difference between $\epsilon'$ and the value of the energy density $\epsilon(t)$. The Heaviside function $\theta(z)$ in Eq. (14) captures the fact that the spectrum of $H_{spin}$ is bounded. Similar results apply to boundary couplings [71]. The distribution of Eq. (14) may also be rationalized geometrically as we will shortly discuss (Eq. (17)). Comparing our result of Eq. (14) to known cases, we remark that, where it is non-vanishing, the distribution of Eq. (14) is the reciprocal of the Wigner's semi-circle law governing the eigenvalues of random Hamiltonians and the associated distributions of Eq. (12), e.g., [72]. We stress that Eq. (14) is exact for the general spin Hamiltonians of Eqs. (8,10) and does not hinge on assumed eigenvalue distributions of effective random matrices.

Performing additional calculations, we find qualitatively similar results for analogous "cooling/heating" protocols. For instance, one may consider, at intermediate times $0 \le t \le t_f$, the Hamiltonian governing the system to be that of a time independent $H_{tr}$ (i.e., one with a constant $B_y(t) = B_y$) augmenting $H_{spin}$ instead of replacing it. That is, we may consider, at times $0 \le t \le t_f$, the total Hamiltonian to be

$$H_a = H_{spin} + H_{tr}. \tag{15}$$

For such an augmented total Hamiltonian $H_a$, the total spin $\vec{S}_{tot}$ precesses around direction of the applied external field $(B_y\hat{e}_y+B_z\hat{e}_z) \equiv B\hat{e}_n$. An elementary calculation analogous to that leading to Eq. (11) then demonstrates that the corresponding standard deviation $\sigma_\epsilon^a$ of the energy density at $t=t_f$,

$$\sigma_\epsilon^a(t=t_f) = \frac{|B_z B_y| S_{tot}\hbar}{NB\sqrt{2}}\sqrt{1+\frac{1}{S_{tot}}-w^2}$$
$$\times\sqrt{\sin^2(Bt_f)+\frac{B_z^2(1-\cos(Bt_f))^2}{B^2}}. \tag{16}$$

We wish to stress that if $S_{tot} = \mathcal{O}(N)$ and $|w| < 1$ then, as in Eq. (11), the standard deviation $\sigma_\epsilon^a = \mathcal{O}(N)$ for general times $t_f$. The distribution of the energy density

following an evolution with this augmented Hamiltonian will, once again, be given by Eq. (14) for macroscopic systems of size $N \to \infty$. The reader can readily see how such spin model calculations may be extended to many other exactly solvable cases. The central point that we wish to underscore is that *a broad distribution of the energy density*, $\sigma_\epsilon \ne 0$, is obtained in all of these exactly solvable spin models in general dimensions.

### 2. Semi-classical spin systems and a geometrical interpretation

The results that we just derived are valid for any spin $S$ realization of the Hamiltonians of Eqs. (8, 10). The standard deviations of Eqs. (11, 16) remain finite for all $S$ (with a scale set by the external magnetic field energies in these Hamiltonians). As long known [73, 74], the $S \to \infty$ limit yields classical renditions of respective quantum spin models. Thus, the finite standard deviation of the energy density in individual eigenstates (Eqs. (11, 16)) and in thermal states formed by these eigenstates implies that the standard deviation of the energy density *remains finite in the classical limit* (as was suggested by the general arguments associated with Eq. (4)). More strongly, all that mattered in our earlier calculation of Section VI A 1 were the $S_{tot}$ and $S_{tot}^z$ values. If $S_{tot} = \mathcal{O}(N)$ then even if the size of the spin $S$ at each lattice site is small, the total system spin $\vec{S}_{tot}$ is a macroscopic classical quantity and our results may be reproduced by a computation for semi-classical spins. Indeed, an explicit calculation for classical spin states trivially illustrates that a finite standard deviation $\sigma_\epsilon > 0$ may arise in semi-classical systems [75]. To make this explicit, we now perform such a computation. This rather elementary calculation will link the geometry of the manifold of possible $S_{tot}^z$ values to the full distribution $P(\epsilon')$ of the possible energy densities. Towards this end, we parameterize the semi-classical total spin by a vector $\vec{S}_{tot}$ on a sphere of fixed radius $S_{tot}$ (the application of the transverse field Hamiltonian of Eq. (10) does not alter $(\vec{S}_{tot})^2$). Herein, at any time $t$, the vector $\vec{S}_{tot}$ may correspond, with equal probability, to any vector on a circular ring, see, e.g., Figures 4 and 5.

In Eq. (14), $\Delta\epsilon'$ denotes the difference between $\epsilon'$ and the value of the energy $\epsilon(t)$. At time $t$, along a ring (see, e.g., Figure 5), that is further parameterized by an azimuthal angle $\varphi'$, the possible values of $S_z$ are given by $S_{tot}^z(\varphi',t) = \langle S_{tot}^z(t)\rangle + S_{tot}\sqrt{1-w^2}\sin\theta(t)\cos\varphi'$. Here, $\theta(t)$ becomes the polar angle of the center of mass of the ring (i.e., $\theta(t)$ is the angle between (i) a vector connecting the origin to the center of the center of the ring (see, e.g., Figures 4 and 5) and (ii) a vector along the positive $S_{tot}^z$ axis). The expectation value $\langle S_{tot}^z\rangle$ is that of $S_{tot}^z$ in the time evolved state (classically, it is the average of $S_{tot}^z$ around the full ring ($0 \le \varphi' < 2\pi$) at time $t$), i.e., $S_{tot}^z(t) = S_{tot}^z\cos\theta(t)$. The possible values of $S_z(\varphi')$ appear symmetrically twice in the interval $0 \le \varphi' < 2\pi$. We

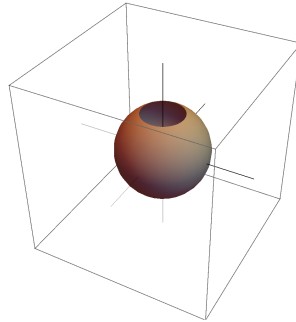

FIG. 4. (Color Online.) Semi-classically, the total spin $\vec{S}_{tot}$ may, with equal probability, correspond to any vector connecting the origin of a sphere of radius $\hbar S_{tot}$ to a point along a ring forming "a line of latitude". In the figure above, this "line of latitude" ring is defined by boundary of the shaded spherical cap near the "north pole". All points along the line of latitude share the same value of $S_{tot}^z$. Here, in the initial state, the polar angle $\theta = 0$.

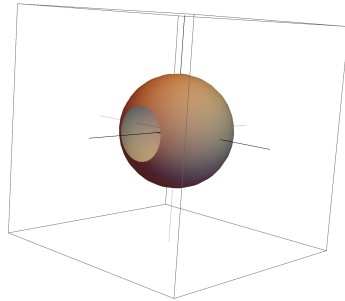

FIG. 5. (Color Online.) Applying the transverse field of Eq. (10) to the ring of Figure 4 leads to precession about the $S_{tot}^y$ axis. For the above displayed ring, $\theta(t) = \pi/2$. During the precession, the semi-classical total spin vectors $\vec{S}_{tot}$ on the ring acquire a range of possible $S_{tot}^z$ values leading to the standard deviation $\sigma_\epsilon$ of the energy density of Eq. (8). The simple (semi-classical) calculation of Eq. (17) for the distribution of $S_{tot}^z$ values for such a uniform ring leads anew to Eq. (14).

may thus consider only $0 \leq \varphi' < \pi$. By the normalization of the probability distribution for $\varphi'$ and the corresponding probability distribution for the energy density, $P(\epsilon')d\epsilon' = \frac{d\varphi'}{\pi}$. Thus,

$$P(\epsilon') = \frac{1}{\pi}\left|\frac{d\varphi'}{d\epsilon'}\right| = \frac{N}{\pi\left|B_z\frac{\partial S_{tot}^z(\varphi')}{\partial \varphi'}\right|}. \qquad (17)$$

Combining Eq. (11) (which may derived from a geometric analysis of Figure 5 as we next explain) with Eq. (17) then provides Eq. (14). We may indeed readily calculate the spread $\sigma_{S_{tot}^z}$ of $S_{tot}^z$ values and rationalize the finite standard deviation $\sigma_\epsilon$ of Eq. (11) from simple geometric considerations, when $1/S_{tot}$ is set to zero (the semi-classical limit). Performing a geometric analysis, one finds that $\sigma_{S_{tot}^z} = \frac{R_{\text{ring}}}{\sqrt{2}}|\sin\theta(t)| \equiv R_g|\sin\theta(t)|$ where $R_{\text{ring}} = S_{tot}\hbar\sqrt{1-w^2}$. Here, $R_g$ is the radius of gyration of the ring of Figure 5 (corresponding to $\theta = \pi/2$) about an axis parallel to the $S_{tot}^z$ axis that passes through the center of mass of this ring. The finite radius of gyration $R_g \neq 0$ implies a spread of energy densities $\sigma_\epsilon = \frac{|B_z \sin\theta(t)|R_g}{N} \neq 0$ at general times. This semi-classical result for $\sigma_\epsilon$ coincides with Eq. (11).

Although the Hamiltonian of Eq. (8) is extremely general as are its eigenstates of high total spin $S_{tot} = \mathcal{O}(N)$ (e.g., states of large total spin in typical low temperature ferromagnets), characteristic equilibrium states of this Hamiltonian will correspond to a special subset having $|w| = 1$ (that is, the total spin will be polarized along the externally applied field direction). As we discussed earlier, such equilibrium states will thus emulate product states (in which all individual spins assume the same polarization. Thus, as was indeed evident in Eqs. (11, 16), when $w = \pm 1$, the broadening $\sigma_\epsilon = 0$. In a related vein, the fully polarized state- a coherent spin state on a sphere of radius $S_{tot}$- is rotated "en block" without any other change of the wavefunction under the action of a transverse field. To see the effect for our exactly solvable system, we have to go away from the limit $|w| = 1$. Away from this limit, the state of the system evolves non-trivially. In the parlance of Section IV, when evolving under the transverse field Hamiltonian of Eq. (10), the $|w| \neq 1$ spin state is not merely "translated" (rotated on a sphere of radius $S_{tot}$) with no other accompanying changes. Appendix I discusses a gedanken experiment in which starting from an equilibrium state, one may

apply transverse fields and let the closed system equilibrate anew so as to generate a state $|\psi_{Spin}^0\rangle$ of total spin $S_{tot} = \mathcal{O}(N)$ with $w \neq \pm 1$.

### 3. Causality, correlations, and a finite $\frac{d\epsilon}{dt}$

We now return to the qualitative discussion of Section IV concerning the causal generation of long range correlations in real physical systems. Eq. (4) suggests that long-range correlations emerge from the coupling between an external environment (which we have not explicitly included in our model system) to the system bulk (e.g., the global coupling of Eq. (10)). As we demonstrate in Appendix B, when the environment is included in the analysis of a non-relativistic system obeying Lieb-Robinson type bounds [8–11, 27], a finite rate of variation of the energy density cannot appear at short times $t < t_{\min} = \mathcal{O}(L/v_{\mathsf{LR}})$. Thus, generally, effective global couplings such as those of Eq. (10) cannot appear instantaneously. Without the bulk coupling of Eq. (10) (and ensuing correlations), the system cannot exhibit a finite rate of change of its energy density (i.e., without such a global coupling $\frac{d\epsilon}{dt} = 0$). It is indeed *after long enough times* (such as those implied by the Lieb-Robinson bounds of Appendix B) that a global coupling such as that of Eq. (10) may appear in effective descriptions not explicitly involving an external environment. It is only after such times (when Eq. (10 applies) that the results that we obtained for the correlations hold.

In an equilibrium state of Eq. (8), the total spin will be polarized along the applied field direction and $w = 1$. In such a case, for the realization of various gednaken experiments (e.g., Appendix I), long-range correlations (Appendix H) may indeed appear in the system after a time that scales with the system size.

As noted after Eq. (11), the calculation of the energy density and its standard deviation for the $w = 1$ system evolving under Eq. (10) is identically the same as that for a product state of $S_{tot}/S$ spins. Indeed, consistent with Section V, this effective product state exhibits no spread of the energy density $\sigma_\epsilon = 0$. In tandem with our main thesis concerning a typical trend between the energy changes and long range correlations, for $w \neq \pm 1$ states, at those times at which the energy density changes at a vanishing rate $d\epsilon/dt = 0$ (corresponding to $\theta(t) \equiv 0(\mathsf{mod}\pi)$), the standard deviations of the energy density (and the associated long-range correlations that it implies) also vanishes, $\sigma_\epsilon = 0$.

### B. Itinerant hard core Bose systems

Our spin model of Section VI A can be defined for local spins of any size $S$. The function $P(\epsilon')$ of Eq. (14) characterizing our investigated states in this system is not a very typical probability distribution. However, the non-local entangled character of states having a finite energy

density relative to the ground state is pervasive for thermal states. This model can be recast in different ways. In what follows we focus on the spin $S = 1/2$ realization of Eq. (8). The Matsubara-Matsuda transformation [76, 77] maps the algebra of spin $S = 1/2$ operators onto that of hard core bosons. Specifically, the bosonic number operator at site $i$ is $n_i = b_i^\dagger b_i = 0, 1$ with $b_i$ and $b_i^\dagger$ the annihilation and creation operators of hard core bosons $((b_i^\dagger)^2 = b_i^2 = 0)$. Following this transformation, the spin Hamiltonian of Eq. (8) is converted into its hard core bosonic dual,

$$H_{Bose} = -\sum_{ij} J_{ij}((b_i^\dagger b_j + h.c.) + n_i n_j)$$
$$- \sum_i (B_z - \sum_j J_{ij}) n_i. \quad (18)$$

The above Hamiltonian describes hard core bosons hopping (with amplitudes $J_{ij}$) on the same $d-$dimensional lattice, featuring attractive interactions and a chemical potential set by $(B_z - \sum_j J_{ij})$. Here, the transverse field cooling/heating Hamiltonian $H_{tr}$ transforms into $H_{Bose-doping} = -\frac{iB_y(t)}{2} \sum_i (b_i^\dagger - b_i)$- a Hamiltonian that alters the number of the bosons (thereby "doping" the system). The hard core Bose states are symmetric under all pairwise permutations $P_{ij}$ of the bosons at occupied sites. The bosonic dual of, e.g., the specific spin product state $|\uparrow_1\uparrow_2\downarrow_3\uparrow_4\downarrow_5\uparrow_6 \cdots \uparrow_{N-1}\downarrow_N\rangle$ corresponds to the symmetrized state of a fixed total number of hard core bosons that are placed on the graph (or lattice) sites $(1, 2, 4, 6, \cdots, (N-1))$. Thus, the bosonic dual of an initial spin state $|\psi_{Spin}^0\rangle$ with a total spin $S_{tot} = S_{\max} = N/2$ is an initial hard core Bose state $|\psi_{Bose}^0\rangle$ that is an equal amplitude superstition of all real space product states with the same total number of hard core bosons $(\sum_{i=1}^N n_i = m + \frac{N}{2})$ distributed over the $N$ lattice sites (an eigenstate of $H_{Bose}$ that adheres to the fully symmetric bosonic statistics). Evolving (during times $0 \leq t \leq t_f$) this initial state with $H_{doping}$, the standard deviation of Eq. (11) and the distribution of Eq. (14) are left unchanged, apart from a trivial rescaling by $\hbar$ (e.g., $\sigma_\epsilon^{Bose} = \frac{|B_z \sin\theta(t_f)|}{2\sqrt{2}} \sqrt{1 + \frac{2}{N} - w^2}$ for $S_{tot} = N/2$). Similar to our discussion of the dual spin system of the previous subsection, the finite standard deviation in this energy density (and of the associated particle density $n = \frac{1}{N} \sum_i n_i$) does not imply that the "doping" is, explicitly, spatially inhomogeneous (indeed, at all times, the expectation value of the particle number $\langle n_i \rangle$ stays uniform for all lattice sites $i$).

We conclude this subsection with three weaker statements regarding viable extensions of the rigorous results that we derived thus far for hard core bosonic systems on general graphs (these graphs include lattices in general dimensions).

(a) We may pass from the above lattice theory to *a continuum scalar field theory* in the usual way. On doing so, it is readily seen that for a continuous scaled $\varphi(x)$ field

replacing $(b_i + b_i^\dagger)$, the canonical Hamiltonian density

$$\mathcal{H}[\varphi] = \frac{1}{2}(m^2\varphi^2 + (\nabla\varphi)^2) + u\varphi^4 \qquad (19)$$

qualitatively constitutes a lowest order continuum rendition of the hard core Bose lattice model of Eq. (18) for a system with uniform nearest neighbor couplings $J_{ij}$. A large value of the constant $u$ in generic bosonic $\varphi^4$ field theories of the type of Eq. (19) yields a large local repulsion between the bosonic fields endowing them with hard core characteristics. The continuum analog of $H_{Bose-doping}$ is the volume integral of the momentum conjugate to $\varphi(x)$. Thus, during various continuous changes of the Hamiltonian, such generic scalar field theories (and myriad lattice system described by them) may exhibit the broad $\sigma_\epsilon$ that we derived for some of their lattice counterpart in this subsection.

(b) The models of Eqs. (8,18) were defined on arbitrary graphs (including lattices in general spatial dimensions). Identical results apply for *spineless fermions* on one dimensional chains with non-negative nearest neighbor hopping amplitudes/coupling constants $\{J_{ij}\}$ and analogs of $H_{Bose-doping}$ capturing a non-local coupling of the system to the external bath. These spinless Fermi systems may be trivially engineered by applying the Jordan-Wigner transformation [78] to Eq. (8).

(c) *Phonons in anharmonic solids.* One may apply the Holstein-Primakoff transformation,

$$S_i^+ = \hbar\sqrt{2}\sqrt{1 - \frac{a_i^\dagger a_i}{2S}}a_i, \;\; S_i^- = \hbar\sqrt{2}a_i^\dagger\sqrt{1 - \frac{a_i^\dagger a_i}{2S}},$$
$$S_i^z = \hbar(S - a_i^\dagger a_i), \qquad (20)$$

to express the local spin operators in Eq. (8) in terms of bosonic creation/annhilation operators ($a_i^\dagger$ and $a_i$). The resulting bosonic Hamiltonian may then be expanded in a series in $1/S$ (as in conventional $1/S$ expansions) [79]. When Fourier transformed, the Hamiltonian describes coupled bosonic modes (involving the bosonic creation/annihilation operators $a_k^\dagger$ and $a_k$ at different Fourier modes $k$) such as those of phonons in anharmonic solids. Here, the heating/cooling protocol of Section VI A corresponds to the creation/annihilation of phonons and leads to identical results for $\sigma_\epsilon$. (Contrary to the anharmonic system, in harmonic theories, the eigenstates have a product state form and some of intuition underlying the product states of Section V comes to life. For completeness, we remark that for harmonic systems, the individual interactions terms in Eq. (1) are unbounded unlike those discussed in Section V.) A Schwinger boson representation may similarly express the spin system of Eqs. (8, 10) in terms of bosonic modes.

## VII. DYSON TYPE EXPANSIONS FOR GENERAL EVOLUTIONS

To make progress beyond intuitive arguments and specific tractable systems, we next compute the standard deviation of the energy density (and, by trivial extension, any other intensive quantity $q$). Towards this end, we examine Dyson type expansions for a general non-adiabatic [24] time dependent Hamiltonian $H(t)$ (of which the piecewise constant Hamiltonians $H_{spin}$ and $H_{tr}$ (or $H_{Bose}$ and $H_{doping}$) are particular instances). Our calculation will demonstrate that in general situations, a finite $\sigma_\epsilon$ will arise. Via a Magnus expansion, the general evolution operator, the time ordered exponential $\mathcal{U}(t) = \mathcal{T}\exp(-\frac{i}{\hbar}\int_0^t H(t')dt')$, may be written as $\mathcal{U} = \exp(\Omega(t))$ with $\Omega(t) = \sum_{k=1}^{\infty}\Omega_k(t)$,

$$\Omega_1(t) = -\frac{i}{\hbar}\int_0^t dt_1 H(t_1),$$

$$\Omega_2(t) = -\frac{1}{2\hbar^2}\int_0^t dt_1 \int_0^{t_1} dt_2 [H(t_1), H(t_2)],$$

$$\Omega_3(t) = \frac{i}{6\hbar^3}\int_0^t dt_1 \int_0^{t_1} dt_2 \int_0^{t_2} dt_3 \Big([H(t_1),[H(t_2),H(t_3)]]$$
$$+ [H(t_3),[H(t_2),H(t_1)]]\Big),$$
$$\cdots. \qquad (21)$$

We may apply the above Magnus expansion to a Heisenberg picture operator $A^H(t) = \mathcal{U}^\dagger A\mathcal{U}$, with $A$ an arbitrary fixed operator, with the above $\Omega(t)$ and subsequently invoke the Baker-Campbell-Hausdorff formula $e^{-\Omega}Ae^{\Omega} = A - [\Omega, A] + \frac{1}{2!}[\Omega,[\Omega, A]] - \frac{1}{3!}[\Omega,[\Omega,[\Omega, A]]] + \cdots$. If no change occurs at intermediate times $t$ and the Hamiltonian is that of the initial system (i.e., $H(t) = H$) then, of course, the standard deviation $\sigma_\epsilon(t)$ will remain unchanged when computed with the (time independent) equilibrium density matrix for which it trivially vanishes. Similarly, if the evolution of $H(t)$ is adiabatic at all times then no broadening of the distribution $P(\epsilon')$ will arise. Our interest, however, lies in the Hamiltonians $H(t) \neq H$ necessary to elicit a change of the energy density $d\epsilon/dt \neq 0$ in a macroscopic system. In particular, we wish to examine the variance of the total energy density,

$$\sigma_\epsilon^2(t) = \frac{1}{N^2}\Big(Tr(\rho(H^H(t))^2) - (Tr(\rho H^H(t)))^2\Big), (22)$$

with $\rho$ the initial density matrix the system (time $t = 0$) when cooling or heating commences. (In the dual examples considered in Section VI, $\rho = |\psi^0\rangle\langle\psi^0|$ with $|\psi^0\rangle$ the initial spin or bosonic wavefunction.) If $\sigma_\epsilon$ is to vanish identically then the resulting series for Eq. (22) must vanish, order by order, for any $H(t)$. Collecting terms to the first two nontrivial orders in $H(t > 0)$,

$$\sigma_\epsilon^2(t) = \sigma_\epsilon^2(0) + \frac{1}{N^2}\langle[(\Delta H)^2, \Omega_1]\rangle$$
$$+ \frac{1}{2N^2}\Big(\langle[\Omega_1,[\Omega_1,(\Delta H)^2]] + [(\Delta H)^2, \Omega_2]\rangle$$
$$- \langle[(\Delta H), \Omega_1]\rangle^2\Big) + \mathcal{O}((H(t > 0))^3). \qquad (23)$$

Here, $\langle - \rangle$ denotes an average computed with $\rho$ and $\Delta H \equiv (H - E_0)$ where $H \equiv H(t = 0)$ and $E_0$ is the

initial energy $\langle H \rangle$. We emphasize that if, at all times $t$, the standard deviation vanishes identically for the heated/cooled system with the time dependent Hamiltonian, then the sum of all terms of a given order in $H(t > 0)$ in the expansion of Eq. (23) must vanish for a general $H(t)$. In the special case $\rho = |\phi_n\rangle\langle\phi_n|$ with $|\phi_n\rangle$ an eigenstate of $H$, the expectation values $\langle[\Delta H, \Omega_1]\rangle = [(\Delta H)^2, \Omega_2]\rangle = 0$. For this density matrix $\rho$, to order $\mathcal{O}((H(t > 0))^2)$, the standard deviation is given by the norm $\sigma_\epsilon = \left|\frac{\Delta H(i\Omega_1(t))}{N}|\phi_n\rangle\right|$ or, equivalently,

$$\sigma_\epsilon(t) = \frac{1}{N\hbar}\left|\int_0^t dt_1[\Delta H, H(t_1)]|\phi_n\rangle\right|. \qquad (24)$$

Because the total energy of the system changes with time (at an $\mathcal{O}(N)$ rate), the commutator of Eq. (24) cannot identically vanish and is, typically, of order $\mathcal{O}(N)$. Nonetheless, it is possible that when acting on the eigenstate $|\phi_n\rangle$, this commutator will yield a vector of size $o(N)$ and thus a vanishing contribution to $\sigma_\epsilon$ in the $N \to \infty$ limit.

Indeed, as is to be expected, in the special product state setting of Section V, we will obtain a vanishing $\sigma_\epsilon$. Specifically, if for all $t$, the Hamiltonian $H(t) = \sum_{i=1}^{N'} \mathcal{H}_i(t)$ is a sum of decoupled commuting local operators that, act on the same $M = N' = \mathcal{O}(N)$ disjoint subspaces (Eq. (7)), then the eigenstates $|\phi_n\rangle$ of $H(t = 0)$ will be a product of $N'$ decoupled states. Under the further constraint that, for all $t$, the operator norm $||\mathcal{H}_i(t)|| \leq Y = \mathcal{O}(1)$, one observes that $(\Delta H)\Omega_1(t)|\phi_n\rangle$ becomes the sum of $N'$ orthogonal local product state vectors, each of which is of length $\mathcal{O}(1)$. Then, from Eq. (24), to second order in $H(t > 0)$,

$$\sigma_\epsilon^{\mathsf{local}}(t) \lesssim \frac{t\sqrt{N'}}{\hbar N}Y^2. \qquad (25)$$

Hence, to this order in the expansion of Eq. (23), for such local product states $|\phi_n\rangle$, we have $\lim_{N\to\infty} \sigma_\epsilon^{\mathsf{local}}(t) = 0$.

Contrary to Eq. (25), however, for *general* non-product state density matrices $\rho$ and non-adiabatic evolution of $H(t)$ (for which the commutators appearing in the series for $\sigma_\epsilon$ tend to zero), the norm of Eq. (24) does not identically vanish as $N \to \infty$ for all functions $H(t)$ and initial density matrices $\rho$ (even if $\rho$ is a stationary under an evolution with the initial Hamiltonian $H$). That a resulting $\sigma_\epsilon = 0$ cannot appear identically is also evident from our examples of Section VI. The non-vanishing series of Eq. (23) illustrates that when the system starts from an equilibrium state with a sharp energy density $\sigma_\epsilon(0) = 0$, then notwithstanding any locality of the Hamiltonian, $\sigma_\epsilon$ may become finite (i.e., $\mathcal{O}(1)$) at later times $t$.

The Dyson type expansion analysis is not limited to the energy density $\epsilon$ (similar results hold for any other intensive quantity $q$) nor to specific continuum or lattice systems. Thus, broad distributions may generally arise in systems displaying an evolution of their intensive quantities. Of course, constrained solutions to the equation

$\sigma_\epsilon(t) = 0$, at all times $t$, may be engineered. Indeed, particular solutions associated with operators that translate the system spectrum bring to life the intuitive analogy that we made with wave packets (Section IV) as well as the special character of product states (Section V). We next study yet another situation in which the demonstration of a finite $\sigma_\epsilon > 0$ is rather trivial.

## VIII. SHORT TIME AVERAGED PROBABILITY DISTRIBUTION

The results of this Section are motivated by and will also apply to averages in classical systems. We examine a time averaged probability density matrix on $\mathcal{S}$,

$$\rho_{\tilde{\tau}}(t) \equiv \frac{1}{\tilde{\tau}}\int_t^{t+\tilde{\tau}} \rho(t')dt'. \qquad (26)$$

Here, $\rho(t') = \mathcal{U}(t)\rho\mathcal{U}^\dagger(t)$ is the (instantaneous) density matrix in the Schrodinger picture. Arguably, any real measurement of a macroscopic quantity $Q$ in large "semiclassical" systems is not instantaneous but rather requires a finite period of time $\tilde{\tau}$; thus the observed values correspond to $Tr(\rho_{\tilde{\tau}}(t)Q)$. Averaging with this probability distribution,

$$\left\langle\left(\frac{H}{N}\right)^2\right\rangle_{\tilde{\tau}} \equiv \frac{1}{N^2}Tr(\rho_{\tilde{\tau}}(t)H^2)$$
$$= \int_t^{t+\tilde{\tau}} \frac{Tr(\rho(t')H^2)}{N^2\tilde{\tau}}dt' \geq \int_t^{t+\tilde{\tau}} \frac{(Tr(\rho(t')H))^2}{N^2\tilde{\tau}}dt'$$
$$= \frac{1}{\tilde{\tau}}\int_t^{t+\tilde{\tau}} \epsilon^2(t')dt'. \qquad (27)$$

Similarly,

$$\left\langle\frac{H}{N}\right\rangle_{\tilde{\tau}} = \frac{1}{\tilde{\tau}}\int_t^{t+\tilde{\tau}} \epsilon(t')dt'. \qquad (28)$$

Hence, $\sigma_{\epsilon,\tilde{\tau}}^2 \equiv \left\langle\left(\frac{H}{N}\right)^2\right\rangle_\tau - \left\langle\frac{H}{N}\right\rangle_\tau^2$ will be finite for an energy density $\epsilon$ that varies at a finite rate in the interval $[t, t+\tilde{\tau}]$. For a short time interval in which $\frac{d\epsilon}{dt'}$ is approximately constant, Taylor expanding $\epsilon(t')$ to linear order in $(t' - (t + \frac{\tilde{\tau}}{2}))$ in the integrands of Eqs. (27, 28),

$$\sigma_{\epsilon,\tilde{\tau}} \gtrsim \frac{\tilde{\tau}}{\sqrt{12}}\left|\frac{d\epsilon}{dt}\right|. \qquad (29)$$

Putting all of the pieces together, we see, from Eq. (3), that macroscopic range $\overline{G} > 0$ will appear when all correlations evaluated with the time averaged density matrix $\rho_{\tilde{\tau}}(t)$ of Eq. (26). Albeit being trivial, this result is extremely general and applies to all density matrices and Hamiltonians whenever $\frac{d\epsilon}{dt} \neq 0$. Returning to the opening sentence of this Section, the inequalities of Eqs. (27, 29) indeed also hold for classical systems (with the trace in Eq. (27) replaced by phase space integrals or other sum over classical microstates and $\rho$ being a classical probability distribution. Although it is somewhat

obvious, it is nonetheless important to emphasize that, in the quantum arena, having an instantaneous density matrix that is a product state does not imply a time averaged density matrix that is also a product state. This is much the same as the two spin $S = 1/2$ density matrix $\frac{1}{2}(|\uparrow\uparrow\rangle\langle\uparrow\uparrow| + |\downarrow\downarrow\rangle\langle\downarrow\downarrow|)$; the latter is an average of the density matrices of two product states yet it is not, of course, the density matrix of a product state. The above result holds for both classical and quantum systems. For classical systems, one replaces the probability density matrix $\rho(t')$ in Eq. (26) by the corresponding classical probability distribution. It should be stressed that in classical ergodic systems, equilibrium (and various non-equilibrium) phase space probability distributions have their conceptual origin in long or finite time averages: an equilibrium ensemble average reproduces the long time expectation values. A variation of the energy density $\epsilon(t')$ in the time interval $[t, t + \tau]$ implies a finite $\sigma_{\epsilon,\tilde{\tau}}$. Similarly, in the next Section we will demonstrate that under certain conditions, $\sigma_\epsilon$ must be finite when $d\epsilon/dt \neq 0$. The converse, however, does not follow: a static energy density $\epsilon$ does not imply that $\sigma_{\epsilon,\tilde{\tau}}$ and $\sigma_\epsilon$ are zero. In Sections X and XII, we will further discuss what occurs once the system is no longer driven. Our result of Eq. (29) implies that there are states for which the standard deviation $\sigma_\epsilon > 0$ when the latter is evaluated for instantaneous expectation values in mixed and pure states evolving under a piecewise constant $H(t)$ (such as that of Section VI). To this end, we may equate $\rho_\tau$ to be the instantaneous density matrix $\rho^{new}(t)$ of a new mixed state or, alternatively, to be the partial trace of the density matrix of a pure state defined on an artificially constructed volume $\mathcal{I}'$ larger than the system volume $(\mathcal{S})$ on which the Hamiltonian $H$ acts ($\mathcal{I}' = \mathcal{S} \cup \mathcal{E}'$ with $\mathcal{E}'$ an artificially constructed "environment") following the "purification" procedure of [65, 66]. In the notation of [66], the dimension $\mathcal{D}$ will correspond to the number of time steps in a discretization of the integral of Eq. (26). Herein, given original pure states $\{|\psi(t')\rangle\}$ (with $t' = t + k\tau/\mathcal{D}$ with integer $1 \leq k \leq \mathcal{D}$), the scaled density matrices $\frac{|\psi(t')\rangle\langle\psi(t')|}{\tau}$ may be summed, as in Eq. (26), to provide an instantaneous density matrix $\rho^{new}(t)$. The latter density matrix may, following [66], be constructed such that its partial trace over the environment $\mathcal{E}'$ yields $\rho_{\tilde{\tau}}(t)$ (i.e., $\rho_{\tilde{\tau}}(t) = \rho^{new}(t) = Tr_{\mathcal{E}'}|\Psi(t)\rangle\langle\Psi(t)|$ with $|\Psi(t)\rangle$ a pure state in $\mathcal{I}'$). This demonstrates, once again, that the standard deviation $\sigma_\epsilon$ as evaluated with instantaneous probability density matrices or pure states can be trivially finite even for local Hamiltonians $H$.

# IX.  GENERALIZED TWO-HAMILTONIAN UNCERTAINTY RELATIONS

We next turn to a more specific demonstration that, in other settings, when evaluated with the instantaneous density matrix, the standard deviation $\sigma_\epsilon > 0$ when the energy density exhibits a finite rate of change. In

this Section, we consider non-relativistic systems $\mathcal{S}$ of arbitrary size $N$ (large or small) that satisfy certain conditions in the order of decreasing generality.

We first derive exact inequalities for closed systems and discuss, once again, how our results relate to causality. We will then derive exact bounds for open systems. In this Section, we will formalize and study procedure (2) of Section II. We will explicitly include the effects of the environment. In Sections IX A and IX B, we will respectively analyze, situations in which the ensuing system-environment hybrids constitute larger closed or open hybrid systems.

## A.  Closed system-environment hybrids

### 1.  Exact Inequalities for closed system-environment hybrids

In this subsection, we will derive inequalities when the following assumptions are satisfied:

**Assumption (1)**: When combined with their physical environment (or "heat bath") $\mathcal{E}$, these systems constitute a larger global *closed isolated* hybrid system $\mathcal{I} = \mathcal{S} \cup \mathcal{E}$ (of $\tilde{N}$ sites) in which the sites in $\mathcal{S}$ do not interact with any sites that are not in $\mathcal{I}$. The number of particles or sites in both $\mathcal{S}$ and $\mathcal{E}$ is held fixed. $\diamond$

**Assumption (2 - weak version)**: The Hamiltonian $H$ describing $\mathcal{S}$ is time independent. $\diamond$

We stress that the Hamiltonian $\tilde{H}$ describing the full hybrid system $\mathcal{I}$ including interactions between $\mathcal{S}$ and $\mathcal{E}$ is, at this stage, kept general and may depend on time.

Denoting the evolution operator (first discussed in Eq. (4)) of the full closed hybrid system $\mathcal{I}$ by

$$\tilde{\mathcal{U}}(t) = \mathcal{T}\exp(-\frac{i}{\hbar}\int_0^t \tilde{H}(t')dt'), \qquad (30)$$

the two Heisenberg picture Hamiltonians $H^H(t) = \tilde{\mathcal{U}}^\dagger(t)H\tilde{\mathcal{U}}(t)(t)$ and $\tilde{H}^H(t) = \tilde{\mathcal{U}}^\dagger(t)\tilde{H}\tilde{\mathcal{U}}(t)$ describe, respectively, the open system $\mathcal{S}$ and the larger closed hybrid system $\mathcal{I}$ at time $t$. The energy of the system $\mathcal{S}$ is $E(t) = Tr_{\mathcal{I}}(\tilde{\rho}H^H(t))$ where $\tilde{\rho}$ is the initial density matrix of $\mathcal{I}$. By the uncertainty relations [80, 81],

$$\sigma_{\epsilon(t)}\sigma_{\tilde{H}^H(t)} \geq \frac{1}{2}\Big|Tr_{\mathcal{I}}(\tilde{\rho}[\frac{H^H(t)}{N}, \tilde{H}^H(t)])\Big|. \qquad (31)$$

Here, $\sigma_{\epsilon(t)}$ and $\sigma_{\tilde{H}^H(t)}$ denote, respectively, the uncertainties associated with $H^H(t)/N$ and $\tilde{H}^H(t)$ (when these uncertainties are computed with the probability density matrix $\tilde{\rho}$). Combined with the Heisenberg equations of motion for the time independent $H$ (Assumption

(2- weak version)), we obtain an extension of the time-energy uncertainty relations for this two Hamiltonian realization,

$$\sigma_{\epsilon(t)}\sigma_{\tilde{H}^H(t)} \geq \frac{\hbar}{2N}\left|\frac{dE}{dt}\right|. \tag{32}$$

Eq. (3) then implies a lower bound on the average macroscopic range correlators in the subsystem $\mathcal{S}$,

$$\overline{G}_{\mathcal{S}} \geq \frac{\hbar^2}{4\sigma_{\tilde{H}(t)}^2}\left(\frac{d\epsilon}{dt}\right)^2. \tag{33}$$

The derivative in Eq. (33) scales as $\mathcal{O}(N^2)$ if the energy $E(t)$ of $\mathcal{S}$ changes at a rate proportional to the size of $\mathcal{S}$ (i.e., if the energy density changes at a finite rate). Eqs. (31,32,33) will remain valid if Assumption (1) is relaxed, i.e., if $\mathcal{I}$ is an open system with a Hamiltonian $\tilde{H}$ that, itself, is in contact with a yet larger system. We next examine what occurs if the local energy density correlators $G_{ij}$ decay with a correlation length $\xi$, i.e., with

$$G_{ij} \sim A\frac{e^{-|i-j|/\xi_H}}{|i-j|^p}, \tag{34}$$

with $A$ a finite constant. Transforming to hyperspherical coordinates, we see that on a $d$ dimensional hypercubic $L \times L \times ... \times L$ lattice with $L \gg \xi_H$, the average correlator of Eq. (3) will, up to factors of order unity, be given by $\overline{G}_{\mathcal{S}} \sim 2A\frac{\pi^{d/2}\Gamma(d-p)}{\Gamma(\frac{d}{2})}(\frac{\xi_H}{L})^d\xi_H^{-p}$. Combined with Eq. (33), this implies a lower bound on the correlation length

$$\xi_H \gtrsim L^{\frac{d}{d-p}}\left[\frac{\hbar^2\Gamma(\frac{d}{2})}{8A\pi^{d/2}\Gamma(d-p)\sigma_{\tilde{H}}^2(t)}\left(\frac{d\epsilon}{dt}\right)^2\right]^{1/(d-p)}, \tag{35}$$

with $\epsilon(t) = E(t)/N$ the energy density of $\mathcal{S}$. Note that the lower bound of Eq. (35) on the correlation length is monotonic in the temporal variation of the energy density $\epsilon(t)$. That is, the larger the rate of change $|\frac{d\epsilon}{dt}|$ of the energy density, the larger the lower bound on the putative finite correlation length $\xi_H$. In particular, for finite $d\epsilon/dt$ and $\sigma_{\tilde{H}}$, such a lower bound will diverge as $L \to \infty$ (indicating that an assumption of small $\xi_H$ cannot be made self-consistently). Moreover, regardless of $p$, if (in any dimension) $\sigma_{\tilde{H}} < \mathcal{O}(\sqrt{N})$ then Eq. (35) illustrates that $\xi_H$ cannot be finite in the $L \to \infty$ limit whenever $d\epsilon/dt$ is finite. Thus, the reader can see how divergent correlation lengths are mandated whenever $\mathcal{I}$ exhibits fluctuations that are smaller than those of typical open systems (i.e., when $\sigma_{\tilde{H}} = o(\sqrt{N})$). The bound of Eq. (35) assumes Eq. (34) and is only suggestive. In what follows, we will examine conditions that will enforce a finite $\sigma_{\tilde{H}}$ and thus divergent correlations when $d\epsilon/dt \neq 0$. Towards that end, we impose a more restrictive condition:

**Assumption (2 - strong version)** : The *fundamental* interactions appearing in the global Hamiltonian $\tilde{H}$

describing $\mathcal{I}$ are time independent. ⋄

This assumption (which, for brevity, we will henceforth simply refer to as **Assumption (2)**) implies Assumption (2 - weak version). This is so since the terms in $\tilde{H}$ include, as a subset, the interactions appearing in the Hamiltonian $H$ describing $\mathcal{S}$. When Assumption (2) holds, the global Heisenberg and Schrodinger picture Hamiltonians coincide, $\tilde{H}^H(t) = \tilde{H}$. If a time independent Hamiltonian $\tilde{H}$ governs the dynamics of the closed hybrid system $\mathcal{I}$, then the energy will not vary with time. Classically, there is no meaningful finite standard deviation $\sigma_{\tilde{H}}$: the energy of the closed system is conserved. By contrast, no quantum dynamics are possible unless $\sigma_{\tilde{H}} \neq 0$. That is, any eigenstate of $\tilde{H}$ (for which $\sigma_{\tilde{H}} = 0$) is trivially stationary under an evolution with $\tilde{H}$. For a general initial state $|\tilde{\psi}^0\rangle$ of the closed hybrid system $\mathcal{I}$, the probability density,

$$\tilde{\rho}(t) = \sum_{\tilde{n}\tilde{m}} e^{-i\frac{(\tilde{E}_{\tilde{n}}-\tilde{E}_{\tilde{m}})t}{\hbar}}\langle\tilde{\phi}_{\tilde{n}}|\tilde{\psi}^0\rangle\langle\tilde{\psi}^0|\tilde{\phi}_{\tilde{m}}\rangle|\tilde{\phi}_{\tilde{n}}\rangle\langle\tilde{\phi}_{\tilde{m}}|, \tag{36}$$

will typically vary on a time scale of order $\tau \equiv \frac{\hbar}{\sigma_{\tilde{H}}}$. In Eq. (36), $\{|\tilde{\phi}_{\tilde{n}}\rangle\}$ are the eigenstates of $\tilde{H}$. The off-diagonal spread of the density matrix (in the eigenbasis of $\tilde{H}$) determines the oscillation frequencies that it displays. For pure states $|\tilde{\psi}^0\rangle$ in the closed hybrid system $\mathcal{I}$, a large $\sigma_{\tilde{H}}$ implies large temporal fluctuations [82]. If, as in many closed energy conserving systems with a well-defined semi-classical limit, the representative frequencies governing the global dynamics (and probability density) do not scale with $N$, i.e., if $\mathcal{O}(\tau) = \mathcal{O}(1)$ [83] then $\sigma_{\tilde{H}}$ will, typically, also not vary with $N$. Inserting $\sigma_{\tilde{H}} = \frac{\hbar}{\tau}$ in Eq. (32),

$$\sigma_{\epsilon(t)} \geq \frac{\tau}{2}\left|\frac{d\epsilon}{dt}\right|. \tag{37}$$

This result is natural for a probability distribution that varies over time scales $\gtrsim \tau$. Along related lines, a time average of the form of Eq. (26) applied to the density matrix $\tilde{\rho}$ on $\mathcal{I}$ (i.e., $\tilde{\rho}_{\tilde{\tau}}(t) \equiv \frac{1}{\tilde{\tau}}\int_t^{t+\tilde{\tau}}\tilde{\rho}(t')dt'$) will remove frequencies higher than a cutoff that scales as $\hbar/\tilde{\tau}$. That is, if $\tau < \tilde{\tau}$, then $\tilde{\rho}_{\tilde{\tau}}(t)$ will not exhibit the higher frequency oscillations present in $\tilde{\rho}(t)$. The removal of these high frequencies (associated with short "virtual events") will render the system more "semi-classical"; in a path integral representation, in the sum of the exponentiated classical action over all possible paths, fluctuations of phases generated by relative energy differences larger than $\mathcal{O}(\hbar/\tilde{\tau})$ will, for an evolution over a time of length $\tilde{\tau}$, lead to oscillatory phases that will cancel. The larger the waiting or averaging time $\tilde{\tau}$ is, the more narrow the range of eigenstates that are relevant to the system evolution will be (i.e., only those with energies in a small window about the average system energy may be considered) on time scales $\geq \tilde{\tau}$.

The above intuition can be made more accurate to further bolster the considerations of Section VIII. The

bound of Eq. (31) is an algebraic identity that may be extended to arbitrary probability density matrices. In particular, in Eq. (31), we may replace $\tilde{\rho} \to \tilde{\rho}_{\tilde{\tau}}$ for general averaging times $\tilde{\tau}$ (Eq. (26)). This implies the inequality

$$\sigma_{\epsilon(t)}^{\tilde{\tau}} \sigma_{\tilde{H}(t)}^{\tilde{\tau}} \geq \frac{\hbar}{2N}\left|\frac{dE_{\tilde{\tau}}}{dt}\right|, \qquad (38)$$

where $E_{\tilde{\tau}}(t) \equiv Tr_{\mathcal{I}}(\tilde{\rho}_{\tilde{\tau}} H^H(t))$. In Eq. (38), $\sigma_{\epsilon(t)}^{\tilde{\tau}}$ and $\sigma_{\tilde{H}(t)}^{\tilde{\tau}}$ denote, respectively, the standard deviations of $(H/N)$ and $\tilde{H}$ as computed with the time averaged probability distribution $\tilde{\rho}_{\tilde{\tau}}$. Thus, we can qualitatively relate the uncertainty relations to the trivial general bounds of Eqs. (27, 29). That is, for any finite (system size independent) averaging time $\tilde{\tau}$, the density matrix $\tilde{\rho}_{\tilde{\tau}}(t)$ will display $\sigma_{\tilde{H}} \leq \hbar/\tilde{\tau}$. Eq. (38) will (in agreement with Eqs. (27, 29)) then imply a finite $\sigma_{\epsilon(t)}^{\tilde{\tau}}$ whenever $\frac{dE_{\tilde{\tau}}}{dt}$ is extensive. As emphasized earlier, of physical relevance are finite time ($\tilde{\tau} > 0$) window measurements.

While bounded system size independent frequencies are natural in quasi-classical and "typical" closed (energy conserving) quantum systems, that is certainly not the case for all constructible model states [84]. With this in mind, we consider the consequences of any one of two additional conditions (labelled Assumption (3) and Assumption (3') in the below). Either of these conditions will lead to a system size independent standard deviation for the energy density (when the latter is evaluated with the instantaneous density matrix $\tilde{\rho}$).

**Assumption (3)**: The closed hybrid system $\mathcal{I}$ equilibrates at long times. Stated more precisely (and automatically accounting for Poincare recurrence type events), the asymptotic long time average of the probability density $\rho_{\mathcal{I}}$ in the larger *closed* hybrid system $\mathcal{I}$ veers towards the *microcanonical* (mc) density matrix applicable for closed energy conserving systems in equilibrium [85]. That is,

$$\tilde{\rho}_{\mathrm{mc};\mathcal{I}} = \lim_{\tilde{\mathcal{T}} \to \infty} \frac{1}{\tilde{\mathcal{T}}} \int_0^{\tilde{\mathcal{T}}} \tilde{\rho}_{\mathcal{I}}(t')dt', \qquad (39)$$

with $\tilde{\rho}_{\mathrm{mc};\mathcal{I}}$ the microcanonical ensemble density matrix for the closed hybrid system $\mathcal{I}$. ⋄

In systems obeying Eq. (39), the uncertainty in the energy of $\mathcal{I}$ at asymptotically long times (i.e., as computed with $\tilde{\rho}_{\mathrm{mc};\mathcal{I}}$) will be system size independent,

$$\sigma_{\tilde{H}} = \mathcal{O}(1). \qquad (40)$$

Eq. (40) constitutes the defining textbook property of the microcanonical ensemble [85]. Since the closed system-environment hybrid $\mathcal{I}$ is governed by the time independent Hamiltonian $\tilde{H}$, the standard deviation $\sigma_{\tilde{H}}$ is time independent and Eq. (40) trivially holds at all times $t$ when the variance $\sigma_{\tilde{H}}$ is computed with the density matrix $\tilde{\rho}(t)$. Assumption (3) and the preceding discussion may seem abstract. The semiclassical intuition

underlying the somewhat axiomatic standard definition of the microcanonical ensemble is rather trivial. We repeat anew some elements below.

For a classical ergodic hybrid system (e.g., that assumed for $\mathcal{I}$ governed by the time independent $\tilde{H}$), the probability density is that associated with the long time average. For a *closed* conservative system, the total energy is conserved and the probability density defined in this way exhibits zero variance of the total energy. In the quantum arena, if the closed ergodic system exhibits non-trivial dynamics then the standard deviation of its Hamiltonian cannot vanish (since the eigenstates of $\tilde{H}$ are trivially stationary). Thus, the common assumption underlying the microcanonical ensemble is that the standard deviation of $\tilde{H}$ is finite (in order to allow for non-vanishing frequencies) yet, for *a closed system* does not diverge as the size increases. This intuition rationalizes the standard use of Eq. (40) defining the microcanonical ensemble.

In the spirit of the above maxim, we next introduce an alternate assumption that does not rely on the closed hybrid system $\mathcal{I}$ being ergodic (nor the use of the microcanonical ensemble):

**Assumption (3')**: A finite time step discretization ($t = t_{\mathsf{k}} = \mathsf{k}\Delta t$ with integer $\mathsf{k}$ and $\Delta t$ a sufficiently small system size independent time step) may simulate the evolution of $\mathcal{I}$ [86]. Here, as before, the (pure) state of the *closed* hybrid system $\mathcal{I}$ may be described by a wavefunction. The uniform discretization of $t$ implies that any function $f(t)$ (including the associated density matrix $\tilde{\rho}(t)$ of Eq. (36)) may be expressed as a Fourier sum $f(t) = \sum_{p'} \hat{f}(\omega_{p'})e^{-i\omega_{p'}t}$ with $\omega_{p'}$ lying in the "first Brillouin zone" ($|\omega_{p'}| \leq \pi/\Delta t$). Thus, the uncertainty in the energy of the closed hybrid system $\mathcal{I}$ satisfies $\sigma_{\tilde{H}} \leq \pi\hbar/\Delta t$- a realization of Eq. (40); gauge invariant [87] expectation values of finite time gradients in $\mathcal{I}$ (including the standard deviation of the discrete time gradient approximation of the Hamiltonian $\tilde{H} = i\hbar\frac{\partial}{\partial t}$) are bounded from above by $\mathcal{O}(1/\Delta t)$. ⋄

Assumptions (1-3) (as well as Assumptions (1,2,3')) [88] imply that when the energy density varies at a finite rate ($dE/dt = \mathcal{O}(N)$) then, from Eqs. (32,37,40), the standard deviation of the energy density of $\mathcal{S}$,

$$\boxed{\sigma_{\epsilon(t)} = \mathcal{O}(1).} \qquad (41)$$

Thus, we discern from Eqs. (3, 33) that long range correlations must appear during the cooling or heating period at which the energy density of the system ($\mathcal{S}$) is varied at a finite rate. Analogs of Eq. (41) are also valid for any other intensive quantity $q$ (different from the energy density $\epsilon$) whenever $\frac{dq}{dt} \neq 0$. Analogs of Eq. (41) are also valid for any other intensive quantity $q$ (different from the energy density $\epsilon$) whenever $\frac{dq}{dt} \neq 0$. When the environment $\mathcal{E}$ is included for (as we do now), the evolution

of the system itself (Figure 2) is non unitary; this non unitary evolution lies in strong contrast to the earlier examples of Section VI in which the system evolved unitarily. One may, nonetheless, still make some non-rigorous pedagogical contact with the spin models of Section VI [89]. Assumptions (1-3) are often employed in standard textbook derivations of the canonical ensemble for open systems $\mathcal{S}$ by applying the microcanonical ensemble averages for the larger equilibrated closed systems $\mathcal{I}$ that include the relevant environments $\mathcal{E}$ that are in contact (or "entangled") with $\mathcal{S}$. If, as evinced by measurements in prototypical states in the composite hybrid system $\mathcal{I}$ at asymptotically long times, ergodicity and equilibrium set in, then the microcanonical ensemble may be invoked. We next turn to the scales of the righthand sides of Eqs. (31,32,33) and their consequence for systems that are cooled/heat at finite rate. By Heisenberg's equation, $\frac{dH^H}{dt} = \frac{i}{\hbar}[\tilde{H}, H^H]$. Therefore, in order to obtain a finite $d\epsilon/dt$ (or an extensive rate $dE/dt$), the total Hamiltonian $\tilde{H}$ of the large hybrid system $\mathcal{I}$ must have a commutator with the Hamiltonian $H$ of $\mathcal{S}$ that is of order $N$, i.e., $Tr_{\mathcal{I}}(\tilde{\rho}[\tilde{H}, H^H]) = \mathcal{O}(N)$. Hence, to achieve a finite global rate of cooling/heating, $\tilde{H}$ must couple to an *extensive number of sites in the volume of $\mathcal{S}$*- it is not possible to obtain an extensive cooling/heating rate by a bounded strength coupling that extends over an infinitesimal fraction of the system size (see also the discussion at the end of Section IV and that appearing after Eq. (10) in Section VI A)). Effectively, a finite fraction of the sites lying in the volume of $\mathcal{S}$ must couple to $\tilde{H}$ whenever $\frac{d\epsilon}{dt} = \mathcal{O}(1)$. The initial state of the system $\mathcal{S}$ prior to its cooling/heating (or variation in its other parameters) may have a well defined energy density $\epsilon$ and other state variables yet nonetheless still be far from a typical equilibrium state. One may introduce various probes, clocks, etc., that start the cooling/heating process in a particular way; the initial state need not be in equilibrium but may rather be specially crafted.

If Assumptions (1-3) are met then at asymptotically long times, memory of the initial state will be lost and all observables may be computed via the microcanonical ensemble with its few thermodynamic state variables. In particular, the defining feature of the microcanonical probability distribution of closed equilibrated systems holds, Eq. (40). For completeness, we conclude by noting that the Dyson type expansion of Section VII may also be reproduced in the setting of the current subsection with a time independent $\tilde{H}$ (for which the evolution operator is $e^{-i\tilde{H}t/\hbar}$ and the global density matrix is given by $\tilde{\rho}$).

### 2. Remarks on causality

In this subsection, the effect of the environment $\mathcal{E}$ driving the system was explicitly included and, as in basic theories, the form of the terms in the system-environment hybrid (i.e., those in $\tilde{H}$) was time independent. While the fundamental interactions in $\tilde{H}$ are time independent, tracing over the environment (Figure 2) may lead to complex dynamical maps. We now revisit, yet again, the constraints implied by causality. As noted in Section IV, in models with local interactions, Lieb-Robinson inequalities [27] generally provide upper bounds on commutators such as those appearing in Eq. (31). These relations lead to bounds on correlations [8, 9]. However, as we explained above, in driven systems for which the energy density is made to vary at a finite rate, commutators such as those of Eq. (31) must be extensive; such commutators may only appear at sufficiently long times (we refer the reader, once againm to Appendix B for an explicit proof of this assertion). In diverse physical situations (i.e., when cooling/heating leads to a finite rate of change of the system energy density or measured temperature), photons and/or other particles/quasiparticles emitted/absorbed by an extensive volume of the surrounding heat bath effectively couple to the system bulk (see Appendix A). In the spin model of Section VI A (in which the system evolution was unitary), the time independent (for all times $t > 0$) transverse field ($B_y$) Hamiltonian of Eq. (10) played the role of $\tilde{H}$ acting on all $N$ sites (so as to have $[\tilde{H}, H^H] = \mathcal{O}(N)$).

### B. Open system-environment hybrids

As we noted above, for a closed system described by a wavefunction, a large $\sigma_{\tilde{H}}$ implies rapid temporal fluctuations. By contrast, the density matrix describing an open system can be time independent yet exhibit large $\sigma_{\tilde{H}}$ [82]. "Canonical" open systems $\mathcal{I}$ feature a large (by comparison to the energy uncertainties of the closed systems that we discussed earlier) $\sigma_{\tilde{H}} \sim \tilde{N}^{1/2}$ scaling. This larger value of $\sigma_{\tilde{H}}$ renders the corollaries of Eq. (32) weaker for open systems. Nonetheless, as we will next demonstrate by a simple "proof by contradiction" argument, if we consider an initial *open* thermal system composite $\mathcal{I}$ at an assumed temperature $T$ (instead of Assumption (1) for the closed systems of Section IX A), then there exists a limiting cooling/heating rate beyond which equilibrium is impossible. The bound that we will present encompasses the physical situation of a general uniform medium that is heated or cooled via contacts with an external environment. Our result pertains to what transpires if the subsystem $\mathcal{S}$ and the larger open hybrid system $\mathcal{I}$ containing it are in equilibrium with one another at a temperature $T$ (see Figure 6). Specifically, we will invoke the following assumptions for open ($^o$) systems:

**Assumption ($1^o$)**: When combined with their environment (or "heat bath") $\mathcal{E}$, these systems constitute a larger *open* hybrid system $\mathcal{I} = \mathcal{S} \cup \mathcal{E}$ in which the sites in $\mathcal{S}$ *do not interact with any sites that are not in $\mathcal{I}$*. $\diamond$

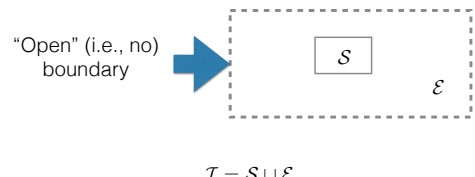

"Open" (i.e., no) boundary

$S$

$\mathcal{E}$

$\mathcal{I} = S \cup \mathcal{E}$

FIG. 6. An open system-environment hybrid $\mathcal{I}$. The degrees of freedom in $S$ may only interact with others in $S$ and/or the environment $\mathcal{E}$. Unlike the analysis of IX A, however, the constituents of $\mathcal{E}$ may now interact also with others not in $\mathcal{I}$.

**Assumption ($2^o$):** The open hybrid system $\mathcal{I}$ is in thermal equilibrium with its environment at a fixed temperature $T$. In particular, the fluctuations (as computed with initial probability density matrix $\tilde{\rho}$) of extensive quantities are those of an equilibrated system at a temperature $T$. ⋄

**Assumption ($3^o$):** The subsystem $S \subset \mathcal{I}$ is in thermal equilibrium with $\mathcal{I}$. ⋄

This last assumption might be regarded as a consequence of Assumption ($2^o$) for the equilibrated hybrid system $\mathcal{I}$ that includes $S$. Nonetheless, we wish to make Assumption ($3^o$) explicit.

The open hybrid system $\mathcal{I}$ (including $S$) may be taken to lie deep in a uniform medium so that it is far from any external contacts that change its temperature. The Heisenberg picture Hamiltonian $\tilde{H}^H$ evolves with an operator different from Eq. (30)- one that involves also the sites exterior to $\mathcal{I}$. The latter coupling allows for a non-trivial time dependence. Equivalently, the Schrodinger picture probability density matrix $\tilde{\rho}(t)$ is generally a function of time [90]. Macroscopic expectation values computed with $\tilde{\rho}(t)$ are those of equilibrated thermal systems yet measurable dynamics also appear (as in, e.g., an equilibrated gas with mobile molecules having correlations set by the diffusion equation). For a static $\tilde{\rho}(t)$, all expectation values will be trivially stationary. Since, by Assumption ($2^o$), the full hybrid system $\mathcal{I} = S \cup \mathcal{E}$ is in equilibrium, the system $S$ must be in equilibrium with its environment $\mathcal{E}$. From the zeroth law of thermodynamics, it then follows that $S$ is also described by a (canonical) probability density matrix at the same inverse temperature $\beta$.

Because the sites in $S$ only interact with those in $\mathcal{I}$ (Assumption ($1^o$)), Eqs. (31,32,33) (as well as the bound of Eq. (35) for correlators of the form of Eq. (34)) remain valid. In what follows, following Assumption ($2^o$), we will set, in Eq. (32), the equilibrium values of standard deviations of the respective Hamiltonians in the appropriate (canonical) ensembles describing the open systems $\mathcal{I}$ and $S$. That is, $\sigma_{\tilde{H}} = \sqrt{k_B T^2 C_{v,\mathcal{I}}(T)}$ (with $k_B$ the Boltzmann constant and $C_{v,\mathcal{I}}(T)$ the constant volume heat capacity of the large system composite $\mathcal{I}$) to be the standard deviation of the large open hybrid system $\mathcal{I}$, and

equate $\sigma_H = \sqrt{k_B T^2 C_{v,S}(T)}$ (where $C_{v,S}(T)$ is the heat capacity of the small system at temperature $T$) to be the standard deviation of the smaller subsystem $S$. We may repeat, *mutatis mutandis*, the steps that led to Eq. (41) when $\mathcal{I}$ was a closed system. Doing so and employing Eq. (32), we discover that if the cooling/heating rate exceeds a threshold value for an equilibrated open hybrid system $\mathcal{I}$ (and any subsystem $S \subset \mathcal{I}$ that is in equilibrium with it (Assumption ($3^o$)),

$$\boxed{\left|\frac{dE}{dt}\right| > \frac{2}{\hbar} k_B T^2 \sqrt{C_{v,\mathcal{I}}(T) C_{v,S}(T)},} \tag{42}$$

then a simple contradiction will be obtained. That is, an assumption of having a sharp equilibrium energy density state variable (by coupling $\mathcal{I}$ to a larger external bath at a well defined temperature) [98] becomes inconsistent once Eq. (42) is satisfied. At sufficiently fast cooling or heating rates (given by Eq. (42)), the inequality of Eq. (32) will be violated when we substitute the equilibrium open system values of $\sigma_{H/N}$ and $\sigma_{\tilde{H}}$.

Using Eq. (42), it is illuminating to estimate the rate of the temperature variation beyond which equilibration of an open system is rigorously impossible. Towards that end, we assume that $\mathcal{I}$ and $S$ are of comparable size ($\mathcal{O}(N)$) and that the heat capacity of both is, up to factors of order unity, given by $dN k_B$ and that the energy density is the order of ($d k_B T$). Hence, if the energy variations fulfill a "Planckian rate" inequality,

$$\boxed{\frac{\left|\frac{d\epsilon}{dt}\right|}{\epsilon} \gtrsim \mathcal{O}\left(\frac{2k_B T}{\hbar}\right),} \tag{43}$$

then, in any dimension $d$, it might be impossible to satisfy all of our assumptions in unison. Interestingly, earlier work established that the thermalization rates for typical random states are given by $\frac{k_B T}{\hbar}$ [91]. The rigorous inequality of Eq. (42) and its common realization of Eq. (43) augment these relations to rigorously demonstrate that in typical situations (when all energy densities and heat capacities are set by the Botlzmann constant, the number of particles, and the energy), whenever *the heating/cooling rate is larger than $\mathcal{O}(2k_B T/\hbar)$ then no thermalization of the open system is possible.* It is important to stress that the variations in the energy need not arise only as a result of an external drive.

Eq. (42) also holds true for any system in equilibrated open systems for which the variations in the energy are *thermally self-generated fluctuations typical to the equilibrium state.* Our result is similar to a bound argued for by Maldacena, Shenker, and Stanford as a bound on Lyapunov exponents ($\lambda_L \leq 2\pi k_B T/\hbar$) in an open thermal system [20]. At room temperature, $2k_B T/\hbar \sim 10^{14}$ Hz. Thus, at low temperatures, pulsed picosecond laser cooling/heating may, in principle, achieve these rates beyond which, as we just demonstrated, quantum uncertainty relations forbid thermalization (even for open systems). Our inequality of Eq. (42) is rigorous. By contrast, Eq. (43) only arises as a possible order of magnitude estimate.

Our two results of Eqs. (32, 42) for, respectively, the closed and open composites $\mathcal{I}$ apply for any rate of the energy change $dE/dt$. These include situations in which $dE/dt$ scales as the surface area of the system ($\mathcal{O}(N^{(d-1)/d})$) for which an extension of Eq. (3) will, in turn, imply that $\overline{G} \geq \mathcal{O}(N^{-2/d})$. The central results of Eqs. (41, 42) hold for any function $f(q)$ of an intensive quantity $q$ that is varied at a finite rate. In particular, setting $f(q) = q^n$, we find that the uncertainties in all moments of $q$ are, typically, finite if the rate $dq/dt$ is finite. With a formal proof at our disposal, we now briefly reflect back on the arguments of Section IV in which we explained why a varying quantity energy density (or any other intensive quantity $q$) with a finite rate of change $d\epsilon/dt$ (or general $dq/dt$) naturally suggests an uncertainty. The arguments of Section IV provide an intuitive basis for the time-energy uncertainty type relations that we derived and employed in this section for our two Hamiltonian system and, more generally for other intensive quantities. We next discuss inequalities that may also be derived when Assumption ($3^o$) is *not invoked*. Replacing the energy density $\epsilon$ in Eqs. (31, 32, 33) by a general quantity $Q$ having its support on a region *of arbitrary size* $N$, we discover that fluctuations in the equilibrium system must always satisfy

$$\tau_Q \equiv \frac{\sigma_Q}{|\langle \frac{dQ}{dt} \rangle|} \geq \frac{\hbar}{2k_B T}\sqrt{\frac{k_B}{C_{v,\mathcal{I}}}}. \quad (44)$$

Similar to Eq. (43), we find that if (i) $\mathcal{I}$ is of comparable size to $\mathcal{S}$ (having $\mathcal{O}(N)$ sites) and (ii) if $C_{v,\mathcal{I}} \lesssim dNk_B$, then $\tau_Q$ cannot be shorter than $\mathcal{O}(\frac{\hbar}{2k_B T\sqrt{dN}})$. Barring critical points/transition regions, in most substances, heat capacities are typically bounded by their high temperature value of $\mathcal{O}(dNk_B)$ making this order of magnitude inequality more stringent than might be suspected otherwise. As remarked above, in Eq. (44), $N$ may be of arbitrary size. (Indeed, what matters is that in the uncertainty relations we may still approximate the equilibrium energy fluctuations in the larger hybrid system $\mathcal{I}$ by the thermodynamic result $\sigma_{\tilde{H}} = \sqrt{k_B T^2 C_{v,\mathcal{I}}(T)}$ and that $\mathcal{S}$ only interacts with sites in $\mathcal{I}$.) In particular, in the limit of $N = 1$, the quantity $Q$ may be set to or emulate a variation in any single phase space variable (thus suggesting that the reciprocal of the righthand side of Eq.

(44) constitutes a *Lyapunov exponent bound*). Note that the upper bound of Eq. (44) becomes more stringent as the size of the open combined environment-system $\mathcal{I}$ decreases (scaling with $\tilde{N}^{-1/2}$). Eq. (44) also provides a lower bound on the average of the long distance correlators,

$$\overline{G}_Q \equiv \sigma_q^2 = \frac{1}{N^2}\sum_{i,j}\left(\langle \mathcal{Q}_i \mathcal{Q}_j \rangle - \langle \mathcal{Q}_i \rangle \langle \mathcal{Q}_j \rangle\right)$$
$$\geq \frac{\hbar^2}{4k_B T^2 C_{v,\mathcal{I}}}\left|\frac{dq}{dt}\right|^2. \quad (45)$$

By the equilibrium fluctuation-response theorem, this inequality implies a lower bound on the uniform susceptibility $\chi_Q$ associated with a general order parameter or field $Q$ for an equilibrated open thermal system in which $Q$ fluctuates at a rate $(dQ/dt)$,

$$\chi_Q \geq \frac{\hbar^2}{4k_B^2 T^3 C_{v,\mathcal{I}}}\left|\frac{dQ}{dt}\right|^2. \quad (46)$$

We conclude this Section by connecting our results concerning uncertainties in intensive quantities to conventional (non-weak [92–95]) quantum measurements. Qualitatively, interactions with the environment might be expected to mimic rapid repeated measurements that collapse the wavefunction and not allow Schrodinger type mixing states of significantly different energies to exist. Such a colloquial "paradox" is somewhat ill formed as we now explain. Continuous measurements by an environment will indeed not enable large uncertainties to appear. However, the putative existence of continuous collapses will also not allow for any change in the energy density or other intensive quantities. This situation is reminiscent to the well-known "Quantum Zeno Effect" [96] and its popularized idioms such as that of "a watched pot may never boil", e.g. [97]. Progressively weaker continuous measurements [92–95] may allow for a more rapid evolution of various quantities hand in hand with larger uncertainties. We will discuss adiabatic process, quantum measurements, and thermalization in Section XIV.

## X. DEVIATION FROM EQUILIBRIUM AVERAGES

In the earlier Sections, we demonstrated that forcefully varying the set of intensive (typical state variable) parameters $\{q'\}$ characterizing the eigenstates of $H$ (such as the energy and particle number densities) at a finite rate generally leads to a widening of the distributions $P(\{q'\})$ of these quantities. This was investigated for systems both in the presence and absence of an explicitly included external environment with similar conclusions. Indeed, the causal constraints on the effective interactions associated with the environment was the greatest physical distinction of interest. In this Section, we wish to underscore that such a widening of the distributions $P$ allows for a natural departure from equilibrium behaviors. That

is, even if the expectation values of general observables in individual eigenstates coincide with equilibrium averages [45–53] and $H$ has no special many body localized eigenstates [54–62], once a broad distribution $P(\{q'\})$ is present, all averages differ from those in true equilibrium ensembles. This will occur since the broad probability distribution $P(\{q'\})$ describing the driven system is different from the corresponding probability distribution in equilibrium systems (where all intensive quantities have vanishingly small fluctuations); thus the broad distribution $P(\{q'\})$ will give rise to expectation values of typical observables that are different from those found in equilibrium. We write the equilibrium averages of quantities $\mathcal{O}_c$ that commute with the Hamiltonian ($[\mathcal{O}_c, H] = 0$) [99] in a general equilibrium ensemble $\mathcal{W}$ for large systems of arbitrary finite size,

$$\langle \mathcal{O}_c \rangle_{eq;\{q\};\mathcal{W}} = \int dq' P_{eq;\{q\}}(\{q'\}; \mathcal{W}) \mathcal{O}_c(\{q'\}; \mathcal{W}) \quad (47)$$

Here, the integration is performed over the full set of intensive variables $\{q'\}$ and the function $P_{eq;\{q\}}(\{q'\}; \mathcal{W})$ denotes the probability distribution in an equilibrium ensemble $\mathcal{W}$ for which the average of the various quantities $q = \int dq'(q' P_{eq;\{q\}}(\{q'\}; \mathcal{W}))$. Lastly, $\mathcal{O}_c(\{q'\}; \mathcal{W}) \equiv \langle \phi(\{q'\}; \mathcal{W})|\mathcal{O}_c|\phi(\{q'\}; \mathcal{W})\rangle$. Augmenting the set of intensive quantities $\{q'\}$ defining any of the standard equilibrium ensemble probability distributions, the index $\mathcal{W}$ may specify any additional quantum numbers. These quantum numbers may be associated with symmetries in which case $\mathcal{W}$ can label the orthogonal degenerate eigenstates $\{|\phi(\{q'\}; \mathcal{W})\rangle\}$) of fixed energy or particle number or other global observables giving rise to the intensive quantities $q$. For instance, in Ising spin systems, the probability distribution $P_{eq;\{q'\}}(\{q'\}'; \mathcal{W})$ may be finite only for states with a positive magnetization $\frac{1}{N}\sum_{i=1}^{N}\langle S_i^z \rangle$ as it is in these systems at temperatures below the ordering temperatures once time reversal symmetry is spontaneously broken. An essential feature of all systems in equilibrium is that they exhibit well defined thermodynamic state variables $\{q'\}$. For instance, as we alluded to in earlier Sections, the energy density exhibits $\mathcal{O}(N^{-1/2})$ fluctuations in the open systems described by the canonical ensemble while it displays $\mathcal{O}(N^{-1})$ fluctuations in closed systems described by the microcanonical ensemble. In all equilibrium ensembles, the width $\sigma_q$ of any intensive quantity $q$ vanishes as $N \to \infty$. This sharp delta-function like characteristic of the probability distribution $P_{eq;\{q'\}}(\{q'\}'; \mathcal{W})$ is diametrically opposite of $P(\{q'\})$ for which $\sigma_q$ is finite. Consequently, the expectation value in the driven system $\langle \mathcal{O}_c \rangle_{driven}$ during the period in which $\{q'\}$ are made to vary with time (that will be given by Eq. (47) with the replacement of the equilibrium probability distribution $P_{eq}$ by its non-equilibrium counterpart with $P(\{q'\})$) will generally differ from the equilibrium average $\langle \mathcal{O}_c \rangle_{eq;\{q\};\mathcal{W}}$.

We now relate the equilibrium and non-equilibrium expectation values. Because the equilibrium distribution $P_{eq;\{q'\}}(\{q'\}'; \mathcal{W})$ is, for large systems, essentially a delta-

function in $\{q'\}$ (and all additional numbers $\mathcal{W}$), we may explicitly write the expectation values in the driven system as

$$\langle \mathcal{O}_c \rangle_{driven} = \int dq' P(q'; \mathcal{W}) \langle \mathcal{O}_c \rangle_{eq;\{q'\};\mathcal{W}}. \quad (48)$$

That is, *the expectation values of the observables $\mathcal{O}_c$ in the driven system may be expressed as weighted sums of the equilibrium averages* $\langle \mathcal{O}_c \rangle_{eq;\{q'\};\mathcal{W}}$ with the weights given by the finite width $\sigma_q$ distribution $P(q'; \mathcal{W})$ that we focused on in the earlier Sections [100]. The equilibrium expectation values $\langle \mathcal{O}_c \rangle_{eq;\{q'\};\mathcal{W}}$ of Eq. (47) are experimentally known in many cases. Thus, to predict the expectation values in the driven system, we need to know $P(q'; \mathcal{W})$. In Eq. (48), we allowed the probability distribution of the driven system to depend both on the general state variables characterizing the eigenstates of $H$ along with any additional quantum numbers $\mathcal{W}$ that might be selected to define various equilibrium ensembles (e.g., the sectors of positive and negative magnetization in low temperature Ising systems or qualitatively similar sectors describing the broken translational and rotational symmetries of an equilibrium low temperature crystal).

We next consider what occurs *if driven systems fail to equilibrate* at times $t' > t_f$ (when the parameters $\{q\}$ are no longer forcefully varied at a finite rate) and the system is effectively governed by the time independent Hamiltonian $H$ and the distribution $P(\epsilon')$ of energy densities as measured by the Hamiltonian $H$ will identically remain unchanged at all times $t' > t_f$. Towards this end, we remark that, for a system with any fixed time independent Hamiltonian $H$, the long time average of a general bounded operator $\mathcal{O}$ (that, unlike $\mathcal{O}_c$, need not commute with the Hamiltonian) is given by

$$\mathcal{O}_{l.t.a.} = Tr\left(\frac{\rho(t_f)}{\tilde{\mathcal{T}}}\int_{t_f}^{t_f+\tilde{\mathcal{T}}} dt' \mathcal{O}^H(t')\right)$$
$$= Tr\left(\frac{\rho_\tau(t_f+\tilde{\tau})}{\tilde{\mathcal{T}}}\int_{t_f+\tilde{\tau}}^{t_f+\tilde{\tau}+\tilde{\mathcal{T}}} dt' \mathcal{O}^H(t')\right). \quad (49)$$

Here, $\rho(t_f)$ the density matrix at the final time $t_f$ after which the Schorodinger picture density matrix no longer changes in time, the Heisenberg picture $\mathcal{O}^H(t') \equiv e^{iH(t'-t_f)/\hbar}\mathcal{O}e^{-iH(t'-t_f)/\hbar}$, and (as we have invoked it earlier) $\tilde{\mathcal{T}}$ is the said long averaging time. The instantaneous density matrix $\rho(t' > t_f)$ is constant in time if and only if the density matrix $\rho_{\tilde{\tau}}(t')$ of Eq. (26) is constant in time for $t' > t_f + \tilde{\tau}$. From the latter "if and only if" relation, the second line in Eq. (49) follows.

Now, by the Heisenberg equations of motion, for bounded operators $\mathcal{O}$, as $\tilde{\mathcal{T}} \to \infty$, the commutator $[H, \frac{1}{\tilde{\mathcal{T}}}\int_{t_f}^{t_f+\tilde{\mathcal{T}}} dt' \mathcal{O}^H(t')] = -\frac{i\hbar}{\tilde{\mathcal{T}}}\int_{t_f}^{t_f+\tilde{\mathcal{T}}} dt' \frac{d\mathcal{O}^H(t')}{dt'} = -\frac{i\hbar}{\tilde{\mathcal{T}}}\left(\mathcal{O}^H(t_f+\tilde{\mathcal{T}}) - \mathcal{O}^H(t_f)\right) = 0$. In other words, $\mathcal{O}_{l.t.a.}$ is trivially diagonal in the eigenbasis of the Hamiltonian [99]. For finite $\tilde{\mathcal{T}}$, there are corrections to the vanishing commutator that scale as $1/\tilde{\mathcal{T}}$. Since, in the long time

limit, $\mathcal{O}_{l.t.a.}$ commutes with the Hamiltonian, we may apply Eqs. (47,48). In particular, with the substitution $\mathcal{O}_c = \mathcal{O}_{l.t.a.}$, Eq. (48) will provide the long time averages of arbitrary observables $\mathcal{O}$. For any ergodic system in equilibrium, the thermal average of the operator of any long time average $\mathcal{O}_{l.t.a}$ of Eq. (49) is the equilibrium average. Substituting in Eq. (48), one thus explicitly has

$$\mathcal{O}_{l.t.a.} = \int dq' P(q'; \mathcal{W}) \langle \mathcal{O} \rangle_{eq;\{q'\};\mathcal{W}}. \qquad (50)$$

Along other lines, a similar conclusion was drawn in [21]. In what follows, we will ask whether an initially driven system may effectively saturate to a distribution $P(\epsilon')$ that relative to time independent Hamiltonian $H$ exhibits a vanishingly narrow ($\sigma_\epsilon = 0$ as in equilibrium systems) or to a finite width ($\sigma_\epsilon \neq 0$) distribution. In Section XII, we will consider a temperature ($T$) dependent $P(\epsilon')$.

## XI. EFFECTIVE EQUILIBRIUM IN DRIVEN SYSTEMS

In this subsection, we will further explore a closed system sans an environment explicitly included in the calculations. In this setting, given the general time ordered exponential $\mathcal{U}(t) = \mathcal{T} \exp(-\frac{i}{\hbar} \int_0^t H(t')dt')$, the Heisenberg picture Hamiltonian will evolve $H \rightarrow H^H(t) = \mathcal{U}^\dagger(t) H \mathcal{U}(t)$. It thus follows that any initial (Schrodinger picture) conventional equilibrium probability distribution $\rho = f(H)$ with $f$ a function of the Hamiltonian will trivially evolve as

$$\rho = f(H) \longrightarrow \rho(t) = f(H^H(t)). \qquad (51)$$

Thus, e.g., a general Boltzmann distribution $f$ in $H$ will evolve into a corresponding one in $H^H(t)$. Eq. (51) may further enable the proof of other relations [101]. A corollary of Eq. (51) is that

> • If the initial density matrix $\rho$ describes a system in thermal equilibrium then all of the usual thermodynamic relations may hold at later times with $H^H(t)$ being the Hamiltonian instead of $H$.

Specifically, in the equilibrium ensemble defined by $H^H(t)$, the system may exhibit equations of state. Thus, e.g., if the system started from a thermal state at inverse temperature $\beta$ (and, by Eq. (51), now has a (Schrodinger picture) density $\rho(t) = Z^{-1}(t)e^{-\beta H^H(t)}$ with $Z = Tr[e^{-\beta H^H(t)}]$) then in the Heisenberg picture all observables $\mathcal{O}$ of the driven system evolve according to

$$\boxed{\frac{d\mathcal{O}^H}{dt} = \frac{i}{\beta\hbar}[\mathcal{O}^H(t), \ln\rho(t)].} \qquad (52)$$

If the dynamics in a given driven system obey local equations of motion (as they typically do) then the commutators of $H^H(t)$ with general observables must be as well.

Equivalently, if at time $t = 0$ the original Hamiltonian was local then *it will remain local* in the time evolved Heisenberg observables of which it is a function. Eq. (52) trivially holds also in the classical limit (with the commutator replaced by Poisson Brackets (PB) in the usual way, $\frac{i}{\hbar}[A^H, B^H] \rightarrow -\{A, B\}_{PB} \equiv -\sum_\alpha(\frac{\partial A}{\partial x_\alpha}\frac{\partial B}{\partial p_\alpha} - \frac{\partial A}{\partial p_\alpha}\frac{\partial B}{\partial x_\alpha})$, with the sum over all generalized coordinates $x_\alpha$ and their conjugate momenta $p_\alpha$). Thus, rather explicitly, for any driven classical system,

$$\frac{d\mathcal{O}}{dt} = k_B T \{\ln\rho(t), \mathcal{O}(t)\}_{PB}. \qquad (53)$$

The new general result of Eq. (53) implies the same equation also for overdamped dissipative systems systems so long as their microscopics are governed by an underlying Hamiltonian (as, indeed, all systems are)) and thus includes earlier analysis (e.g., [102]) as particular limiting cases. Our general results of Eqs. (52, 53) further call into focus the important role of the modular Hamiltonian ($-\ln\rho$) studied in previous works [103]. If the temperature varies with time then in Eqs. (52, 53) the relevant value of the temperature $T$ (and of the inverse temperature $\beta$) is that of the initial equilibrium state [104].

If the system no longer varies (or varies weakly) in time (e.g., the system approaches a nearly stationary time independent Heisenberg picture Hamiltonian $H^H(t)$) then the probability density matrix $\rho = f(H^H(t))$ will become (nearly) time independent. In particular, all expectation values computed with the probability density will be (nearly) time independent in much the same way that they were in the original equilibrium distribution. Note that if the evolution $\mathcal{U}(t)$ and initial Hamiltonian $H$ are both spatially uniform then the resulting $H^H(t)$ defining the effectively equilibrated system will also be translationally invariant. Thus, if $f$ represents the Boltzmann or any other distribution, then the standard deviation of $H^H(t)/N$ as computed with $f(H^H(t))$ will be zero. However, as we explained in the earlier sections, the variance of the original Hamiltonian $H$ (not the variance of $H^H(t)$) can generally scale as $N^2$. Having such a large variance ($\sigma_H = \mathcal{O}(N)$) allows for (yet, of course, does not mandate) rapid dynamics under $H$. That is, the von Neumann equation $\frac{\partial\rho(t)}{\partial t} = \frac{i}{\hbar}[\rho(t), H]$ allows for stationary $\rho(t)$ regardless of the magnitude of $\sigma_H = N\sigma_\epsilon$. A trivial example is afforded by a Schrodinger picture density matrix that is diagonal in the eigenbasis of $H$, and thus trivially stationary once the system evolves under $H$ in the absence of external driving terms. Similar to the discussion following Eq. (36), the off-diagonal spread of $\rho$ determines its fluctuation frequencies. Indeed, some systems (e.g., glasses that we turn to next) do not adhere to the same equations of state as their true equilibrium state (e.g., equilibrium solid and fluid) counterparts yet may, nonetheless, appear stationary on very long time scales.

## XII. "TO THERMALIZE OR TO NOT THERMALIZE?"

The above question alludes to possible differences between (i) an *effective equilibrium* density matrix associated with a density matrix $\rho(t)$ (including those for the systems discussed in Section XI) that becomes nearly stationary (and thus a nearly constant $\rho_{\bar{\tau}}(t)$) at finite long times $t$ and (ii) the density matrix associated with the *truly asymptotic long time equilibrium* density matrix $\rho_{eq}$. Most of our focus thus far has been on intermediate times $0 \le t \le t_f$ during which the energy density (or any other intensive quantity $q$) varied. We showed that during these times, the standard deviation of $q$ may be finite, $\sigma_q = \mathcal{O}(1)$. Thus, the variation of general quantities $q$ (including, notably, the energy density or temperature) may trigger long range correlations. As we explained towards the end of Section V, this effect may be further exacerbated by "non-self-averaging" [29–32] found in disordered systems. Our inequalities of Eqs. (41, 42) hold for general fluctuations (regardless of the magnitude of their "classical" and "quantum" contributions [25] to the variance). In most systems coupled to an external bath, after the temperature or field no longer changes (e.g., when $|d\epsilon/dt|$ vanishes at times $t > t$) thermalization rapidly ensues already at short times after $t_f$. Indeed, there are arguments (including certain rigorous results) that "typical" states [91] might thermalize on times set by Planck's constant and the temperature, viz. the "Planckian" time scale $\mathcal{O}(\frac{h}{k_B T})$ encountered in Eq. (43). Other, exceedingly short (as well as long), equilibration time scales may be present [105]. The Planckian rate of Eq. (43) appears in a host of interacting systems, e.g., [106–111]. Various reaction times are often given by such minimal Planckian time scales multiplied by $e^{\Delta G/(k_B T)}$ with $\Delta G$ the effective Gibbs free energy barrier for the reaction or relaxation to occur, e.g., [111, 112]. However, some systems such as glasses do not achieve true equilibrium: measurements on viable experimental time scales differ from the predictions of the microcanonical or canonical ensemble averages. (The difference between the microcanonical and canonical ensembles is irrelevant for all intensive quantities in the absence of long range interactions for which "ensemble inequivalence" is known to appear [113–116]). In such cases (including, e.g., rapid supercooling of liquids that can lead to glass formation), the system may effectively exhibit self-generated disorder. As is well known, structural glasses are disordered relative to their truly thermalized crystalline counterparts. It is important to stress, however, that both structural glasses and crystalline solids are governed by the very same (disorder free) Hamiltonian. The effective disorder that glasses exhibit is not intrinsic but merely self-generated by the rapid supercooling protocol of non-disordered liquids. Thus, as hinted in Section X, the question remains as to whether, once the energy density or other intensive quantity no longer varies (e.g., once the glass is formed and its temperature is no longer low-

ered), the system will thermalize on experimental time scales (and display the rightmost distribution of Figure 1) or not be able to do so. Similar to Assumption (3) of Section IX A, starting from a glassy state, supercooled liquids can achieve their true equilibrium (crystalline) state only at asymptotically long times [122]. In systems that do not thermalize on experimental time scales, the discrepancy between equilibrium ensemble averages and empirical observables hints that the width $\sigma_\epsilon$ of the energy density might become smaller than it was during the cooling process yet is not vanishingly small. Indeed, if $\sigma_\epsilon = 0$ and no special "many body localized" states [54–60] exist (especially in the most relevant physical situation of more than one spatial dimension [61, 62]) *then the long time averages of all observables must be equal their microcanonical expectation values.* Specifically, similar to Eq. (50), the time average of a general quantity $\mathcal{O}$ over a long (finite) time $\mathcal{T}$ during which the probability distribution $P(q'; \mathcal{W})$ is nearly stationary will be identical to the equilibrium average, i.e., $\mathcal{O}_{l.t.a.} = \langle \mathcal{O} \rangle_{eq;\{q'\};\mathcal{W}}$ when the distribution $P(q'; \mathcal{W})$ is of a delta-function type nature in the energy density $\epsilon$ and all other intensive quantities $q$. If the expectation values of the thermodynamic equilibrium observables depend on the temperature or energy density (and are the same for all states related by symmetries of the Hamiltonian) then a deviation of the long time average values of observables $\mathcal{O}$ from their true equilibrium average values [21],

$$\mathcal{O}_{l.t.a.} \neq \langle \mathcal{O} \rangle_{eq;\{q\};\mathcal{W}} \tag{54}$$

will imply that *the width of the energy density may remain finite even after the system is no longer driven,* $\sigma_\epsilon > 0$. In glassy systems that, by their defining character, cannot achieve true equilibrium (and thus satisfy Eq. (54)) on relevant experimental time scales, the link to the external bath is effectively excised since the dynamics are so slow that little flow may appear. Here, the finite long time averages of Eq. (49) may be employed. If, in such instances, the probability density becomes time independent on measurable time scales then only an effective equilibrium (different from the true equilibrium defined by an equilibrium ensemble for the Hamiltonian defining the system) may be reached. That is, in systems with an effective equilibrium at sufficiently long times, see Section XI, the probability density $P(\epsilon')$ may be history independent and be a function of only a few global state variables yet differ from the conventional equilibrium statistical mechanics probability density in which the standard deviations of all intensive quantities vanish, e.g., $\sigma_\epsilon = 0$. Since the probability density determines all observable properties of the system, interdependences between the state variables (i.e., equations of state) may result [21]. Such a nearly static effective long time equilibrium distribution bears some resemblance to "prethermalization" in perturbed, nearly-integrable, models and other systems, e.g., [117–121]. Indeed, if local observables do not vary rapidly in time then, by the Heisenberg equations of motion, these observables nearly commute

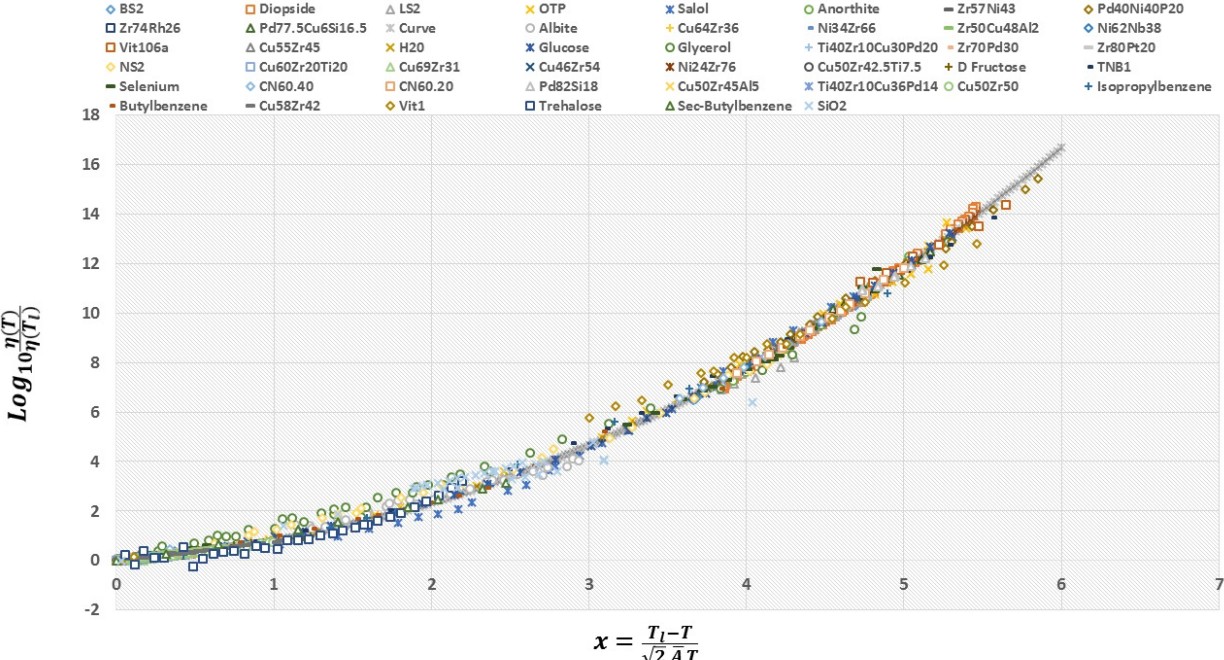

FIG. 7. (Color Online.) Reproduced from [123]. On the vertical axis, we plot the experimentally measured viscosity data divided by its value at the liquidus temperature ($\eta(T_l)$) as a function of a dimensionless temperature ratio. The viscosities of 45 liquids of diverse classes/bonding types (metallic, silicate, organic, and others) collapse on a single curve. The underlying continuous "curve" (more clearly visible at high viscosities where fewer data exist) is predicted by Eq. (56). Since $\overline{A}$ varies from fluid to fluid (albeit weakly) [123], the shown collapse does not imply a corresponding collapse of the viscosity as a function of $T_l/T$ nor as a function of $T_l/(\overline{A}T)$ (due, relative to the latter, to an additional shift along the $x$ axis that is set by $-1/(\overline{A}\sqrt{2})$).

with the Hamiltonian (and constitute nearly integrable constants of motion). We remark that by applying the Mastubara-Matsuda transformation [76] (similar to that invoked in Section VI B), we may map the prethermalized three-dimensional spiral spin states of [121] to establish the existence of long-lived effective equilibrium crystals of hard-core bosons. As stated above (see also Section XI), at asymptotically long times, systems such as glasses finally truly thermalize to true equilibrium solids [122]. However, prior to reaching the true equilibrium defined by the any of the canonical ensembles for the full system Hamiltonian, over very long finite times, the supercooled liquid/glass may display a nearly static distribution $P$ and thus obeys equations of state, absence of memory effects and other hallmarks of effective equilibrium. Even when the equilibrium averages $\langle \mathcal{O} \rangle_{eq;\{q'\};\mathcal{W}}$ feature non-analyticities at specific $q'$, the smeared average of Eq. (50) can be analytic (e.g., no measurable phase transitions might appear as $T$ is varied). In [21], we introduced this notion of an effective long time distribution $P$ of finite $\sigma_\epsilon$ and employed it to predict the viscosity of all glass formers. This prediction was later tested [123, 124] for the published viscosity data of all known glass formers when they are supercooled below their melting temperature. Figure 7 reproduces the result. Here, the (finite)

energy density width was set to be

$$\sigma_\epsilon = \overline{A}\frac{T(\epsilon_{melt} - \epsilon)}{T_{melt} - T}. \tag{55}$$

Here, $\overline{A} > 0$ is a liquid dependent constant ($0.05 \lesssim \overline{A} \lesssim 0.12$ for all liquids with published viscosity data [123, 124]). In equilibrium, such values of $\overline{A} \sim 0.1$ would be typically anticipated for effective classical harmonic solids/clusters (displaying a Gaussian distribution of the energy density with $\sigma_\epsilon = \sqrt{k_B T^2 C_v}/N_{eff}$ where the heat capacity $C_v = dN_{eff}k_B$ and $\epsilon = dk_BT$) of $N_{eff} \sim 30$ atoms in $d = 3$ dimensions. Albeit emulating such effective finite size equilibrium clusters, the energy densities $\epsilon$ and $\epsilon_{melt}$ in Eq. (55) are, respectively, those of the genuinely *macroscopic* supercooled liquid or glass at temperature $T < T_{melt}$ and at the melting (or "liquidus") temperature $T_{melt}$. The wide distribution of Eq. (55) mirrors that present in non-self-averaging disordered classical systems with an approximately linear in $T$ standard deviation and energy density $\epsilon(T)$. (All eigenstates of the density matrix may share the same energy while displaying a finite standard deviation $\sigma_\epsilon$.) In the models of Section VI (with the distribution of Eq. (14) that was far from the canonical normal form of equilibrium systems), the systems were driven by an external source whose effect on general quantities was cyclic in time. The situation may be radically different when the system is no

longer forcefully driven out of equilibrium yet, nonetheless, is still unable to fully equilibrate. If, as in equilibrium thermodynamics, the final state maximizes the Shannon entropy for a given energy then the probability distribution of the energy density will be a Gaussian of width $\sigma_\epsilon = \frac{T\sqrt{k_B C_v}}{N}$ and standard $\frac{1}{\sqrt{N}}$ fluctuations result. For systems of temperature $T$ that have not fully equilibrated, we may (as illustrated in the earlier Sections) find finite width $P_T(\epsilon')$. If the distributions $P_T(\epsilon')$ minimally differ in form from those in equilibrium then they may still be Gaussian with $\sigma_\epsilon \propto T$. Indeed, the general distribution that maximizes the Shannon entropy given a *finite* standard deviation $\sigma_\epsilon$ (and average energy density) is a Gaussian. Adhering to Occam's razor, the sole difference between the distribution of the energy density in equilibrium systems and those that we assume here for systems that have not yet achieved equilibrium is that in the latter systems $\sigma_\epsilon = \mathcal{O}(1)$ (while $\sigma_\epsilon = 0$ for equilibrium systems in their thermodynamic limit.) Non-rigorous considerations further suggest a Gaussian distribution once the system is no longer further cooled (or heated), see Appendix J [125, 126]. Assuming a normal distribution $P_T(\epsilon')$ of width $\sigma_\epsilon$, the viscosity $\eta$ of supercooled liquids at temperatures $T \le T_{melt}$ was predicted (by an application of Eq. (50)) to be [21],

$$\eta(T) = \frac{\eta_{s.c.}(T_{melt})}{erfc\left(\frac{\epsilon_{melt}-\epsilon(T)}{\sigma_\epsilon\sqrt{2}}\right)} = \frac{\eta_{s.c.}(T_{melt})}{erfc\left(\frac{T_{melt}-T}{AT\sqrt{2}}\right)}. \quad (56)$$

Eq. (56) is a direct consequence of Eqs.(48, 49). This prediction is indicated by the continuous curve in Figure 7. The coincidence between this non-perturbative prediction and the experimental data extends 16 decades of the viscosity increase and is a compilation of the analysis of the data of 45 fluids [123]; the corresponding dimensionless ratio in the argument of Eq. (56), $x \equiv \frac{T_{melt}-T}{AT\sqrt{2}}$ (the abscissa of Figure 7), varies up to a value of six. Unlike well known data collapse forms in equilibrium transitions and conventional critical phenomena in particular, the agreement between Eq. (56) and the experimental data does not wane for the larger $x$ (and viscosity) values. In fact, beyond an intermediate temperature range at which some scatter is seen in Figure 7, the quality of the data collapse improves as one progresses to lower temperatures more removed from the equilibrium melting temperature $T_{melt}$ [127]. At the so-called "glass transition temperature" $T_g$, the viscosity $\eta(T_g) = 10^{12}$ Pascal $\times$ second [128]. At lower temperatures $T < T_g$, the viscosity is so large that it is hard to measure it on experimental time scales. Apart from predictions for the viscosity, more general transition and relaxation rates may be investigated along similar lines [21, 129].

The very same distribution $P_T(\epsilon')$ invoked in deriving Eq. (56) may relate other properties of supercooled liquids and glasses to those of equilibrium systems. For instance, the experimentally measured thermal emission from supercooled fluids may differ in a subtle manner from one that is typical of equilibrium fluids. This de-

viation may be found by replacing Planck's law for the spectral radiance $I$ for photons of frequency $\nu$ in a system with well defined equilibrium temperature $T$ by a weighted average of Planck's law over effective equilibrium temperatures $T'$ that are associated with internal energy densities of equilibrium systems that are equal to $\epsilon'$,

$$I(\nu,T) = \frac{2h\nu}{c^3} \int dT' \frac{\tilde{P}_T(T')}{e^{h\nu/(k_B T')}-1} + I_{\mathcal{PTEI}}(\nu,T).$$

Here, $\tilde{P}_T(T') = P_T(\epsilon')c_v^{eq}$ (with the equilibrium specific heat capacity $c_v^{eq} \equiv \frac{d\epsilon'}{dT'}$) is the distribution of effective equilibrium temperatures $T'$ associated with the probability distribution $P_T(\epsilon')$ of the energy densities. The second term, $I_{\mathcal{PTEI}} \equiv \frac{2h\nu}{c^3} \frac{\int_{\mathcal{PTEI}} d\epsilon' P_T(\epsilon')}{e^{h\nu/(k_B T)}-1}$, captures viable contributions from any "Phase Transition Energy Interval" [21] (wherein the energy density $\epsilon'$ of an equilibrium system may vary by an amount set by the latent heat without concomitant changes in the corresponding equilibrium temperature $T'$). More accurately stated, in Eq. (57), we may replace $\frac{2h\nu}{c^3(e^{h\nu/(k_B T')}-1)}$ by $u(\nu,T')$- the energy density carried by photons of frequency $\nu$ when the equilibrium system is at a temperature $T'$ [21]. We highlight that this prediction for the emission spectrum $I(\nu,T)$ is determined by the same distribution predicting the viscosity collapses of Figure 7 (the Gaussian $P_T(\epsilon')$ of the width given by Eq. (55)). As such, this prediction may, in principle, be experimentally tested. Similarly, the temperature dependence of other observables (including various response functions) may be expected to have the same increase in the time scale as that characterizing the viscosity. Indeed, the time dependent heat capacity response follows exhibits a dynamical time that increases with temperature in a manner similar to the viscosity, e.g., [130]. We suspect that this increase in the relaxation time scale as the temperature is dropped may account very naturally for the experimentally observed smooth specific heat peak[131] near the glass transition temperature $T_g$ when the system is heated from lower temperatures (consistent with $T_g$ marking a dynamical crossover rather than a bona fide thermodynamic transition [123, 124]). This is so since, at temperatures $T \le T_g$, on the time scales of the experiment, the system is essentially static (e.g., the viscosity of the Eq. (56) and the associated measured relaxation times are large). Consequently, the relatively stable nearly static structures that appear once the glass is formed need not significantly respond to a small amount of external heat. The situation is somewhat reminiscent of the extensive latent heat that is required to melt equilibrium crystals. Pronounced thermodynamic changes appear at the transition between equilibrium fluids and crystals. Once the supercooled liquid or glass becomes effectively static on experimental time scales at $T_g$, it may weakly emulate the latent heat signature of the equilibrium liquid to solid transition sans having true latent heat required to elevate the temperature. Contrary to the weak peak in the

heat capacity on heating, when the system is cooled from temperatures above $T_g$, the heat capacity typically drops monotonically near $T_g$ and does not exhibit a peak (this may reflect a memory of larger mobility at higher temperatures). A finite $\sigma_\epsilon$ may naturally allow for a finite width temperature interval about $T_g$ where the empirically observed crossover in the heat capacity and/or other quantities can appear on experimental time scales. In line with our earlier discussion concerning general properties stemming $P_T(\epsilon')$, Eqs. (48,50) [21] further suggest that similar features may appear at other temperature at which other crossovers appear (i.e., the ratio of the width of the temperature range where a crossover is observed to the crossover temperature itself may be set by the scale of dimensionless parameter $\overline{A}$ appearing in Eq. (56) for the viscosity). Indeed, simple estimates illustrate that experimentally observed heat capacity crossover region is of the same scale as $\overline{A}T_g$ [133]. This broadening due to the finite $\sigma_\epsilon$ may supplant any existing features of the equilibrium system (having $\sigma_\epsilon = 0$). More general than heat capacity measurements alone, we stress that, experimentally, supercooled liquids indeed exhibit effective smooth crossovers instead of true singularities associated with thermodynamic phase transitions that appear at well defined transition temperatures. Thus, our suggestion is that the size of the temperature interval over which these crossovers arise/are enhanced as a result of smearing by the finite width distribution $P_T(\epsilon')$ is set by the effective crossover temperature scale multiplied by $\overline{A}$. An energy density distribution of a finite width $\sigma_\epsilon$ allows for a superposition of low energy density solid type eigenstates (that may break continuous translational and rotational symmetries) and higher energy density liquid type eigenstates [21]. Such a general combination of eigenstates does not imply experimentally discernible equilibrium solid (crystalline) order. Sharp Bragg peaks need *not appear* in states formed by superposing eigenstates that, individually, display order [21, 132]. This absence of ordering reflects the possible lack of clear structure when, e.g., randomly superposing different Fourier modes with each Fourier mode displaying its defining periodic order. In an interesting preprint [134] that appears after an earlier version of the current paper [135], it was found that effectively superposing (periodically replicated) finite size states of 16 or 24 atoms (so as have these states as unit cells that are repeated to span all space) according to their Boltzmann weights accurately reproduces the structure factor of certain structural glasses. This latter result is in accord with our approach to glasses; the size of these 16 and 24 atom states is not too dissimilar from the order of magnitude estimate, provided earlier in this Section, of the requisite number of atoms $N_{eff} \sim 30$ in an effective equilibrium solid that would lead to a Gaussian distribution of a width consistent with our theory of glasses and the ensuing collapse of Figure 7. It will be interesting to examine in more quantitative detail whether distributions associated with states similar to those examined in [134] adhere to the normal form that we invoked for $P$.

The mixing of eigenstates of different energy densities over a range set by $\sigma_\epsilon$ further suggests the appearance of non-uniform local dynamics. Interestingly, in accord with this consequence of our theory, *dynamical heterogeneities* are empirically ubiquitous in supercooled fluids [136–140]; these large fluctuations are still present even after the fluids remain in contact with an external bath for a long time. A simple calculation illustrates that the long time fluctuations in the *local* energy density given a general initial state relate to the width of this state in the eigenbasis of the Hamiltonian governing the system [141]. The presence of a spatially non-uniform energy density is very natural during general heating or cooling processes (e.g., the exterior parts of a system being supercooled may be colder than its interior, see also the discussion towards the end of Section IV). Once supercooling stops, heat may diffuse through the system yet heterogeneities (that are borne in our framework from a distribution of finite $\sigma_\epsilon$) may persist for a long time [142].

Eq. (50) that enabled the prediction of the viscosity of Eq. (56) and others quantities does not rely on quantum effects. An advantage of the quantum approach described in this Section is that it allows for an accurate definition of the (eigen)states of the systems as opposed to the more loosely defined classical microstates in which Planck's constant needs to be introduced by hand in order to produce a dimensionless number of states from phase space volumes [143]. Furthermore, classically, one often needs to integrate the equations of motion numerically in order to obtain results for various particular systems (this is particularly time consuming for slow glassy systems). Alternatively, if numerics are to be avoided, assumptions may be made about the classical energy landscape and configurational entropies. The quantum treatment invoked in this Section is devoid of such assumptions. Nonetheless, one may still translate the more fundamental and precise quantum description into a corresponding classical one [21, 123].

## XIII. POSSIBLE EXTENSIONS TO ELECTRONIC AND LATTICE SYSTEMS

The spin and hard core Bose models of Section VI were defined on lattices. In this Section, we will speculate and further discuss possible extensions to other, experimentally relevant, theories and lattice systems. The electronic properties of many materials are well described by Landau Fermi Liquid Theory [144–147]. This theory is centered on the premise of well defined quasiparticles leading to universal predictions. Recent decades have seen the discovery of various unconventional materials displaying rich phases [144, 148–167] that often defy Fermi liquid theory. Given the results of the earlier Sections, it is natural to posit that as these systems are prepared by doping or the application of external pressure and fields (in which case the varied parameter $q$ may be the carrier density, specific volume, or magnetization),

a widening $\sigma_q$ will appear during the process. This wide distribution might persist also once the samples are no longer experimentally altered. In such cases, the density matrices (and associated response functions) describing these systems may exhibit finite standard deviations $\sigma_q > 0$. The broad distribution may trigger deviations from the conventional behaviors found in systems having sharp energy and number densities ($\sigma_n = 0$) or, equivalently, sharp chemical potentials and other intensive quantities. Theoretically, non-Fermi liquid behavior may be generated by effectively superposing different density Fermi liquids (with each Fermi liquid having a sharp carrier concentration $n$) in an entangled state. Systems harboring such an effective distribution $P(\mu')$ of chemical potentials may be described by a mixture of Fermi liquids of different particle densities. Any non-anomalous Green's function is manifestly diagonal in the total particle number. Thus, the value of any such Green's function may be computed in each sector of fixed particle number and then subsequently averaged over the distribution of total particle numbers in order to determine its expected value when $\sigma_n \neq 0$. In particular, this implies that the conventional jump (set by the quasiparticle weight $Z_{\vec{k},\mu'}$) of the momentum space occupancy [144–147], in the coherent part of the Green's function ($G = G_{coh} + G_{incoh}$) will be "smeared out" when $\sigma_\mu \neq 0$. Similar to Eq. (48), a distribution of chemical potentials will lead to the replacement of the coherent Green's function of ordinary Fermi liquids by

$$G_{coh}(\vec{k}, \omega) = \int d\mu' P(\mu') \frac{Z_{\vec{k},\mu'}}{\omega - \epsilon_{\vec{k}} + \mu' + i/\tau_{\vec{k},\mu'}}. \quad (57)$$

Here, $\tau_{\vec{k},\mu'}$ is the quasi-particle lifetime in a system with sharp $\mu'$ at wave-vector $\vec{k}$. The denominator in Eq. (57) corresponds to the coherent part of the Green's function of a Fermi liquid of a particular chemical potential $\mu'$ and quasi-particle weight $Z = 1$ [144–147]. Qualitatively, Eq. (57) is consistent with indications of the very poor Fermi liquid type behavior reported in [169]. The effective shift of the chemical potential in Eq. (57) is equivalent to a change in the frequency dependence while holding the chemical potential $\mu$ fixed; the resulting nontrivial dependence of the correlation function on the frequency (with little corresponding additional change in the momentum) is, qualitatively similar to that advanced by theories of "local Fermi liquids", e.g., [144, 168]. Our considerations suggest a similar smearing with the distribution $P(\mu')$ will appear for any quantity (other than the Green's function of Eq. (57)) that is diagonal in the particle number. Analogous results will appear for a distribution of other intensive quantities. The prediction of Eq. (57) (and similar others [21] in different arenas) may be tested to see whether a single consistent probability distribution function $P$ accounts for multiple observables. General identities relate expectation values in interacting Fermi systems to a weighted average of the same expectation values in free fermionic systems [170]. These relations

raise the possibility of further related smeared averages, akin to those in Eq. (57), in numerous systems. Numerically, in various models of electronic systems that display non-Fermi liquid type behaviors, the energy density differences between contending low energy states $\{|\psi^\alpha\rangle\}$ (not necessarily exact eigenstates) are often exceedingly small, e.g., [171]. Since these states globally appear to be very different from one another, the matrix element of any local Hamiltonian between any two such orthogonal states vanishes, $\langle\psi^\alpha|H|\psi^\beta\rangle = 0$ for $\alpha \neq \beta$. We notice that, given these results, arbitrary superpositions of these nearly degenerate states, $\sum_\alpha a_\alpha|\psi^\alpha\rangle$, will have similar energies. Thus, for many body Hamiltonians modeling these systems, a superposition of different eigenstates may be natural from energetic considerations. Towards the end of Section XII, we remarked on the viable disordered character of the states formed by superposing eigenstates that break continuous symmetries. We now briefly speculate on the corresponding situation for eigenstates in electronic lattice systems that break discrete point group symmetries on a fixed size unit cell. Here, due to the existence of a finite unit cell in reciprocal space, a superposition of eigenstates that are related to each other by a finite number of discrete symmetry operations may not eradicate all Bragg weights. In other words, order may partially persist when superposing states on the lattice that, individually, display different distinct structures.

## XIV. ADIABATIC PROCESSES, THERMALIZATION, AND QUANTUM MEASUREMENTS

As we demonstrated in the current work, rapidly driven systems may exhibit uncertainties in their energy and/or other densities. We now close our circle of ideas and focus on the diametrically opposite case of unitary evolutions- slow adiabatic processes (for which, obviously, $dq/dt = 0$); this discussion will complement that of Section XI. In this Section, we will further speculate on relations concerning thermalization that superficially emulate those of quantum measurements. In line with the focus of the current work, the latter purely hypothetical connections suggest that the absence of thermalization may allow for broad distributions.

As well known, a basic tenet of quantum mechanics is that a measurement will project or "collapse" a measured system onto an eigenstate of the operator being measured. A natural question to ask is whether such effective projections may merely emerge as a consequence of an effective very rapid thermalization of microscopic systems. To motivate this query and more generally examine effectively adiabatic processes, we consider a Hamiltonian

$$H_{A \cup B}(t) = H_A + H_{AB}(t) + H_B \quad (58)$$

describing the combined system of two systems and the coupling between them ($H_{AB}$). This Hamiltonian em-

ulates $\tilde{H}$ of the subsystem-environment hybrid of Section IX. We first examine what occurs when the coupling $H_{AB}(t)$ changes adiabatically from zero. Consider the situation wherein, initially, at times $t \leq 0$, systems $A$ and $B$ were in respective eigenstates $|\phi_{n_A}\rangle$ and $|\phi_{n_B}\rangle$ of $H_A$ and $H_B$. That is, at times $t \leq 0$, the state of the combined system $A \cup B$ was described by the product state of these two eigenstates. We further assume that at times $t < 0$, the coupling $H_{AB}(t) = 0$ and for times $t \geq 0$ an adiabatic change of $H_{AB}(t)$ ensues. Under these conditions, by the adiabatic theorem, at any later time $t$, the initial state has evolved into a particular eigenstate $|\phi_{n_{AB}}(t)\rangle$ of $H_{A \cup B}(t)$, we have $|\phi_{n_A}\rangle |\phi_{n_B}\rangle \to |\phi_{n_{AB}}(t)\rangle$. We may expand the density matrices $\rho_{A,B}$ of the initial system $A$ and $B$ in terms of the eigenvectors of $H_A$ and $H_B$. Explicitly expressing the density matrix $\rho_{A \cup B}(t)$ of the combined system at time $t$ in the eigenbasis of $H_{measure}(t)$, i.e., in the evolution from $t = 0$ to a time $t > 0$, the density matrix trivially evolves as

$$\sum_{n_A n_B} \rho_{n_A n_B} |\phi_{n_A}\rangle |\phi_{n_B}\rangle \langle \phi_{n_A}| \langle \phi_{n_B}|$$
$$\to \sum_{n_A n_B} \rho_{n_A n_B} |\phi_{n_{AB}}(t)\rangle \langle \phi_{n_{AB}}(t)|. \qquad (59)$$

Hence, if both systems $A$ and $B$ start from equilibrium (and thus have sharp energy densities- i.e., if at $t = 0$ the eigenstates of $H_A$ and $H_B$ of significant amplitude were clustered around a given energy density) then an adiabatic evolution of $H_{AB}(t)$ will yield a density matrix $\rho_{A \cup B}$ having a sharp energy density, $\sigma_{\epsilon_{A \cup B}}(t) = 0$. Thus, the notion that sufficiently slow processes enable systems to remain in equilibrium is indeed consistent with the adiabatic theorem of quantum mechanics.

We next comment on how such adiabatic processes (and later briefly discuss more general thermalization events that need not be adiabatic) may superficially emulate certain features of a wavefunction collapse. Towards that end, we consider the extreme case of a microscopic system $A$ ("being measured") and a macroscopic system $B$ that we may regard as an environment that includes a coupling to an experimental probe at the measurement time $t_{measure}$. As earlier, for a general adiabatic evolution, $|\phi_{n_{AB}}(0)\rangle \to |\phi_{n_{AB}}(t_{measure})\rangle$. We now allow the coupling $H_{AB}(t)$ to be non-vanishing at all times $t$ (i.e., also including times $t \leq 0$) and, due to its ease, first briefly discuss the case when its evolution is adiabatic.

Under these circumstances, by the adiabatic theorem, $|\phi_{A \cup B}(t_{measure})\rangle$ must be an eigenstate of $H_{A \cup B}(t_{measure})$. Thus, such an adiabatic evolution emulates an effective "collapse" onto an eigenstate of the Hamiltonian that measures the state of the microscopic system $A$. We emphasize that the state $|\phi_{A \cup B}(t_{measure})\rangle$, describing both the microscopic system $A$ and the large system $B$, will be in an eigenstate of $H_{A \cup B}(t_{measure})$- i.e., not only the small system $A$ will be altered by the measurement. While, at any time $t$, the state $|\phi_{A \cup B}(t)\rangle$ is an eigenstate of $H_{A \cup B}(t)$, its highly entangled content largely remains unknown. Thus, unique

predictions for the outcome of other future evolutions cannot be made in such a case. It may be noted that certain "realistic" setups involving quantum measurements often entailing higher energy "thermal" states of the measurement device (e.g., the reaction between silver ions and the screen that they strike in a Stern-Gerlach type experiment creating visible spots on a screen). The collapsed system is in an excited state.

The effective "collapse" brought about by such an adiabatic process may be nearly immediate for microscopic systems $A$. Typical lower bounds on time scales for adiabatic processes defined by an energy difference $\Delta E$ are set by $\hbar/\Delta E$ (for precise bounds see, e.g., [176]). Such scales are consistent with the uncertainty relations and our bounds of Section IX. For small energy splittings $\Delta E$, this adiabatic time scale may become large. The above discussion of a hypothetical adiabatic evolution is merely illustrative. A potentially more practical question concerning realistic $H_{AB}(t)$ is that of the thermalization of the full system. At room temperature, the "Planckian time" scale for the equilibrium thermalization of random initial states [91] (see also Section IX B) is $h/(k_B T) \sim 10^{-13}$ seconds (e.g., the typical period of a thermal photon). The latter time scale may be smaller than that required for an adiabatic evolution yet is still finite; one may attempt to probe for such an effective finite time collapse borne by thermalization (cf., any such deviations from the textbook "instantaneous collapse") only at extremely low temperatures. The very rapid thermalization evolution suggested here allows for multiple measurement outcomes with different probabilities. A measurement provides only partial information on the many body entangled state $|\phi_{A \cup B}(t)\rangle$ formed by $A$ and $B$- it does not specify it. Conditional probabilities may be assigned to the possible future evolutions of this entangled state (and thus of future measurement outcomes thereof). In this regard, our suggestion concerning thermalization is somewhat similar to existing frameworks including "Quantum Bayesianism" [177] and others relying on entanglement, e.g., [178, 179].

Further parallels between equilibration and certain features of an effective collapse in quantum measurements are partially motivated by the Eigenstate Thermalization Hypothesis [45–53]. When valid, this hypothesis allows us to effectively equate the results of local measurements of general observables $\mathcal{O}$ in pure eigenstates $\{|\phi_n\rangle\}$ (of energies $\{E_n\}$) of a general Hamiltonian (including Hamiltonians describing a coupling between a measurement device and a microscopic system) with expectation values in equilibrated thermal systems defined by the full system Hamiltonian,

$$\langle \phi_n | \mathcal{O} | \phi_n \rangle = Tr(\rho(E_n)\mathcal{O}). \qquad (60)$$

Here, $\rho(E_n)$ is an equilibrium density matrix associated with the energy $E = E_n$ (and, when applicable, any other conserved quantities defining the state $|\phi_n\rangle$ and the thermal system). Taken to the extreme, Eq. (60) suggests that we may relate two seemingly very different concepts:

(**i**)*An effective collapse to an eigenstate.* The lefthand side of Eq. (60) yields the results of quantum expectation values associated with (projecting the system onto) eigenstates of the Hamiltonian (also describing, as in a realization of Eq. (58), the measurement process- the substantial coupling $||H_{AB}|| \gg ||H_A||$ of the environment ($B$) containing a measurement device to the measured quantity ($A$) and) providing the dynamics.

(**ii**) *Equilibration.* The righthand side of Eq. (60) reflects the outcomes of equilibration (in which, inasmuch as any observable $\mathcal{O}$ can inform, the system effectively becomes indistinguishable from an eigenstate of the very same Hamiltonian associated with item (**i**)). As noted above, in a realization of Eq. (58) describing a typical measurement, this Hamiltonian displays a dominant coupling between the measurement device and the quantity being measured, $||H_{AB}|| \gg ||H_A||$.

More general than adiabatic processes alone, thermalization indeed shares other commonalities with quantum measurements. Just as a quantum measurement (and ensuing collapse) is not a time reversal invariant operation [180] so, too, is a typical finite $T$ thermalization process in a highly entangled many body system. The second law of thermodynamics is consistent with an evolution of the entangled $A \cup B$ system displaying a non-decreasing entropy upon performing consecutive measurements (compatible with indeterminate outcomes for other subsequent measurements thereafter). A notional link between (**i**) and (**ii**) is naturally compatible with the appearance of wide distributions of various measurable quantities in non-equilibrium systems. Regardless of the validity of the Eigenstate Thermalization Hypothesis of Eq. (60), any equilibrium expectation value is an ensemble average over states having a sharp value of intensive state variables $q$. Thus, as alluded to in Section X, barring special eigenstates [54–62], the system may rather straightforwardly exhibit non-equilibrium behaviors if the distribution of its intensive thermodynamic state variables $q$ is, quite simply, not a delta function. We conclude this Section by underscoring that (as we explained in several of the previous Sections) the central result of the current paper regarding the existence of wide distributions in non-equilibrium systems does not rely on quantum effects nor the character of quantum measurement (on which we speculated above). Similar behaviors are anticipated in classical systems. The use of the quantum language in the current article merely made our considerations more precise and also gave rise to the bounds of Section IX.

## XV. CONCLUSIONS

We illustrated that a finite rate variation of general intensive quantities may lead to long range correlations. In the simplest variant of this effect, in systems having varying intensive observables $q$ (such as the energy density $\epsilon$) for which $\frac{dq}{dt} = \mathcal{O}(1)$, an average connected two site correlation functions need not vanish even for sites are arbitrarily far apart. Trivial extensions hold for weaker variations of the intensive quantities. For instance, if only short range effects of the environment appear and, consequently, for an $N$ site system residing in $d$ spatial dimensions, $\frac{dq}{dt} = \mathcal{O}(N^{-1/d})$ then the average value of the connected two point correlation function for an arbitrary pair $(i, j)$ of far separated sites may be asymptotically bounded as $\overline{G} \geq \mathcal{O}(N^{-2/d})$.

In the quantum arena, the general non-local correlations that we found relate to the macroscopic entanglement present in typical thermal states. Our results highlight that, even in seemingly trivial thermal systems, one cannot dismiss the existence of long range correlations. Our analysis of non-equilibrium systems does not appeal to conventional coarsening and spinodal decomposition phenomena (although the departure from a spatially uniform true equilibrium state in spinodal systems is very naturally consistent with a distribution of low energy solid like and higher energy fluid like states). Cold atom systems may provide a controlled testbed for our approach. We speculate that our results may also appear in naturally occurring non-equilibrium systems. As we explained (Section XII), the peculiar effect that we find may rationalize the unconventional behaviors of glasses and supercooled fluids. Our effect might further appear in electronic systems that do not feature Fermi liquid behavior (Section XIII). Here, a broad distribution of effective energy densities and/or chemical potentials may appear. The validity of weighted averages such as that of Eq. (57) may be assessed by examining whether a unique distribution $P$ simultaneously accounts for all measurable quantities. The long range correlations that we derived *do not hinge* on the existence of non-local interactions; the examples that we studied in Section VI embodied quintessential local interactions. In Section XIV, we illustrated how adiabatic processes maintain sharp thermodynamic quantities and speculated that a nearly instantaneous equilibration of small systems with macroscopic ones may emulate certain features of quantum measurements. We hope our suggested effect and analysis will be further pursued in light of their transparent mathematical generality and ability to suggest new experimental behaviors (e.g., the universal viscosity collapse of supercooled liquids that it predicted and is indeed empirically obeyed over sixteen decades as seen in Figure 7).

While deriving the above, we arrived at additional results en route. These include the finding of *universal bounds relating thermalization and time derivatives of general observables* (Section IX B), explaining how driven system may be described by an effective equilibrium distribution in which *the dynamics are universally generated by the logarithm of the corresponding probability density matrix* (Section XI), and speculatively pointing to similarities between unitary dynamics, thermalization, and quantum measurements (Section XIV). Additional technical details have been relegated to the Appendices. In Appendix J, we further motivate *the appearance of long time Gaussian distributions in both equilibrium and non-*

*equilibrium systems.*

**Acknowledgments**

I am grateful to S. Gopalakrishan for encouraging me to write up these results and am very thankful to interest by and discussions with M. Alford, N. Andrei, R. Bruinsma, K. Dahmen, E. Demler, A. Dymarsky, S. Ganeshan, A. K. Gangopadhyay, S. Gopalakrishan, E. Henriksen, D. A. Huse, K. F. Kelton, A. Kuklov, A. J. Leggett, K. Murch, F. Nogueira, S. Nussinov, V. Oganesyan, G. Ortiz, A. Polkovnikov, L. Rademaker, S. Ryu, C. Sa de Melo, M. Schossler, A. Seidel, D. Sels, K. Slagle, and N. B. Weingartner. I wish to thank S. Nussinov and L. Rademaker for a critical reading of the manuscript. This work was principally supported the National Science Foundation under grant NSF 1411229 and further supported by NSF PHY17-48958 (KITP) and NSF PHY-160776 (the Aspen Center for Physics).

## Appendix A: Order of magnitude estimates on the minimal time for changing the energy density (and establishing long range correlations) and the continuity equation

Given radiation traveling at a speed $c$, during a time interval $\Delta t$, an extensive (i.e., volume proportional) amount of radiative heat $\Delta Q_{rad}$ may flow into a hypercubic $d$ dimensional system of spatial dimensions $L \times L \times \cdots \times L$ whenever $L \lesssim c(\Delta t)$. Thus, bulk effects from radiative heat exchange may only be present only after a sufficiently long time $t \gtrsim L/c$ from the instant at which radiative heating or cooling begins. Similarly, if the effective radiative absorption lengths $\ell_S$ and $\ell_B$ of, respectively, the media comprising the system and the surrounding heat bath both satisfy $\ell_{S,B} \gtrsim L$ then the total radiative heat flow rate out of or into the system may be proportional to the volume of the system, $đQ_{rad}/dt = \mathcal{O}(V)$. The existence of a minimal time scale also appears in non-relativistic systems as may be proven from the Lieb-Robinson bounds (see Appendix B).

We now briefly provide order of magnitude estimates. If, e.g., $L$ is the order of 1cm for a macroscopic sample for an index of refraction $\sim 1$ and, the relevant velocity $v = c$ is a typical radiation speed (as in, radiative cooling or heating) then the requisite minimal time scale will be exceedingly short

$$ t_{\min} = \frac{L}{c} \sim 3 \times 10^{-11} \text{sec}. \qquad (A1) $$

Experiments on supercooled liquids typically involve cooling at a rapid finite rate (thus, the experimental time scale $t \geq t_{\min}$). In metallic liquids (that form glasses when supercooled), often in experiments one uses (radiative) laser beam heating. In typical metals, both the heat and charge effectively travel at a finite fraction (typically of the order of $10^{-2}$) of the speed of light $c$ the effective speed of electrons in a metal; both effective heat and charge transport velocities are possibly the same in

conventional metals obeying the Wiedemann-Franz law). The speed/rate of heat transfer is bounded from below by the speed/rates of any of the individual (radiative/conduction/convection) processes that contribute to it. Thus, if either the typical radiative or conductive processes occur at speeds associated with a finite fraction of the speed of light $c$ then so, too, is the total heat transfer. In metals as well as in systems where the radiative penetration depth is larger than or of the scale of the linear dimension of the material, the speed associated with heat transfer is rather large and, correspondingly, the minimal time scale can, in these instances, become very short.

These considerations may also be formally and rather trivially rationalized from the continuity equation applied to the local energy density, $\partial_t \epsilon(\vec{x}) + \vec{\nabla} \cdot \vec{j}(\vec{x}) = 0$ where $\vec{x}$ denotes a spatial location in the continuum limit. If the average current flowing through the system surface $|\bar{j}| \equiv |\epsilon| v_Q$ where $\epsilon$ is the global average of the local energy density and $v_Q$ a speed characterizing heat or energy flow through the system boundary (of surface area $\tilde{A}$) and the volume $V = \mathcal{O}(\tilde{A}L)$ then the rate of energy density change is $(dE/dt)/E = \mathcal{O}(v_Q/L)$. In other words, the minimal time required to change the system energy by an extensive amount is proportional to $L$.

## Appendix B: A finite rate of change of intensive quantities is only possible within the Lieb-Robinson light cone

In driven systems with $d\epsilon/dt = \mathcal{O}(1)$, the commutators with expectation values equal to $dE/dt$ must be extensive. Specifically, both in (1) closed systems with a time dependent Hamiltonian (as in, e.g., Section VII), the commutator $[H^H(t), H]$ (where $H^H(t)$ is the Heisenberg picture Hamiltonian) as well as in (2) settings similar to those in Sections IV (Eq. (4) therein) and IX, namely a subsystem with Hamiltonian $H$ in contact with the full system of Hamiltonian $\tilde{H}$, where the relevant commutator is given by Eq. (31), the above two-Hamiltonian commutators are of order $\mathcal{O}(N)$. In both (1) and (2), for local Hamiltonians, one may examine the constraints implied by causality as these appear via the Lieb-Robinson bound [27] for commutators $[\mathcal{A}_H(t), \mathcal{B}(0)]$ of local Heisenberg picture operators $\mathcal{A}$ and $\mathcal{B}$ that have their support centered about sites $i$ and $j$. In particular, whenever the Lieb-Robinson bound applies, the operator norm ($|| \cdot ||$) of commutators between any two local quantities $\mathcal{A}$ and $\mathcal{B}$ is bounded from above by

$$ |||[\mathcal{A}_H(t), \mathcal{B}(0)]||| \leq c' e^{(-a(|i-j| - v_{\mathsf{LR}}|t|))}. \qquad (B1) $$

Here, $a$ and $c'$ are constants and $v_{\mathsf{LR}}$ is the Lieb-Robinson speed of Section IV. The Lieb-Robinson speed plays the role of the velocity of light in relativistic theories. Since, by the Heisenberg equations of motion, the commutators in both cases (1) and (2) have an average given by the derivative of the energy $dE/dt$ and since the latter is of order $N$, i.e., $dE/dt = \mathcal{O}(N)$ when the energy

density varies at a finite rate, the upper bounds on the two Hamiltonian commutators must also be of order $N$. Equivalently, as we next detail, the Lieb-Robinson "light cone" [27], during the times at which the energy density as measured by $H/N$ varies at a non-zero rate, is of the scale of the entire system.

The Schrodinger picture Hamiltonian $\tilde{H}$ of the combined system ($\mathcal{S}$) + environment ($\mathcal{E}$) hybrid may be expressed as $\tilde{H} = H + H_{\mathcal{S}-\mathcal{E}} + H_{\mathcal{E}}$ where $H$ is the system Hamiltonian, $H_{\mathcal{S}-\mathcal{E}}$ denotes the coupling of the system to its environment, and $H_{\mathcal{E}}$ is the Hamiltonian of the environment. When only bounded local interactions appear in the system-environment hybrid, we will write the Hamiltonians (in the form of Eq. (1) (explicitly rewritten below) and its generalization),

$$H = \sum_i \mathcal{H}_i \qquad (B2)$$

and

$$H_{\mathcal{S}-\mathcal{E}} + H_{\mathcal{E}} = \sum_{j'} \mathcal{H}_{j'} \qquad (B3)$$

as, respectively, sums of the bounded local operators $\{\mathcal{H}_i\}$ and $\{\mathcal{H}_{j'}\}$. In what follows, as throughout the main text of the paper, $\tilde{\rho}$ will denote the density matrix of the system-environment hybrid. By Heisenberg's equations of motion, with $\epsilon = \frac{1}{N} Tr[\tilde{\rho} H^H(t)]$ the energy density of the system (where $H^H(t) = \tilde{\mathcal{U}}^\dagger(t) H \tilde{\mathcal{U}}(t)$, with $\tilde{U}(t) = e^{-i\tilde{H}t/\hbar}$, for a time independent $\tilde{H}$ in Eq. (30)), the derivative $i\hbar \frac{d\epsilon}{dt}$ is given by

$$\frac{1}{N} \sum_i Tr(\tilde{\rho}[\mathcal{H}_i^H(t), \tilde{H}])$$

$$= \frac{1}{N} \sum_i Tr(\tilde{\rho}[\mathcal{H}_i^H(t), H^H(t) + H_{\mathcal{S}-\mathcal{E}}^H(t) + H_{\mathcal{E}}^H(t)])$$

$$= \frac{1}{N} \sum_i Tr(\tilde{\rho}[\mathcal{H}_i^H(t), H_{\mathcal{S}-\mathcal{E}}^H(t) + H_{\mathcal{E}}^H(t)]). \qquad (B4)$$

The first equality of Eq. (B4) invoked the trivial invariance of $\tilde{H}$ under time evolution with $\tilde{\mathcal{U}}(t) = e^{-i\tilde{H}t/\hbar}$ (i.e., $\tilde{H} = \tilde{\mathcal{U}}^\dagger(t)\tilde{H}\tilde{\mathcal{U}}(t) = \tilde{H}^H(t)$). The last equality in Eq. (B4) follows since, in the second commutator, $H^H(t) = \sum_i \mathcal{H}_i^H(t)$ similarly commutes with itself. For all times $t > 0$, the norm of the above commutator average

$$\frac{1}{N} |\sum_i Tr(\tilde{\rho}[\mathcal{H}_i^H(t), H_{\mathcal{S}-\mathcal{E}}^H(t) + H_{\mathcal{E}}^H(t)])|$$

$$= \frac{1}{N} |\sum_{i,j'} Tr(\tilde{\rho}[\mathcal{H}_i^H(t), \mathcal{H}_{j'}^H(t)])|$$

$$\leq \frac{c'}{N} \sum_{i,j'} e^{-a(|i-j'|-v_{LR}t)}. \qquad (B5)$$

The decomposition of the system Hamiltonian $H = \sum_i \mathcal{H}_i$ into a sum over local regions spans $N' = \mathcal{O}(N)$ terms- the number of sites in the system. In the last inequality above, $c'$ is a constant, and $a$ and $v_{LR}$ denote the Lieb-Robinson decay constant (inverse correlation length) and speed respectively of Eq. (B1) [27]. Rather explicitly,

$$|Tr(\tilde{\rho}[\mathcal{H}_i^H(t), \mathcal{H}_{j'}^H(t)])| \leq ||[\mathcal{H}_i^H(t), \mathcal{H}_{j'}^H(t)]||. \qquad (B6)$$

In order to derive Eq. (B5), we note that the Lieb-Robinson bounds of Eq. (B1) [27], $||[\mathcal{H}_i^H(t), \mathcal{H}_{j'}^H(t)]|| \leq c'e^{-a(|i-j'|-v_{LR}t)}$, imply Eq. (B5). For each $i \in \mathcal{S}$, there is a minimum distance $D(i)$ between $i$ and the surrounding region where $H_{S-\mathcal{E}} + H_{\mathcal{E}}$ has its support. For any such $i$, we may bound (from above) the sum over all $j'$ of the exponential $e^{-a(|i-j'|-v_{LR}t)}$ by a sum of this exponential over the larger domain external to a sphere of radius $D(i)$ around $i$ (such a volume contains $\mathcal{E}$ as a subset). For sufficiently short times $t$, in Eq. (B5), the sum $\frac{c'}{N} \sum_{i,j'} e^{-a(|i-j'|-v_{LR}t)}$ tends to zero for macroscopic systems (since the minimal distance $D(i)$ of a typical $i \in S$ to its surrounding environment is of the order of the system length); for vanishingly small times, the latter sum of $e^{-a(|i-j'|-v_{LR}t)}$ over such a larger domain of $j'$ values with $|i - j'| \geq D(i)$ decays exponentially in $D(i)$. Specifically, in $d$ spatial dimensions, $\frac{c'}{N} \sum_{i,j'} e^{-a(|i-j'|-v_{LR}t)}$ scales as $O(D^d e^{-aD})$ for large $D$. Putting all of the pieces together, we see that the Lieb-Robinson bounds imply that at vanishingly short times, $\frac{1}{N} |\sum_i Tr(\tilde{\rho}[\mathcal{H}_i^H(t), \tilde{H}])|$ is bounded from above by a function that is exponentially small in the length of the system. In other words, under the above specified locality conditions, the energy density of a macroscopic system cannot change at a finite rate at sufficiently short times. A corollary of these inequalities is that in a local theory in which the the Lieb-Robinson bounds hold, a transient time Hamiltonian describing the effects of the environment cannot change instantaneously in such a way as to give rise to a finite change in the energy density of the system. Thus, generally, the environment may not truly instantaneously couple to (nor decouple from) a finite fraction of a macroscopic system (in the form of an effective instantaneously varying Hamiltonian $H(t')$ (as in Section VI) when procedure (1) of Section II is invoked). The influences of the environment (and variations in any Hamiltonian that emulate the effects of the environment) are limited those associated with "light cone" distances of size ($v_{LR}t$).

The above calculations may be replicated, nearly verbatim, for operators associated with other intensive quantities $q$ different from $\frac{H}{N}$ of Eq. (B2).

## Appendix C: Relating Heisenberg's equation of motion to correlations: A finite rate of change of the system energy density necessitates system spanning correlations between the environment and the system

In what follows, we explicitly demonstrate that (when all interactions (i.e., $\tilde{H}$) are time independent):

• If the energy density of the system changes at a finite rate then there must be *system length spanning correlations between the external environment and the system itself.*

A formal proof of this assertion is rather straightforward. Using the notation of Appendix B and the main text, by the Heisenberg equations of motion,

$$0 < \left|\frac{d\epsilon}{dt}\right| = \frac{1}{N\hbar}\left|Tr(\tilde{\rho}[\tilde{H}, H^H])\right| = \frac{1}{N\hbar}\left|Tr(\tilde{\rho}[\delta\tilde{H}, \delta H^H])\right|$$

$$= \frac{1}{N\hbar}\left|\sum_i Tr(\tilde{\rho}[\delta\mathcal{H}_i^H(t), \delta H_{\mathcal{S}-\mathcal{E}}^H(t) + \delta H_{\mathcal{E}}^H(t)])\right|. \quad \text{(C1)}$$

For any operator $Q$ appearing in the last line of Eq. (C1), we define $\delta Q \equiv (Q - \langle Q \rangle) \equiv (Q - Tr(\tilde{\rho}Q))$. Apart from these trivial shifts by $(-\langle Q \rangle)$, Eq. (C1) and its derivation are identical to those of Eq. (B4). For all operators $\hat{\mathcal{A}}$ and $\hat{\mathcal{B}}$,

$$\left|Tr(\tilde{\rho}[\hat{\mathcal{A}}, \hat{\mathcal{B}}])\right|$$
$$\leq 2 \times \max\left\{\left|Tr(\tilde{\rho}(\hat{\mathcal{A}}\hat{\mathcal{B}}))\right|, \left|Tr(\tilde{\rho}(\hat{\mathcal{B}}\hat{\mathcal{A}}))\right|\right\}. \quad \text{(C2)}$$

Thus, Eq. (C1) implies that

$$0 < \frac{2}{N}\sum_i \max\left\{\left|Tr\left(\tilde{\rho}(\delta\mathcal{H}_i^H(t)(\delta H_{\mathcal{S}-\mathcal{E}}^H(t) + \delta H_{\mathcal{E}}^H(t)))\right)\right|,\right.$$
$$\left.\left|Tr\left(\tilde{\rho}(\delta H_{\mathcal{S}-\mathcal{E}}^H(t) + \delta H_{\mathcal{E}}^H(t))\delta\mathcal{H}_i^H(t))\right)\right|\right\}. \quad \text{(C3)}$$

Since the number $(N')$ of system sites $i$ associated with the bounded local operators $\mathcal{H}_i^H$ (Eq. (1)) is $N' = \mathcal{O}(N)$, from Eq. (C3), we see that the average correlator between the local $\delta\mathcal{H}_i^H$ (that, apart from a set of vanishing measure, all lie in the system bulk at a distance $D = \mathcal{O}(L)$ from the surrounding environment) and the fluctuations $(\delta H_{\mathcal{S}-\mathcal{E}}^H(t) + \delta H_{\mathcal{E}}^H(t))$ must be finite. In other words (as is expected), the correlator between the bulk and the Hamiltonian coupling it to the surrounding environment is of order unity. The above holds irrespective of how large $N$ may be so long as $\left(\frac{d\epsilon}{dt}\right)$ is non-vanishing. We next consider what occurs when, similar to Appendix B, we invoke Eq. (B3) and express the Hamiltonian of the environment and its coupling to the system as a sum of local terms ($\{\mathcal{H}_j\}$ with $j \notin \mathcal{S}$). In such a case, Eq. (C3) will imply that if there is an exponential decay length $\xi$ associated with the larger of the two correlators $G_{ij'}(t) \equiv \langle\delta\mathcal{H}_i(t)\delta\mathcal{H}_{j'}(t)\rangle \equiv Tr(\tilde{\rho}(\delta\mathcal{H}_i(t)\delta\mathcal{H}_{j'}(t)))$

and $G_{j'i}(t) \equiv \langle\delta\mathcal{H}_{j'}(t)\delta\mathcal{H}_i(t)\rangle \equiv Tr(\tilde{\rho}(\delta\mathcal{H}_{j'}(t)\delta\mathcal{H}_i)(t))$ then $\xi \gtrsim \mathcal{O}(L)$. Similarly, if the correlator decays algebraically, $|G_{ij'}| \sim |i - j'|^{-p}$, then Eq. (C3) implies that a finite rate of change of the energy density for large systems sizes $L$ only if $p < d$ with $d$ the spatial dimensionality of the system and the environment. It is noteworthy that the commutator of Eq. (C1) has (when evaluated with $\tilde{\rho}$) an imaginary expectation value for the Hermitian Hamiltonian operators. For semiclassical systems, the real component of the correlator $G_{ij'}$ is, typically, far larger than its imaginary part (which we bounded in the above). Stated equivalently, the expectation value of the anticommutator $\{\delta\mathcal{H}_i(t), \delta\mathcal{H}_{j'}(t)\}$ is, in semiclassical systems, normally far larger than the expectation value of the commutator $[\delta\mathcal{H}_i(t), \delta\mathcal{H}_{j'}(t)]$.

## Appendix D: Conditional probability arguments for long range correlations

As we explained in Appendix C, a driven system (one in which the intensive quantities change at a finite rate) must exhibit long range correlations between observables ($\mathcal{H}_i$) at sites $i$ in the bulk to the environment ($\mathcal{E}$). We now apply "classical" probability arguments to demonstrate that when these long range correlations between different sites in the system and its environment are present, then the local Hamiltonian terms $\mathcal{H}_i$ at different sites in the system bulk may exhibit long range correlations. Towards this end, we write the classical joint probability distribution $P(E_{\mathcal{E}}, E_i, E_j)$ associated with the values $(E_{i,j})$ of the energies $\mathcal{H}_i$ and $\mathcal{H}_j$ at the two sites $i, j$ in the bulk (in the system $\mathcal{S}$) and the energy $(H_{\mathcal{S}-\mathcal{E}}^H(t) + H_{\mathcal{E}}^H(t))$ affiliated with the environment $\mathcal{E}$ (denoted by $E_{\mathcal{E}}$). In the context of Appendix C, the joint probability distribution

$$P(E_{\mathcal{E}}, E_i, E_j) \equiv Tr\left[\tilde{\rho}\ \delta(H_{\mathcal{S}-\mathcal{E}}^H(t) + H_{\mathcal{E}}^H(t) - E_{\mathcal{E}}(t))\right.$$
$$\left.\times\delta(\mathcal{H}_i - E_i(t))\ \delta(\mathcal{H}_j - E_j(t))\right]. \quad \text{(D1)}$$

Other joint probabilities are defined similarly. By the chain rule of conditional probabilities,

$$P(E_{\mathcal{E}}, E_i, E_j) = P(E_i|E_{\mathcal{E}}, E_j)P(E_{\mathcal{E}}|E_j)P(E_j). \text{(D2)}$$

Here, $P(E_i|E_{\mathcal{E}}, E_j) = \frac{P(E_i, E_{\mathcal{E}}, E_j)}{P(E_{\mathcal{E}}, E_j)}$ is the conditional probability of measuring a local energy (with a local "thermometer") of value $E_i$ given a value of the local energy ($E_j$) at site $j$ and the above defined energy $E_{\mathcal{E}}$ associated with the environment $\mathcal{E}$. Now, if $i$ is independent of $j$ then

$$P(E_i|E_{\mathcal{E}}, E_j) = P(E_i|E_{\mathcal{E}}). \quad \text{(D3)}$$

Subsequently, Eq. (D2) reduces to

$$P(E_{\mathcal{E}}, E_i, E_j) = P(E_i|E_{\mathcal{E}})P(E_{\mathcal{E}}|E_j)P(E_j). \quad \text{(D4)}$$

The classical joint probability $P(E_i, E_j)$ then reads

$$P(E_i, E_j) = \sum_{E_\mathcal{E}} P(E_\mathcal{E}, E_i, E_j)$$
$$= \sum_{E_\mathcal{E}} P(E_i | E_\mathcal{E}) P(E_\mathcal{E} | E_j) P(E_j). \quad (D5)$$

This, in turn, implies that the conditional probability between the values of $E_i$ and $E_j$ at the two sites in the system bulk is given by

$$P(E_i | E_j) = \sum_{E_\mathcal{E}} P(E_i | E_\mathcal{E}) P(E_\mathcal{E} | E_j)$$
$$= \frac{\sum_{E_\mathcal{E}} P(E_i | E_\mathcal{E}) P(E_j | E_\mathcal{E}) P(E_\mathcal{E})}{\sum_{E_\mathcal{E}} P(E_j | E_\mathcal{E}) P(E_\mathcal{E})}. \quad (D6)$$

In the second (alternate form) line of Eq. (D6), we invoked Bayes' theorem. Appendix C demonstrated that in a (quantum) system in which the energy density varies at a finite rate, there are nontrivial correlations between the energy fluctuations in $i$ and $\mathcal{E}$ (i.e., these fluctuations are not independent of one another). Similarly, the energy fluctuations in $j$ and in $\mathcal{E}$ are correlated and not independent of one another. Thus, in general, the conditional probabilities

$$P(E_i | E_\mathcal{E}) \neq P(E_i) \text{ and } P(E_\mathcal{E} | E_j) \neq P(E_\mathcal{E}). \quad (D7)$$

(Analogously, for the conditional probabilities appearing in the second line of Eq. (D6), a coupling between the driving environment and the bulk implies (as formalized in Appendix C) that $P(E_j | E_\mathcal{E}) \neq P(E_j)$.) These inequalities are expected to generally hold for both quantum as well as classical systems since, at their core, these relations indeed reflect the bulk coupling between the environment driving the system and the system itself necessary to induce a finite rate of change of the energy density. (See also the discussion in Appendix C concerning semiclassical systems.) When the inequalities of Eq. (D7) are substituted in Eq. (D6), we will generally have

$$P(E_i | E_j) \neq P(E_i). \quad (D8)$$

That is, the local energy fluctuations at (arbitrarily far separated) sites $i$ and $j$ in the system bulk are *not independent* of one another as we assumed in deriving Eq. (D5). Thus, there are non-trivial correlation between any sites $i$ and $j$ in the driven system $\mathcal{S}$ (even if the spatial separation $|i - j|$ is large). With reference to Eq. (3), we now see that (even for large $|i - j|$) the covariance

$$G_{ij} = \sum_{E_i, E_j} \Big( P(E_i | E_j) - P(E_i) \Big) P(E_j) E_i E_j, \quad (D9)$$

need not vanish (and may be of order unity). If the coupling to the environment is the dominant contribution to the correlations $G_{ij}$ when $|i - j|$ is large then when the coupling between the environment and different sites $i$ in the bulk is (nearly) constant, then all connected pair correlators $G_{ij}$ appearing in Eq. (3) will be of (almost) uniform magnitude (and sign). Under these conditions, $\sigma_\epsilon = \mathcal{O}(1)$.

## Appendix E: Entangled Ising chain eigenstate expectation values produce thermal averages

In order to explicitly illustrate how macroscopic entanglement may naturally appear in typical thermal states (even those of closed systems that have no explicit contact with an external bath), we turn to a simple example-that of the uniform coupling one dimensional Ising model (the Hamiltonian $H_I$ of Section V on an open chain with uniform nearest neighbor coupling- $J_{ij} = J$). In these appendices, we will dispense with factors of $\hbar/2$ and use the conventional definition of the Ising model Hamiltonian with the spin at any site $r$ being $S_r^z = \pm 1$ (i.e., the diagonal elements of the Pauli matrix $\sigma_r^z$). In each Ising state product state, the value of $\langle S_r^z S_{r'}^z \rangle$ is either 1 or (-1). This single Ising product state expectation value differs from that of the equilibrium system at finite temperatures. It is only if we compute the expectation value within a state formed by a superposition of many such product states (i.e., an expectation value within such a highly entangled state) or if we average under uniform translations of the origin (i.e., entangle with equal weights all states related by translation) that we will obtain the equilibrium result. The Ising operators $S_i^z$ are diagonal in the product basis; different product states are orthogonal to each other. In a superposition of different product states, only the diagonal (i.e., weighted Ising product expectation values) terms are of importance when computing $\langle S_r^z S_{r'}^z \rangle$.

We consider a highly entangled eigenstate $|\Psi\rangle$ of the one-dimensional Ising model. Such an entangled state emulates, in real space, entangled eigenstates $|v_\alpha; S_{tot}, S_{tot}^z\rangle$ with (for systems in their thermodynamic limit) $|S_z^{tot}/S_{\max}| < 1$ (i.e., not product states of all spins maximally polarized up or down along the field direction) of the spin models discussed in Section VI. For an Ising model $H_I$ on a one dimensional chain of length $L$, given an eigenstate of energy $E$, the frequency of low energy nearest neighbor bonds (namely, $S_r^z = S_{r'}^z = \pm 1$ ("↑↑ "or "↓↓")) is $p$ and that of having higher energy bonds (i.e., "↑↓" or "↓↑") is $q$. Clearly, $p + q = 1$ and $(q - p) = E/(LJ)$ where $J$ is the Ising model exchange constant and $E$ is the total energy. In the one dimensional Ising model there is no constraint on the nearest neighbor bonds $S_i^z S_{i+1}^z$ (these products are all independent variables that are "+1" or "-1" that sum to the scaled total energy $E/J$). Consider a spin at site $r$ which is, say, "↑". We may now ask what is the average value of a spin at another site $r'$. Evidently, if there is an even number of domain walls (or even number of energetic bonds) between sties $r$ and $r'$ then the spin at site $r'$ is "↑" while if there is an odd number of domain walls between the two sites then the spin at site $r'$ is "↓". The average $\langle S_r^z S_{r'}^z \rangle = (p - q)^{|r - r'|}$. That is, if we have an even number of bad domain walls (corresponding to n even power of $q$) then the contribution to the correlation function will be positive while if we have an odd number of domain walls (odd power of $q$) then the contribution to the correlation function will be negative. The pref-

actors in the binomial expansion of $(p-q)^{|r-r'|}$ account for all of the ways in which domain walls may be placed in the interval $(r, r')$. However, $(p-q) = (-E)/(LJ)$. Thus, the correlator $\langle S_r^z S_{r'}^z \rangle = [(-E)/(LJ)]^{|r-r'|}$. This single eigenstate result using the binomial theorem indeed matches with the known results for correlations in the Ising chain in the canonical ensemble at an inverse temperature $\beta = \frac{1}{k_B T}$ where $E = -J(L-1)\tanh \beta J$ and $\langle S_r^z S_{r'}^z \rangle = (\tanh \beta J)^{|r-r'|}$. The agreement of the spatially long distance correlator result in one eigenstate with the prediction of the fixed energy microcanonical ensemble is obvious. The above probabilistic derivation for general sites $r$ and $r'$ will hold so long as the eigenstate $|\psi\rangle$ is a sum of numerous Ising product states (all having the same energy or, equivalently, the same number of domain walls). If this result holds for all site pairs $(r, r')$ then the entanglement entropy is expected to scale monotonically in the size (or "volume") of this one dimensional system. Indeed, a rather simple calculation (outlined in Appendix F) illustrates that if the $L$ site system is partitioned into subregions $A$ and $B$ of "volumes" $L_A$ and $L_B$ (with $L = L_A + L_B$) then if, e.g., $|\Psi\rangle$ is an equal amplitude superposition $|\Psi_+\rangle$ of all Ising product states (i.e., an equal amplitude superposition of the product states $|s_1 s_2 \cdots s_N\rangle$ of Section V) that all have a given fixed energy then the entanglement entropy between regions $A$ and $B$ scales as $\min\{\ln L_A, \ln L_B\}$.

Broader than the specific example of this Appendix, the coincidence between the single (entangled) eigenstate expectation values with the equilibrium ensemble averages is expected to hold for general classical systems in arbitrary dimensions. To see why this is so consider the expectation value of a general observable (including any correlation functions) that is diagonal in the basis of degenerate classical product states. When computed in a state formed by a uniform modulus superposition of degenerate states (e.g., the equal amplitude sum of all local product states of the same energy), the expectation value of such an observable may naturally emulate the microcanonical ensemble average of this observable over all classical states of the same energy. Finite energy density states (i.e., states whose energy density is larger than that of the ground state) formed by a uniform amplitude superposition of all product states generally exhibit macroscopic entanglement. As we have elaborated on in this Appendix, this anticipation is realized for the classical Ising chain. For the classical Ising chains discussed above, the below two general quantities are the same for a general observable $\mathcal{O}$: (i) the mean of the expectation values of $\mathcal{O}$ in all local product states that are superposed to form general (*not necessarily* an exact uniform modulus superposition of degenerate states) highly entangled states and (ii) the average of $\mathcal{O}$ as computed by a classical microcanonical ensemble calculation. As we emphasized earlier, general thermal states may exhibit "volume" law entanglement entropies [64]. However, not all eigenstates that display the equilibrium value of the correlators $\langle S_r^z S_{r'}^z \rangle$ need to exhibit volume law entangle-

ment. As alluded to above, in the next Appendix, we will compute the entanglement entropy associated with $|\Psi_+\rangle$ and show that it is macroscopic even in one dimensional systems albeit being logarithmic in the "volume".

## Appendix F: Entanglement entropies of a uniform amplitude superposition of classical product states

We next discuss the reduced density matrices and entanglement entropies associated with (1) the eigenstates $|\phi_\alpha\rangle = |v_\alpha; S_{tot}, S_{tot}^z\rangle$ of Section VI when $S_{tot}$ happens to be maximal ($S_{tot} = S_{\max}$), (2) the symmetric quantum states described of Appendix E, and a generalization thereof that we now describe. Specifically, we will consider general Hamiltonians that may be expressed as a sum of decoupled commuting local terms, $H = \sum_{i=1}^{L} \mathcal{H}_i$ (i.e., $N' = L$ in the notation of the Introduction) on a Hilbert space endowed with a simple local tensor product structure. We denote the eigenstates (of energies $\varepsilon_{n_i}$) of each of the local operators $\mathcal{H}_i$ by $\{|\nu_i^{n_i}\rangle\}$. For such systems, any product state $|c\rangle = |\nu_1^{(n_1)}\rangle \otimes |\nu_2^{(n_2)}\rangle \otimes \cdots \otimes |\nu_L^{(n_L)}\rangle$ is, trivially, a eigenstate of $H$ (of total energy $E_c = \sum_{i=1}^{L} \varepsilon_{n_i}$). Formally, one may think of $\mathcal{H}_i$ as decoupled independent commuting "quasi-particle" operators (i.e., colloquially, $H$ describes "an ideal gas" of such quasi-particles). We now explicitly write the states that are equal amplitude superpositions of all such product states $|c\rangle$ of a given total energy,

$$|\Psi_+\rangle \equiv \frac{1}{\sqrt{\mathcal{N}(E)}} \sum_{E_c = E} |c\rangle, \qquad (F1)$$

Similar to the discussion of Appendix E, for observables $\mathcal{O}_d$ that are diagonal in the $\{|c\rangle\}$ basis, the single eigenstate expectation values $\langle \Psi_+ | \mathcal{O}_d | \Psi_+ \rangle$ are equal to the microcaonical equilibrium averages of $\langle \mathcal{O}_d \rangle_{eq;\mathsf{mc}}$ in which the energy $E$ is held fixed. In Eq. (F1), $\mathcal{N}(E) = e^{\mathsf{S}(E)/k_B}$ is the number of product states $|c\rangle$ that have a total energy $E$ (and $\mathsf{S}(E)$ is the associated Boltzmann entropy). The states of Eq. (F1) describe those of the Ising spin states alluded to in Appendix E. Such states rear their head also in other arenas. For instance, since, in a many body spin system, the state of maximal total spin $S_{tot} = S_{\max}$ is a uniform amplitude superposition of all product states having a given value of $S_z^{tot}$ (i.e., a uniform amplitude superposition of all states of decoupled spins in a uniform magnetic field that share the same energy), states of the type $|\Psi_+\rangle$ include the eigenstates that we analyzed in Section (VI) (when these states are those of maximal total spin). The entanglement entropy that we will compute for $|\Psi_+\rangle$ will thus have implications for these and other systems. We partition the $L$ site system into two disjoint regions $A$ and $B$ and examine the entanglement between these two subvolumes. To facilitate the

calculation, we will employ the symmetric combinations

$$|E_A\rangle_+ \equiv \frac{1}{\sqrt{\mathcal{N}_A(E_A)}} \sum_{E(\{c_A\})=E_A} |\{c_A\}\rangle,$$

$$|E_B\rangle_+ \equiv \frac{1}{\sqrt{\mathcal{N}_B(E_B)}} \sum_{E(\{c_B\})=E_B} |\{c_B\}\rangle. \quad \text{(F2)}$$

In the first of Eqs. (F2), the sum is over all product states $\{|c_A\rangle\}$ having their support on the sites $1 \le i \le L_A$ that are of fixed energy $E_A$. Similarly, the symmetric state $|E_B\rangle_+$ extends over the sites $L_A + 1 \le i \le L$. With these definitions, we rewrite Eq. (F1) as

$$|\Psi_+\rangle = \sum_{E_A} \sqrt{\frac{\mathcal{N}_A(E_A)\mathcal{N}_B(E-E_A)}{\mathcal{N}(E)}}$$
$$\times |E_A\rangle_+ |E_B = E - E_A\rangle_+. \quad \text{(F3)}$$

The density matrix associated with this state is $\rho_+ \equiv |\Psi_+\rangle\langle\Psi_+|$. To compute the entanglement entropy, we next write the reduced density matrix

$$\rho_{B,+} \equiv Tr_A\rho_+ = \frac{1}{\mathcal{N}(E)} \sum_{E_A} (\mathcal{N}_A(E_A)\mathcal{N}_B(E_B = E - E_A)$$

$$\times |E_B = E - E_A\rangle_+\langle E_B = E - E_A|_+). \quad \text{(F4)}$$

If a given system is partitioned into two non-interacting subsystems $A$ and $B$ then the sole relation linking the two subsystems will be the constraint of total energy $E = E_A + E_B$. Of all possible ways of partitioning the total energy $E = E_A + E_B$, one pair of energies $\overline{E}_A$ and $\overline{E}_B$ will yield the highest value of $S_A(\overline{E}_A) + S_B(\overline{E}_B)$. The ratios appearing in Eq. (F4),

$$\frac{\mathcal{N}_A(E_A)\mathcal{N}_B(E-E_A)}{\mathcal{N}(E)}$$
$$= e^{(S_A(E_A)+S_B(E-E_A)-S(E))/k_B}, \quad \text{(F5)}$$

follow, upon Taylor expanding the ratio to quadratic order about its maximum at $\overline{E}_A$ and $\overline{E}_B = E - \overline{E}_A$, a Gaussian distribution with a standard deviation set by

$$\sigma_B = \sqrt{k_B T^2 C_v^{eff}(T)}. \quad \text{(F6)}$$

In Eq. (F6),

$$C_v^{eff}(T) \equiv \frac{C_v^{(A)}(T)C_v^{(B)}(T)}{C_v^{(A)}(T) + C_v^{(B)}(T)}. \quad \text{(F7)}$$

The latter Taylor expansion may be carried out for energy densities associated with finite temperatures. (In the vicinities of either the ground state value of the energy density or the highest energy density, the derivatives of the entropy relative to the energy diverge and the Taylor expansion becomes void.) The entropies $S_A(E_A)$ and $S_B(E_B)$ appearing in Eq (F5) are those of subsystems $A$ and $B$ that, as emphasized above for this system of non-interacting particles, are merely constrained by the

condition that $E_A + E_B = E$. For this non-interacting system,

$$e^{S(E)/k_B} = \sum_{E_A} e^{S_A(E_A)/k_B} e^{S_B(E_B=E-E_A)/k_B}, \quad \text{(F8)}$$

and thus, trivially, $S(E) \ge S_A(\overline{E}_A) + S_B(\overline{E}_B)$. As throughout the current work, in Eqs. (F6, F7), $T$ denotes the temperature (set by the condition that the canonical ensemble equilibrium internal energy $Tr(He^{-\beta H})/Tr(e^{-\beta H})$ is equal to the total energy $E$). The entropy of the Gaussian distribution scales as the logarithm of its width. Specifically, for the saddle point Gaussian approximation of Eqs. (F5,F6,F7),

$$S_{ent,+} \equiv -Tr(\rho_{B,+} \ln \rho_{B,+}) = \frac{1}{2}\ln(2\pi\sigma_B^2 + 1)$$
$$\sim \ln \sigma_B, \quad \text{(F9)}$$

where in the last line, we made manifest the assumed extensive $L_{A,B} \gg 1$ (and thus $\sigma_B \gg 1$). If $S_A(\overline{E}_A) = \mathcal{O}(L_A)$ and $S_B(\overline{E}_B) = \mathcal{O}(L_B)$ when $L_{A,B} \gg 1$ then, from Eqs. (F6, F7, F9), the entanglement entropy for states of finite temperature (i.e., states exhibiting a finite energy density above that of the ground state value),

$$S_{ent,+} = \mathcal{O}(\min\{\ln L_A, \ln L_B\}). \quad \text{(F10)}$$

We reiterate that generic states of fixed total energy will exhibit an entanglement entropy proportional to the system volume (see, e.g., the considerations of [64]). Even though a system of non-interacting particles is trivial and its properties may, generally, be exactly computed, its entanglement entropy may be macroscopic. Beyond systems of non-intreating particles, more profound counterparts to this well known maxim are realized in bona fide interacting one-dimensional quantum systems in which the entropy associated with thermofield double states does not scale with the system indicating that the system may be efficiently represented via a matrix-product representation [12] while, as we highlighted, extensive entanglement entropies may appear. We next discuss two specific realizations of Eqs. (F9, F10).

### 1. Maximal total spin eigenstates

As we noted above, for any fixed $S_{tot}^z$, the eigenstates of Eq. (8), $|\Psi_+\rangle$ corresponds to a maximal total spin ($S_{tot} = S_{\max}$) state of the $L = N$ spins (with the given value of $S_{tot}^z$). In order to relate this to our general results of Eqs. (F9, F10) for the entanglement entropy, we consider the local Hamiltonians $\mathcal{H}_i$ forming the Hamiltonian $H = \sum_{i=1}^{N} \mathcal{H}_i$ to be given by $\mathcal{H}_i = -B_z S_i^z$. With this, $|\Psi_+\rangle$ of Eq. (F1) is an eigenstate of $S_{tot}^z$ (with each product state $|c\rangle$ being an eigenstate of all $\{S_i^z\}$ operators). We consider what occurs if the $N$ spins are partitioned into the two groups $A$ and $B$ of approximately equal numbers $L_A$ and $L_B$, and $|w| \equiv |S_{tot}^z/(\hbar S_{tot})| < 1$. In this case, the saddle point approximation of Eq. (F5)

yields, as before, a Gaussian distribution and, a consequent logarithmic entanglement entropy logarithmic,

$$\mathsf{S}_{ent,+} = \mathcal{O}(\ln N). \qquad (\text{F11})$$

Thus, as highlighted in Section VI, initial states $|\psi^0_{Spin}\rangle$ of maximal total spin and $|w| < 1$ feature logarithmic in volume entanglement entropies.

### 2. Ising chains

Returning to the considerations of Appendix E and the notation introduced in Section V, we now consider the symmetric sum of all Ising product states that share the same energy (as measured by the Ising Hamiltonian $H_I$ of Section V). As in Section F 1, we can transform the problem of computing the entanglement entropy of such symmetric states $|\Psi_+\rangle$ into that involving eigenstates of decoupled local Hamiltonians $\mathcal{H}_i$ that led to Eqs. (F10). Towards this end, we focus on the nearest neighbor spin products that were crucial to our analysis in Appendix E, and define the operators

$$1 \le i \le L-1: \quad R_i \equiv S^z_i S^z_{i+1},$$
$$R_L \equiv S^z_L. \qquad (\text{F12})$$

The Ising Hamiltonian now explicitly becomes a sum of the above defined decoupled commuting operators, $H_I = -J \sum_{i=1}^{L-1} R_i$. Using the vocabulary that we employed earlier, the "quasi-particle" operators $\{R_i\}_{i=1}^{L:-1}$ are associated with the existence ($R_i = -1$) or absence ($R_i = 1$) of domain walls between neighboring Ising spins. On the two subregions $A$ and $B$, we define $H_{AI} = -J \sum_{i=1}^{L_A} R_i$ and $H_{BI} = -J \sum_{i=L_A+1}^{L-1} R_i$. The equal amplitude superposition of all Ising product states of fixed energy can now be rewritten as

$$|\Psi_{I+}\rangle = \frac{1}{2^{L/2}} \sum_{r_1, r_2, \cdots, r_L} |r_1 r_2 \cdots r_L\rangle, \qquad (\text{F13})$$

where $r_i = \pm 1$ denote the eigenvalues of $R_i$. When evaluating the reduced density matrix $\rho_{BI+} = Tr_A |\Psi_{I+}\rangle\langle\Psi_{I+}|$, the trace over all Ising spins $\{s_{i \le L_A}\}$ that lie in the spatial region $A$ is replaced by that over $\{r_{i \le L_A}\}$. Repeating the earlier calculations we find, once again, the entanglement entropy of Eqs. (F6,F7,F9) [181]. Equating the internal energy of a system given by $H_I$ to $E$ we see that, when $L \gg 1$, the temperature appearing in Eqs. (F6,F7,F9) is given by

$$\frac{1}{k_B T} = -\tanh^{-1}\left(\frac{E}{LJ}\right). \qquad (\text{F14})$$

In Eq. (F7), the heat capacities of the Ising chain subsystems $A$ and $B$ (when $L_{A,B} \gg 1$) are

$$C_v^{(A,B)}(T) = k_B L_{A,B}\left((\beta J)^2 - \left(\frac{\beta E_{A,B}}{L_{A,B}}\right)^2\right). \qquad (\text{F15})$$

Eq. (F10) provides the asymptotic scale of the entanglement entropy; similar to Eq. (F11), if $L_A$ and $L_B$ are both of order of the system size, $L_{A,B} = \mathcal{O}(N)$ then the entanglement entropy $\mathsf{S}_{ent;+}$ of the symmetric state will scale logarithmically in $N$. General eigenstates may exhibit larger entanglement entropies (see Appendix K).

### Appendix G: The total spin of large systems

We now discuss the total spin sectors that may appear in the spin model of Section VI A. Our aim is to highlight both statistical and physical aspects of the total spin and its scaling with the system size $N$.

All states with maximal total spin and definite eigenvalues of the total $S^z_{tot}$ operator are eigenstates of the general Hamiltonian $H_{spin}$ of Eq. (8). (These eigenstates span the basis of all ferromagnetic spin states with spins uniformly polarized along different directions.) This assertion may be explicitly proven by the following simple observations: (i) For any two spin $S = 1/2$ operators, the scalar product $\vec{S}_i \cdot \vec{S}_j = \hbar^2(\frac{1}{2}P_{ij} - \frac{1}{4})$ where $P_{ij}$ is the operator permuting the two spins, (ii) Any state of maximum total spin ($S_{tot} = S_{\max} = N\hbar/2$) is a symmetric state that is invariant under all permutations $\{P_{ij}\}$. From properties (i) and (ii), it follows that any state $|S_{tot} = S_{\max} = N\hbar/2, S^z_{tot}\rangle$ is an eigenstate of both the first and second terms of Eq. (8) and therefore of the full Hamiltonian $H_{spin}$. Thus, any state of maximal total spin $S_{tot} = S_{\max}$ that is an eigenstate of $S^z_{tot}$ is automatically an eigenstate of $H_{spin}$ of Eq. (8). In general, when $S_{tot} < S_{\max}$, only some linear combinations of the multiple states of given values of $S_{tot}$ and $S^z_{tot}$ are eigenstates of $H_{spin}$ (hence the appearance of additional quantum numbers $v_\alpha$ defining general eigenstates $|\phi_\alpha\rangle$). To make this clear, we can explicitly write down the total spin for a system of $N$ spin $S = 1/2$ particles. That is,

$$
\begin{aligned}
N = 2: \quad & \frac{1}{2} \otimes \frac{1}{2} = 1 \oplus 0, \\
N = 3: \quad & \frac{1}{2} \otimes \frac{1}{2} \otimes \frac{1}{2} = \frac{3}{2} \oplus \frac{1}{2} \oplus \frac{1}{2}, \\
N = 4: \quad & \frac{1}{2} \otimes \frac{1}{2} \otimes \frac{1}{2} \otimes \frac{1}{2} = 2 \oplus 1 \oplus 1 \oplus 1 \oplus 0 \oplus 0, \\
& \cdots \quad .
\end{aligned}
\qquad (\text{G1})
$$

The first (textbook type) equality of Eq. (G1) states that singlet ($S = 0$) and triplet ($S = 1$) total spin combinations may be formed by adding two ($N = 2$) spins of size $S = 1/2$. Other well known relations are similarly tabulated for higher $N$. Since $H_{spin}$ is defined on a $(2S+1)^N$ dimensional Hilbert space, its eigenstates span all states in the direct product basis on the lefthand side of Eq. (G1). For each $N$, the sector of maximal spin ($S_{tot} = S_{\max} = NS$) is unique. However, for $N > 2$, all other total spin sectors in Eq. (G1) exhibit a multiplicity $\mathcal{M}_{S_{tot}}$ larger than one. While it is, of course, possible to simultaneously diagonalize the Hamiltonian of Eq. (8) with the two operators $S_{tot}$ and $S^z_{tot}$, there are multiple

states that share the same eigenvalues of $S_{tot}$ and $S_{tot}^z$. One needs, of course, to diagonalize $H_{spin}$ in every subspace of given $S_{tot}$ and $S_{tot}^z$ in order to find its eigenstates in each such subspace. This task is not necessary for the calculations of Section VI A. Rather, the total spin is the quantity of relevance. Using the characters $\overline{\chi}_{S_{tot}}$ of spin $S_{tot}$ representations of SU(2), we find that there are

$$\mathcal{M}_{S_{tot}}^N = \frac{N!(2S_{tot}+1)}{(\frac{N}{2}+S_{tot}+1)!(\frac{N}{2}-S_{tot})!} \qquad \text{(G2)}$$

sectors of total spin $0 \le S_{tot} \le \frac{N}{2}$ on the righthand side of Eq. (G1). The decomposition into characters of $SU(2)$ has a transparent physical content. Consider a global rotation by of all spins an arbitrary angle $\theta'$ about the $z$ axis. The trace of the operator that implements this rotation is the same into the different basis appearing in Eq. (G1): (1) the product basis (the lefthand side of Eq. (G1)) of $N$ spins of size $S = 1/2$ and (2) the basis comprised of the total spin sectors (the righthand side of Eq. (G1)). When expressing the basis invariant trace of the arbitrary rotation evolution operator in terms of the Laurent series in $e^{i\theta'/2}$ that arises when taking the trace of the rotation operator, the series must identically match in both of these bases of Eq. (G1). Equating the trace as evaluated in (1) and (2) as discussed above, we explicitly have $(2\cos\frac{\theta'}{2})^N = \sum_{S_{tot}} \mathcal{M}_{S_{tot}}^N \overline{\chi}_{S_{tot}}$ with $\overline{\chi}_{S_{tot}} = \sum_{\mathsf{S}=-S_{tot}}^{S_{tot}} e^{i\mathsf{S}\theta'} = \frac{\sin(2S_{tot}+1)\frac{\theta'}{2}}{\sin\frac{\theta'}{2}}$ from which Eq. (G2) follows by Fourier transformation. Perusing Eq. (G2), we see that for large $N$, the highest values of $\mathcal{M}_{S_{tot}}^N$ occur for small $S_{tot}$; in Eq. (G1), a "randomly" ("infinite temperature") chosen state of $N \gg 1$ spins is most likely to have $S_{tot} \le \mathcal{O}(\sqrt{N})$. Specifically, if we approximate, for fixed $N \gg S_{tot} \gg 1$, the distribution of binomial coefficients in Eq. (G2) by a Gaussian, we trivially obtain

$$\mathcal{M}_{S_{tot}}^N \sim \frac{2^{N+\frac{5}{2}} e^{-\frac{2S_{tot}^2}{N}}}{N^{\frac{3}{2}}\sqrt{\pi}} S_{tot}. \qquad \text{(G3)}$$

The binomial character of Eqs. (G2) with the associated asymptotic Gaussian form of Eq. (G3) is not unexpected: a summation of $N \gg 1$ random classical spins (when these are viewed as uniform length displacement vectors) leads to a total spin that, similar to that appearing for the total displacement in random walks (sum of the uniform length displacements), follows a Gaussian distribution. As seen in our equations, the situation for quantum spins is qualitatively similar. Even though, when $N \gg 1$, states of low $S_{tot}/N \sim \mathcal{O}(N^{-1/2})$ are statistically preferred in the entries of Eq. (G1), physically finite $S_{tot}/N$ ratios are, of course, mandatory in numerous instances (including the ability to cool/heat the energy density of the system at a finite rate). For instance, sans symmetry breaking fields, in low temperature ferromagnetic states (having a finite magnetization density or, equivalently, an extensive total spin), the total spin value $S_{tot} = \mathcal{O}(N)$. In the presence of the applied symmetry breaking field in $H_{spin}$ of Eq. (8), such a finite average of $(S_{tot}^z/N)$ arises

at general finite temperatures. Furthermore, as noted above, in order to have a finite rate of change of the energy density by applying the transverse field $B_y$ of Eq. (10), we must have that the total spin $S_{tot} = \mathcal{O}(N)$.

## Appendix H: Correlations in rotationally invariant spin systems driven by a uniform field and their inevitability when energy density derivatives have finite cumulants

### 1. Long range correlations

We will now briefly underscore that any eigenstate of $|\phi_\alpha\rangle$ of Eq. (8) having $S_{tot} = \mathcal{O}(N)$ with $|w| < 1$ displays long range correlations. As we will further explain, such macroscopic spin states with $|w| < 1$ must appear if the application of a transverse field in the example of Section VI A leads to, e.g., either (1) finite second cumulants (i.e., variances) the change of the energy density (in addition to a finite rate of variation of the energy density as required for the systems that we analyze) or generally leads to (2) finite second derivatives of the energy density for time dependent external fields (such as those of Eq. (H9) below).

First, we make the correlations in these states explicit by writing down the below two simple equalities,

$$\langle (S_{tot}^x)^2 \rangle = \frac{1}{2}\Big\langle \Big( \vec{S}_{tot}^2 - (S_{tot}^z)^2 \Big) \Big\rangle$$
$$= \frac{1}{2}\Big[ S_{tot}(S_{tot}+1)\hbar^2 - (S_{tot}^z)^2 \Big], \qquad \text{(H1)}$$

and

$$\langle (S_{tot}^x)^2 \rangle = \sum_{i \ne j} \langle S_i^x S_j^x \rangle + \sum_i \langle (S_i^x)^2 \rangle$$
$$= \sum_{i \ne j} \langle S_i^x S_j^x \rangle + \frac{N\hbar^2}{4}. \qquad \text{(H2)}$$

Combining Eqs. (H1, H2), and noting that in any eigenstate $|\phi_\alpha\rangle$ of the $S_{tot}^z$ operator, the expectation value $\langle S_i^x \rangle = 0$, one finds that, on average, for all $i \ne j$, the pair correlator

$$\frac{1}{N(N-1)} \sum_{i \ne j} (\langle S_i^x S_j^x \rangle - \langle S_i^x \rangle \langle S_j^x \rangle)$$
$$= \frac{S_{tot}(S_{tot}+1)\hbar^2 - (S_{tot}^z)^2}{2N(N-1)} - \frac{\hbar^2}{4(N-1)}. \qquad \text{(H3)}$$

For fully symmetric states $|\phi_\alpha\rangle$ (those associated with a maxima total spin, $S_{tot} = S_{\max} = NS$), all of the correlators when $i \ne j$ are equal to each other and given by the righthand side of Eq. (H3). The possibility of correlations in the initial state is consistent with our discussion following Eqs. (5,6). In the exactly solvable model system of Section VI, these correlations are of a particularly simple form of Eq. (H3).

## 2. The inevitability of long range correlations if a driving field leads to finite cumulants and/or finite averages of higher order derivatives of the energy density

### a. Finite variances of the derivative of the energy density

As noted earlier, in order for the system to display a finite rate of variation of its energy density (the focus of the systems discussed in our work), the spin system of Eq. 8 must have macroscopic ($\mathcal{O}(N)$) total $S_{tot}^z$ (as in a ferromagnet). While a finite average correlator for large $|i-j'|$ (such as that resulting when $S_{tot} = \mathcal{O}(N)$ and $|w| < 1$) might appear paradoxical, one must recall that for these states $|\phi_\alpha\rangle$, the application of the transverse field of Eq. (10) led to a finite range of change of the energy density. That is, when evaluated in these states, the expectation value of the time derivative of the Heisenberg picture Hamiltonian $\frac{d\epsilon}{dt} = \frac{1}{N}\langle \frac{dH^H(t)}{dt}\rangle \neq 0$ for general times $t$. Indeed, the latter inequality defined our problem (that of a finite rate of change of the energy density). Given that, at most times $t$, the first moment of $\frac{dH^H(t)}{dt}$ in the state $|\phi_\alpha\rangle$ is finite, it is no surprise that its second cumulant (i.e., the variance) may also be finite at these or other times. Indeed, when $\int_0^t B_y(t')\,dt' \equiv 0(\text{mod } \pi)$,

$$\frac{1}{N^2}\left(\left\langle \left(\frac{dH^H(t)}{dt}\right)^2 \right\rangle - \left\langle \frac{dH^H(t)}{dt}\right\rangle^2\right)$$
$$= \left(\frac{B_y(t)B_z}{N}\right)^2\left\langle (S_{tot}^x)^2\right\rangle. \qquad \text{(H4)}$$

Thus, for those times $t$ at which $\theta(t) \equiv 0(\text{mod } \pi)$ (which, coincidently, for $w \neq 0, \pm 1$, are the only times at which $\frac{d\epsilon}{dt} = \sigma_\epsilon = 0$) if the second cumulant of $\frac{1}{N}\frac{dH^H(t)}{dt}$ is finite then the initial state $|\psi_{Spin}^0\rangle = |\phi_\alpha\rangle$ must display a finite $\left\langle \left(\frac{S_{tot}^x}{N}\right)^2\right\rangle$. From Eq. (H2), a non-vanishing $\left\langle \left(\frac{S_{tot}^x}{N}\right)^2\right\rangle$ implies a finite average value of $(\langle S_i^x S_j^x\rangle - \langle S_i^x\rangle\langle S_j^x\rangle)$ for far separated sites $i$ and $j$. Hence, the correlations of Eq. (H3) are not unexpected in systems generally exhibiting finite cumulants of $\frac{1}{N}\frac{dH^H(t)}{dt}$. We must caution that, of course, the possibility of a finite first cumulant of $\frac{1}{N}\frac{dH^H(t)}{dt}$ at general times does not mandate the existence of a finite second cumulant (i.e., a variance). However, it certainly does not preclude it (as is indeed the case for our example of Section VI A). Generally, one anticipates a finite variance from the different local contributions to $\frac{dH^H(t)}{dt}$. These contributions are generally correlated due to the coupling between the local contributions (the local spins) to the external drive (the transverse field of Eq. (10)) to all spins in the system so as to change the energy density at a finite rate (as motivated by the qualitative discussion of Eq. (4)). That the variance of $\frac{1}{N}\frac{dH^H(t)}{dt}$ is given by Eq. (H4) may be explicitly seen as follows. In the Heisenberg picture, an evolution under the transverse field Hamiltonian of Eq. (10) leads to the precession

$$S_{tot}^z(t) = S_{tot}^z \cos\theta(t) - S_{tot}^x \sin\theta(t), \qquad \text{(H5)}$$

where, as in the main text, $\theta(t) \equiv \int_0^t B_y(t')dt'$. Invoking Eq. (8), this yields

$$\frac{dH^H(t)}{dt} = B_z B_y(t)\left(S_{tot}^z \sin\theta(t) - S_{tot}^x \cos\theta(t)\right)\text{(H6)}$$

This gives rise to Eq. (H4) when $\theta(t) \equiv 0(\text{mod } \pi)$.

### b. Finite averages of the second order derivative of the energy density

Higher order derivatives may be similarly examined. We next discuss the average of the second derivative of the energy density,

$$\frac{d^2 S_{tot}^z(t)}{dt^2} = -S_{tot}^z[B_y^2(t)\cos\theta + \frac{dB_y}{dt}\sin\theta]$$
$$+ S_{tot}^x[B_y^2 - \frac{dB_y}{dt}\cos\theta]. \qquad \text{(H7)}$$

In the following, we will very briefly discuss two special simple cases: (1) a time dependent and (2) a constant external field.

### Time dependent external field.

From Eq. (H7), if $\frac{1}{N^2}\langle (\frac{d^2 S_{tot}^z(t)}{dt^2})^2\rangle = \mathcal{O}(1)$ then whenever $[B_y^2(t)\cos\theta + \frac{dB_y}{dt}\sin\theta] = 0$, the variance $\langle (S_{tot}^x)^2\rangle = \mathcal{O}(N^2)$. Since $B_y(t) = \frac{d\theta}{dt}$, this yields the ordinary differential equation

$$\left(\frac{d\theta}{dt}\right)^2 = -\frac{d^2\theta}{dt^2}\tan\theta. \qquad \text{(H8)}$$

Explicitly integrating $(\frac{d^2\theta}{dt^2})/(\frac{d\theta}{dt}) = -\frac{d\theta}{dt}\cot\theta$ once implies $\ln\frac{d\theta}{dt} = -\ln(\sin\theta) + C_1$. An exponentiation and a second integration result in $\cos\theta = C_2 - Ct$ (with $C, C_{1,2}$, arbitrary integration constants). Hence, if $\theta(0) = 0$ then the solution to Eq. (H8) is, for $0 \leq t \leq \frac{2}{C}$, given by $\theta(t) = \cos^{-1}(1 - Ct)$ for general $C > 0$. Thus, if an applied field

$$B_y(t) = \frac{d\theta}{dt} = \frac{1}{\sqrt{\frac{2t}{C} - t^2}}, \qquad \text{(H9)}$$

not only trivially leads to a finite rate of change of the energy density but also to a finite $\frac{1}{N^2}\langle\frac{d^2 H^H}{dt^2}\rangle$ on a continuous time interval then $\langle (S_{tot}^x)^2\rangle = \mathcal{O}(N^2)$ (i.e., $|w| < 1$) signaling, as discussed in Appendix H 1, the existence of long range correlations.

### Constant external field.

Analogously, if apart from having a finite rate of change of the energy density, the square of the second order derivative $\frac{1}{N^2}\langle (\frac{d^2 H^H}{dt^2})^2\rangle > 0$ when $\theta(t) = \pi/2$ for a uniform time independent $B_y$ then, from Eq. (H7), $\langle (S_{tot}^x)^2\rangle = \mathcal{O}(N^2)$, i.e., $|w| < 1$ (implying long range correlations once again).

## Appendix I: Preparation of the initial spin states of Section VI A

The results of Section VI A hold for *any* initial state $|\psi_{Spin}^0\rangle$ that is an eigenstate of the Hamiltonian $H_{spin}$ of Eq. (8) evolved under the transverse field Hamiltonian $H_{tr}$ of Eq. (10). We reiterate that a finite rate of cooling or heating can be achieved by $H_{tr}$ only when the initial state $|\psi_{Spin}^0\rangle$ is of a macroscopic total spin $S_{tot} = \mathcal{O}(N)$ (e.g., a ferromagnet) and the ratio $w \equiv S_{tot}^z/(\hbar S_{tot}) \neq 0$. Furthermore, as noted earlier, the inequality $w \neq \pm 1$ must be satisfied in order for the initial state to differ qualitatively from a product state in which all spins are polarized along the $z$ direction. Indeed, as we explained in Section V, for initial product states, no spreading is possible (i.e., $\sigma_\epsilon = 0$). In a related manner, if $w = \pm 1$ then the transverse field Hamiltonian $H_{tr}$ will act as a pure displacement operator on the spin coherent state initially polarized along the $z-$ axis and lead to no spreading of the energy density as evaluated with Eq. (8). It is only for the fully polarized states $w = \pm 1$ that no spreading occurs. The states $|\psi_{Spin}^0\rangle$ that we considered are, of course, somewhat special (see also Appendix H). In this Appendix, we describe a purely gedanken experiment for preparing states (with either quantum or classical probability densities) of high spin $S_{tot} = \mathcal{O}(N)$ with $|w| < 1$. Towards this end, we first consider the Hamiltonian of Eq. (8) as that describing a typical ferromagnet F associated with the Hamiltonian $H_{Heisenberg}$ of Eq. (9) on a lattice of $N$ sites (having, e.g., all of the couplings in Eq. (9) non-negative) that is subjected to, at low temperatures, to a longitudinal external field ($B_z$). The latter external field is created by a large permanent magnet M of size $N_M = \mathcal{O}(N)$. The global magnetic field $B_z$ generated by M has small $\delta B_z = \mathcal{O}(N^{-1/2})$ fluctuations in its magnitude. We consider the "F − M" hybrid to be, initially, in contact with a thermal bath. In equilibrium, at low temperatures, the spins in F become polarized with the resulting total magnetization being parallel to the applied external field $B_z$ (viz., $S_{tot}^z = S_{tot} = \mathcal{O}(N)$). Next, we introduce a transverse field $B_y$ (captured by Eq. (10) or Eq. (15)) that acts on F. Following the application of the transverse field, the total spin will precess about the $y$ axis (see Figures 4 and 5). Next, we turn off the transverse field and let the system evolve under Eq. (8). As earlier, we do so by considering the F − M hybrid which is now closed (i.e., with no connection to an external heat bath). Now that the total spin is no longer polarized along the $z$ axis, the fluctuations in the values of $B_z$ will lead to a spread of precession of the total spin about the $z$ axis. After a time $\tau_{cover} \sim 2\pi/\delta B_z$ (assuming that this time scale is larger than the Lieb-Robinson time of Section IV, $\tau_{cover} > t_{LR}$), the probability distribution for the total spin covers uniformly a "line of latitude" of fixed $S_{tot}^z$ (see Figure 4). This resulting probability distribution for the total spin emulates that associated with $|\psi_{Spin}^0\rangle$ of Section VI A or that affiliated with the semiclassical distribution of Section VI A 2. Once a strong transverse field ($\|H_{tr}\| \gg \|H_{spin}\|$) is applied anew to this state, the results Eqs. (11, 14) will follow (the ring of Figure 4 will rotate to that of Figure 5). Similarly, Eq. (16) will yield the standard deviation of the energy density for the more general situation of Eq. (15) for an arbitrary size $H_{tr}$ augmenting $H_{spin}$.

## Appendix J: Intuitive arguments for the appearance of long time Gaussian distributions

The prediction of Eq. (56) for the viscosity of quintessential non-equilibrium liquids (supercooled liquids and glasses) that yielded the 16 decade collapse of Figure 7 was first derived [21] by computing long time averages and invoking a Gaussian distribution of finite width $\sigma_\epsilon$. At the other extreme, equilibrium systems also display a Gaussian distribution of their energy density $P(\epsilon')$. In [21], we motivated the presence of a Gaussian distribution by maximizing the Shannon entropy for a given $\sigma_\epsilon$. We now suggest that long time normal distributions (both in systems that exhibit long time equilibrium and those that do not such as glasses) might also be natural from other considerations. In general, the probability distribution $P(\epsilon')$ may be calculated along lines similar to those that led to Eq. (14) in our toy example of Eq. (8) where the system was continuously driven by an external transverse field. However, unlike the models studied in Section VI, at long times, supercooled liquids and glasses are no longer driven by an external bath $H_{tr}$ that continuously cools/heats them in a predetermined fashion. Instead, for supercooled liquids and glasses, at long times, the external heat bath (similar to the situation in equilibrium thermodynamics), becomes a source of stochastic noise (whose strength is set by its temperature $T$). Thus, the initially driven (i.e., continuously cooled) supercooled fluids or glasses will, at these long times, be effectively exposed to random noise. Following the reasoning that led to Eq. (14), we examine general moments of the Heisenberg picture Hamiltonian

$$\langle (\Delta \epsilon)^p \rangle \equiv \frac{1}{N^p} \langle (H^H - \langle H^H \rangle)^p \rangle \equiv \langle (\frac{\Delta H^H}{N})^p \rangle \quad \text{(J1)}$$

when evaluated in the initial equilibrium state prior to cooling $|\psi^0\rangle = \sum_n c_n^0 |\phi_n\rangle$. Here, $\{c_n^0\}$ are the amplitudes of the initial state $|\psi^0\rangle$ in the eigenbasis of the system Hamiltonian $H$. Writing Eq. (J1) longhand as a product of $p$ factors of $(\frac{\Delta H^H}{N})$, we have

$$\langle (\Delta \epsilon)^p \rangle = \sum_{n_1 n_2 \cdots n_p} c_{n_1}^{(0)*} c_{n_p}^{(0)} \Big( \frac{(\Delta H^H)_{n_1 n_2}}{N} \Big)$$
$$\times \Big( \frac{(\Delta H^H)_{n_2 n_3}}{N} \Big) \cdots \Big( \frac{(\Delta H^H)_{n_{p-1} n_p}}{N} \Big), \text{(J2)}$$

where $(\Delta H^H)_{ab}$ are the matrix elements of $\Delta H^H$ in the eigenbasis of $H$. If, at long times, the matrix elements of the scaled Heisenberg picture Hamiltonian $\frac{\Delta H^H}{N}$ (now

evolved with the stochastic influence of the environment at long times) attain random phases relative to each other then the only remaining contributions in Eq. (J2) will be those in which all matrix elements come in complex conjugate pairs of the type $\left(\frac{(\Delta H^H)_{ab}}{N}\right)\left(\frac{(\Delta H^H)_{ba}}{N}\right)$. More precisely, in the calculation of the long time average of Eq. (J2), only the temporal average of such complex conjugate pairs will not vanish. Thus, similar to the calculation that led to Eq. (14), only even moments $p = 2g$ may be finite. Now, however, the number of non-vanishing contributions (the number of ways in which the elements of $H^H$ may be matched in complex conjugate pairs) will scale as $\left(\frac{(2g)!}{2^g g!}\right)$. This, in turn, prompts us to consider the possibility that, approximately, $\langle(\Delta\epsilon)^{2g}\rangle \sim \left(\frac{(2g)!}{2^g g!}\right)\sigma_\epsilon^{2g}$. (This is especially the case if the initial state $|\psi^0\rangle$ corresponds to a single eigenstate of the Hamiltonian $H$, i.e., $c_{n_1}^{(0)} = \delta_{n_1,n}$ and $c_{n_p}^{(0)} = \delta_{n_p,n}$). If, for all $g$, these moments of $\Delta\epsilon$ are equal to those evaluated with a Gaussian distribution (as follows from Wick's theorem- the combinatorics of which essentially reappeared in the above), then the probability distribution $P(\epsilon')$ for obtaining different energy densities in the final state must, indeed, be a Gaussian.

---

• The above simple (non-rigorous) derivation rationalizes the appearance of Gaussian distributions in systems that equilibrate at long times (standard thermal systems) as well as the conjectured Gaussian form of $P(\epsilon')$ for supercooled liquids (Section XII) that led to Eq. (56).

---

For thermal fluctuations in standard ("canonical") systems, the resulting Gaussian distribution is defined by its average and a standard deviation that is linear in the temperature ($\sigma_\epsilon \propto T$) suggestive of Eq. (55). In a somewhat qualitatively similar manner, the stochastic effects of the environment are often simulated by Gaussian distributed forces whose standard deviation depends on the temperature $T$. The assumption of random phases in the above derivation of the Gaussian form does not, of course, imply small variances; the standard deviation of the energy density $\sigma_\epsilon$ (possibly still linear in the temperature) may be finite. As emphasized in Sections IX and XII, in thermal systems the (typically linear in $T$) standard deviation characterizing the distribution $P(\epsilon')$ is $\sigma_\epsilon = \sqrt{k_B T^2 C_v / N} \sim \mathcal{O}(N^{-1/2})$. More complex multi-scale probability distributions are, of course, possible (e.g., Appendix 6 of [21].) The above arguments for Gaussian distributions at long times may be replicated, by a trivial change of variables, to general intensive quantities $q$ other than the energy density.

## Appendix K: Typical High Entanglement Entropy States

As we underscored earlier, typical "thermal states" may exhibit an entanglement entropy that scales with the system volume [64], not its logarithm. The eigenstates $|\Psi_+\rangle$ examined in Appendix F were special in two different ways: (i) The eigenstates were constructed as an equal weight symmetric combination of all local product states and (ii) The systems that we examined were endowed with a local "quasi-particle" structure embodied by the independent commuting operators $\{\mathcal{H}_i\}$ (and associated local product eigenstates). In general, even when only property (i) is violated, larger entropies may arise. It is instructive to see why this is so and how the state $|\Psi_+\rangle$ is special inasmuch as the calculation of its entanglement entropy is concerned. In the space spanned by all product states $|c\rangle$ that given energies $E_B$ (instead of that performed in Appendix F in the basis of the symmetric basis of Eqs. (F2)), the reduced density matrix $\rho_{B,+}$ becomes block diagonal. Repeating the calculation of Appendix F in this basis, we find that in each region of fixed energy $E_B$, the block matrix is equal to

$$\mathsf{One} = \begin{pmatrix} 1 & 1 & 1 & \cdots & 1 \\ 1 & 1 & 1 & \cdots & 1 \\ . & . & . & \cdots & 1 \\ . & . & . & \cdots & 1 \\ 1 & 1 & 1 & \cdots & 1 \end{pmatrix} \qquad (K1)$$

multiplied by the factor $e^{(\mathsf{S}_A(E-E_B)-\mathsf{S}(E))/k_B}$. The dimensions of the matrix $\mathsf{One}$ are determined by the number $\mathcal{N}_B(E_B) = e^{\mathsf{S}_B(E_B)/k_B}$ of degenerate states $\{|E_B, j\rangle\}_{j=1}^{\mathcal{N}_B(E_B)}$ that have an energy $E_B$ on the spatial region $B$. We may perform a unitary transformation to the discrete Fourier basis (spanned by the states $|E_B, k_{E_B}\rangle \equiv (\mathcal{N}_B(E_B))^{-1/2}\sum_{j=1}^{\mathcal{N}_B(E_B)} e^{ik_{E_B}j}|E_B, j\rangle$ with the wavenumber $k = 2\pi m/\mathcal{N}_B(E_B)$ where $m = 0, 1, 2, \cdots, \mathcal{N}_B(E_B) - 1$). This transformation reduces the (generally large) matrices of the form of Eq. (K1) to a single non-vanishing entry. Indeed, up to a constant prefactor (of $\mathcal{N}_B(E_B)$), the matrix $\mathsf{One}$ is set by the outer product

$$|E_B, k_{E_B} = 0\rangle\langle E_B, k'_{E_B} = 0|. \qquad (K2)$$

To make the contact with Appendix F lucid, we remark that in the notation of Eqs. (F2), the single non-vanishing Fourier mode $|E_B, k_{E_B} = 0\rangle = |E_B\rangle_+$. In each block of fixed energies $E_B$, all other discrete Fourier ($k_{E_B} \neq 0$) modes have a vanishing amplitude. Such a Fourier transformation immediately yields the eigenvalue spectrum,

$$\mathsf{Spec}\{\mathsf{One}\} = \{\mathcal{N}_\mathsf{B}(\mathsf{E}_\mathsf{B}), \underbrace{0, 0, 0, \cdots, 0}_{\mathcal{N}_\mathsf{B}(\mathsf{E}_\mathsf{B})-1}\}. \qquad (K3)$$

Thus, upon performing a unitary transformation to the Fourier basis, the block diagonal matrix $\rho_{B,+}$ becomes sparser (each vanishing eigenvalue of the reduced density matrix $\rho_{B,+}$ does not contribute to the entropy) and only the completely symmetric states of Eq. (F2) are of relevance. If the equal amplitude eigenstates $|\Psi_+\rangle$ are replaced by a general linear combination $|\Psi_{\{a_c\}}\rangle = \sum_{E_c=E} a_c |c\rangle$ (with $\sum_c |a_c|^2 = 1$) then the associated reduced density matrix $\rho_{B,\{a_c\}} = Tr_A |\Psi_{\{a_c\}}\rangle\langle\Psi_{\{a_c\}}|$ will remain block diagonal. However, the block matrices that span each region of fixed energy $E_B$ will, generally, look very different from One. Intuition concerning the larger

entanglement entropies that may generally result can be gained by suggestive arguments. Towards this end, we may consider what occurs if each diagonal block of $\rho_{B,+}$ of the type One is replaced by other block diagonal matrices with a wider distribution of the eigenvalues such that, e.g., each of the non-vanishing eigenvalues of $\rho_{B,+}$ for energies $E_B$ (close to the energy $\overline{E}_B$ that maximizes the sum $\mathsf{S}_A(\overline{E}_A = E - \overline{E}_B) + \mathsf{S}_B(\overline{E}_B)$) is, effectively, split into $K$ equal parts. In such a case, the entanglement entropy $\mathsf{S}_{ent,\{a_c\}}$ will be larger than $S_{ent,+}$ by an additive contribution of $\ln K$. If, for $L_B < L_A$, the logarithm $(\ln K) = \mathcal{O}(\mathsf{S}_B(\overline{E}_B)) = \mathcal{O}(L_B)$ then this additive contribution to the entanglement entropy may be linear in the volume $L_B$ of subsystem $B$.

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

$$\sigma_\epsilon^2 = (\sigma_\epsilon^{\text{class.}})^2 + (\sigma_\epsilon^{\text{quant.}})^2. \qquad (K4)$$

Here,

$$(\sigma_\epsilon^{\text{class.}})^2 \equiv \frac{1}{N^2}(\sum_\ell p_\ell E_\ell^2 - E^2), \qquad (K5)$$

(note that, generally, $\langle \psi_k | H^2 | \psi_\ell \rangle \neq E_\ell^2$) and

$$(\sigma_\epsilon^{\text{quant.}})^2 \equiv \frac{1}{N^2}(\sum_\ell p_\ell \sigma_\ell^2), \qquad (K6)$$

where the variances

$$\sigma_\ell^2 \equiv \langle \psi_\ell | (H - E_\ell)^2 | \psi_\ell \rangle, \qquad (K7)$$

stem from the fluctuations of Hamiltonian in given eigenstates of the density matrix. (Similar classical and quantum standard deviations may be defined for quantities $q$ different from the energy density $\epsilon$.) In classical systems where $H$ and $\rho$ commute, $(\sigma_\epsilon^{\text{quant.}})^2 = 0$ and the sole contribution to the variance is that of Eq. (K5) as it is in, e.g., equilibrium classical systems in the canonical ensemble. The standard deviation of $(H/N)$ (i.e., $\sigma_\epsilon$) should not, of course, be confused with $\sigma_\epsilon^{\text{class.}}$. Indeed, in a general single pure state (a situation for which $\sigma_\epsilon^{\text{class.}} = 0$), the standard deviation of the energy density may be finite, $\sigma_\epsilon = \mathcal{O}(1)$. Concrete realizations of this maxim will be provided in the examples of Section VI. Physically, one generally anticipates that when reaching a quasi-static state, different eigenstates of the density matrix will exhibit similar energy densities, viz. $\sigma_\epsilon^{\text{class.}} = 0$ (for otherwise heat may be rapidly exchanged between states $|\psi_\ell\rangle$ that are of higher/lower energy densities ($E_\ell/N$) and their given surrounding heat bath). A rather example of a classical broad distribution mixing various classical product states leading to a nonvanishing standard deviation is afforded by mixing in the ground state sector of the ferromagnetic Ising model. An equal probability mixing of the two fully spin polarized product states of the classical Ising model, $| \uparrow\uparrow \cdots \uparrow\rangle$ and $| \downarrow\downarrow \cdots \downarrow\rangle$ exhibits a finite standard deviation of the value of the local magnetization density $\mathsf{m} = \frac{1}{N}\sum_{i=1}^N s_i$ (i.e., $\sigma_\mathsf{m} = 1$ irrespective of the system size $N$). An equal amplitude superposition of these two (product) ground states of the ferromagnetic Ising model, viz., the ground state $|+\rangle \equiv \frac{1}{\sqrt{2}}(| \uparrow_1\uparrow_2 \cdots \uparrow_N\rangle + | \downarrow_1\downarrow_2 \cdots \downarrow_N\rangle)$, features a standard deviation $\sigma_\mathsf{m} = 1$. In the latter two (classical and quantum) examples concerning the ground states of the ferromagnetic Ising model, the uncertainty in the energy of the ground states of the ferromagnetic Ising model is, of course, zero ($\sigma_\epsilon = 0$).

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

ergies between the final and initial states will be given by $E_\chi - E_{\psi^{(0)}} = NdJ$. In this example, the standard deviation of the energy in the final state $|\chi\rangle$ is equal to $J\sqrt{Nd}$. Note that for randomly chosen initial states, the difference in the energy densities of the final and initial states will tend to zero in the thermodynamic limit (i.e., the probability distribution of $(E_{\psi^{(0)}}/N)$ for all possible initial states $|\psi^{(0)}\rangle$ will become a delta function as $N \to \infty$).

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

mal system in $\mathcal{S}_{th}$, i.e., $\mathsf{S}_{ent} \equiv -Tr[\rho_{S;th} \ln \rho_{S;th}] = -Tr[\rho_{eq}(S_{th}) \ln \rho_{eq}(S_{th})] \equiv \mathsf{S}_{eq}$. Since the entropy $\mathsf{S}_{eq}$ of an equilibrium system at temperatures $T > 0$ is an extensive quantity, it follows that the entanglement entropy $\mathsf{S}_{ent}$ of an equilibrium system with its environment obeys a "volume" law (i.e., the entanglement entropy will be proportional to the spatial volume of $\mathcal{S}_{th}$).

[65] H. Araki and E. H. Lieb, "Entropy Inequalities", Communications in Mathematical Physics **18**, 160-170 (1970) (see Lemma 4, in particular).

[66] Using the "purification" of [65], the proof of this assertion can be made trivial as we now explain. Diagonalizing the density matrix of dimension $\mathcal{D}$ on a region $\mathcal{S}$ we have $\rho = \sum_{a=1}^{\mathcal{D}} p_a |\psi_a\rangle\langle\psi_a|$. If we associate to each state $|\psi_a\rangle$ that has its support in $\mathcal{S}$ a different orthogonal state $|\overline{\psi}_a\rangle$ (with $\langle\overline{\psi}_a|\overline{\psi}_{a'}\rangle = \delta_{aa'}$) having its support in the environment $\mathcal{E}'$ (i.e., $\mathcal{S} \cup \mathcal{E}' = \mathcal{I}'$ and $\mathcal{S} \cap \mathcal{E}' = \{\emptyset\}$). With these elements in tow, we will then have $\rho = Tr_{\mathcal{E}'} |\Psi\rangle\langle\Psi|$ where $|\Psi\rangle \equiv \sum_{a=1}^{\mathcal{D}} \sqrt{p_a} |\psi_a\rangle \otimes |\overline{\psi}_a\rangle$ is a pure state in $\mathcal{I}'$.

[67] When adding the spin of $N$ particles, as in, e.g., Eq. (G1), there is a unique sector of maximal total spin $S_{tot} = S_{\max}$. For lower values $S_{tot}$, the eigenstates $|\phi_\alpha\rangle$ of the Hamiltonian $H_{spin}$ need to be specified by an additional index $(\upsilon_\alpha)$, see also Appendix G.

[68] The result of Eq. (11) may readily be seen by inspection. The applied transverse field leads to a global Larmor precession of the spins about the $y-$axis. While the first term of Eq. (8) is manifestly invariant under rotations, the second term (that of $(-B_z \sum_i S_i^z)$) will change. In the Heisenberg picture after the evolution with the transverse field, each such local operator $(B_z S_i^z)$ transforms into $B_z(S_i^z \cos(\int_0^{t_f} B_y(t') \, dt') + S_i^x \sin(\int_0^{t_f} B_y(t') \, dt'))$. Since in any eigenstate of $S_{tot}^z$ (including $|\psi_{Spin}^0\rangle$), the expectation value $\langle S_{tot}^x S_{tot}^z\rangle = \langle S_{tot}^x\rangle\langle S_{tot}^z\rangle(= 0)$, the only non-vanishing contributions to the variance of the Hamiltonian of Eq. (8) will originate from the expectation value of the square of the second term of $H_{spin}$ and thus (up to a trivial prefactor of $(B_z^2 \sin^2(\int_0^{t_f} dt' B_y(t'))))$ from

$$\sigma_{S_{tot}^x}^2 = \langle (S_{tot}^x)^2\rangle = \frac{1}{2}\langle (S_{tot}^x)^2 + (S_{tot}^y)^2\rangle$$
$$= \frac{1}{2}\langle (\vec{S}_{tot})^2 - (S_{tot}^z)^2\rangle$$
$$= \frac{1}{2}\left(\hbar^2 S_{tot}(S_{tot}+1) - (S_{tot}^z)^2\right). \quad (K8)$$

Substituting $w \equiv S_{tot}^z/(\hbar S_{tot})$ (and rescaling by a factor of $N^2$ to determine the variance of the energy density) leads to the square of Eq. (11).

[69] Even in the state of maximal spin $S_{tot} = S_{\max}$, so long as $|w| < 1$, the standard deviation will generally be $\sigma_\epsilon = \mathcal{O}(1)$. Furthermore, although they are statistically preferable values for $S_{tot}$ when adding angular momenta in the large $N$ limit (e.g., Appendix G), regardless of the form of $H_{symm}$ (for instance, irrespective of the specific couplings in Eq. (9)), in this $N \gg 1$ limit, states of vanishingly small $\frac{S_{tot}}{N}$ will not allow for a for a finite change of the energy density, $\Delta\epsilon = B_z \frac{S_{tot}^z}{N}(1 - \cos(\int_0^{t_f} B_y(t') \, dt'))$, via the application of the transverse field (as embodied by the Hamiltonian $H_{tr}$). Indeed, the central point that we wish to emphasize and is evident in our example of Eq. (8) is that, generally, when the energy density $\Delta\epsilon$ does change at a non-vanishing rate, a finite $\sigma_\epsilon > 0$ is all but inevitable.

[70] Explicitly, $\langle e^{i\mathsf{q}(\Delta\epsilon')}\rangle = \sum_{g=0}^{\infty} \frac{(i\mathsf{q})^{2g}}{2^g(2g)!}\binom{2g}{g}\sigma_\epsilon^{2g} = \sum_{g=0}^{\infty} \frac{(-1)^g}{(g!)^2}(\frac{\mathsf{q}\sigma_\epsilon}{\sqrt{2}})^{2g} = J_0(\mathsf{q}\sigma_\epsilon\sqrt{2})$.

[71] We quickly comment, in the spirit of the discussion appearing at the end of Section IV, on what transpires if the forms of the Hamiltonian $H_{spin}$ and, notably, the coupling of the spins to the transverse field as embodied by the Hamiltonian $H_{tr}$ apply only for spins on the surface of the system (the number of these spins $N_{surface} = \mathcal{O}(N^{(d-1)/d})$). In such situations, the standard deviation in Eq. (11) would naturally scale as $\sigma_\epsilon = \mathcal{O}(N^{-1/d})$. The distribution of Eq. (14) will retain its form. If this circumstance arises then, in the discussions and equations in the main text concerning $H_{spin}$ and $H_{tr}$, the total number of spins $N$ will be replaced by the number of surface spins $N_{surface}$.

[72] E. J. Torres-Herrera, M. Vyas, and L. F. Santos, "General features of the relaxation dynamics of interacting quantum systems", New J. of Physics **16**, 063010 (2014).

[73] E. H. Lieb, "The Classical Limit of Quantum Spin Systems", Commun. Math. Phys. **31**, 327 (1973).

[74] B. Simon, "The classical limit of quantum partition functions", Commun. Math. Phys. **71**, 247 (1980).

[75] The calculation for any ensemble of initial microstates that is invariant under rotations about the $z$ axis that have fixed values of $S_z^{tot}$ and $\vec{S}_{tot}^2$ mirrors that of [68] with the eigenstate expectation values replaced by ensemble averages. Eq. (K8) will be trivially modified in the classical case. Specifically, in the last line of Eq. (K8), the quantum $\hbar^2 S_{tot}(S_{tot}+1)$ will be replaced by the classical $\vec{S}_{tot}^2$.

[76] T. Matsubara and H. Matsuda, "A Lattice Model of Liquid Helium, I", Prog. Theor. Physics **16**, 569 (1956).

[77] Specifically, the Matsubara-Matsuda transformation [76] from spin $S = 1/2$ raising and lowering operators to the creation and annihilation operators of hard core bosons is given by $\frac{S_i^+}{\hbar} \to b_i^\dagger$ and $\frac{S_i^-}{\hbar} \to b_i$ (from which it follows that $\frac{S_i^z}{\hbar} \to (n_i - \frac{1}{2})$ where $n_i = b_i^\dagger b_i$ is the hard core boson number operator).

[78] With $c_i$ and $c_i^\dagger$ denoting respectively, the spineless Fermi annihilation and creation operators (and with $n_i = c_i^\dagger c_i$), the Jordan-Wigner dual of the initial spin Hamiltonian of Eq. (8) reads

$$H_{Fermi} = \sum_{|i-j|=1} J_{ij}((c_i^\dagger c_j + h.c.) + n_i n_j)$$
$$- \sum_i (B_z - \sum_{|j-i|=1} J_{ij})n_i. \quad (K9)$$

We may consider the initial spinless fermion system of Eq. (K9) that is governed, at intermediate times, by the (now non-local) Jordan-Winger dual of $H_{tr}$ or, equivalently, the fermonized dual of $H_{Bose-doping}$. Following a temporal evolution with the latter intermediate time Hamiltonian, an initial eigenstate of $H_{Fermi}$ formed by the equal amplitude superposition of all spineless Fermi states of a fixed total particle number (the dual of $|\psi_{Spin}^0\rangle$) will transform into a final state displaying the standard deviation and distribution of Eqs. (11, 14). (In order to make the transformation lucid, we remark that the spinless Fermi dual of the particular spin $S = 1/2$ state $|\uparrow_1\uparrow_2\downarrow_3\uparrow_4\downarrow_5\uparrow_6 \cdots \uparrow_{N-1}\downarrow_N\rangle$ describes an antisym-

metrized superposition of all product states in which the fixed total number of spineless fermions $\sum_i n_i$ are distributed over the specific sites $(1, 2, 4, 6, \cdots, (N-1))$.)

[79] A. Auerbach, "Interacting Electrons and Quantum Magnetism", Springer-Verlag (1994).

[80] For a discussion of the standard uncertainty relations for mixed states see, e.g., L. Ballentine, "Quantum Mechanics" (World Scientific, Singapore, 1998).

[81] We remark that a shorter, one line, proof of the uncertainty relations for mixed states follows from the fact that any density matrix may be expressed in terms of pure states [65, 66]. If, in the notation of [66], we take $\mathcal{I} \subset \mathcal{I}'$, then the uncertainty relation for pure states implies the corresponding uncertainty relations for general mixed states with arbitrary density matrices.

[82] In mixed states, all density matrices that are diagonal in the eigenbasis of $\tilde{H}$ are trivially stationary under the evolution with $\tilde{H}$. Such diagonal matrices may, however, display a standard deviation $\sigma_{\tilde{H}}$ that is large as $(\tilde{E}_{\max} - \tilde{E}_{\min})/2$ (where $\tilde{E}_{\max}$ and $\tilde{E}_{\min}$ denote the maximal and minimal eigenvalues of $\tilde{H}$). By contrast, if the pure state density matrix $\tilde{\rho}(t)$ of Eq. (36) exhibits a large standard deviation $\sigma_{\tilde{H}}$ then it also has high weight components that fluctuate rapidly in time (of frequency scale $\sigma_{\tilde{H}}/\hbar$).

[83] An all too simple example of a closed large (semi-classical) system with typical system size independent frequencies governing the global evolution is that of nearly harmonic elastic solids. The eigenmodes providing the frequencies associated with system dynamics do not scale with the system size. (Note that in this closed system example, the different eigenmodes cannot couple independently to different baths- the total energy of the (semi-classical) system is conserved.)

[84] For instance, one may concoct states $|\tilde{\psi}\rangle$ (or density matrices $\tilde{\rho}$) in $\mathcal{I}$ that exhibit large variances $\sigma_{\tilde{H}}^2 = (\langle\tilde{\psi}|\tilde{H}^2|\tilde{\psi}\rangle - (\langle\tilde{\psi}|\tilde{H}|\tilde{\psi}\rangle)^2)$ when the system size increases. (This may be achieved by superposing eigenstates that span an extensive ($\mathcal{O}(E) = \mathcal{O}(N)$) range of $\tilde{H}$ eigenvalues. (As discussed in Section V, on their own, product states on their own will lead to $\sigma_{\tilde{H}} \sim \mathcal{O}(\sqrt{N})$.)) The microcanonical ensemble (for which $\sigma_{\tilde{H}}$ is bounded) will be rendered incompatible in states like these for which the standard deviation scales with the system size.

[85] For textbook discussions, see, e.g., K. Huang, "Statistical Mechanics", Wiley, New York (1987) or F. Schwabl, "Statistical Mechanics", Springer series, Berlin Heidelberg New York (2002). As noted in [84], one can certainly construct states with extensive uncertainties. A system size independent $\sigma_{\tilde{H}} = \mathcal{O}(1)$ for a closed classical system in equilibrium (the defining property of the microcanonical ensemble) implies the same for a large many body semi-classical quantum system.

[86] Aside from considerations regarding fundamental time measurement bounds (and the equations that describe the measurable system), such an assumption is further plausible since during any time step, for a system with short range interactions, the error generated by a finite time discretization of the evolution of the charge/spin/... density at any point is independent of the system size (as it depends only on the local surrounding densities of the said point).

[87] The uncertainty relation of Eq. (32) was derived from Eq. (31) by expressing $\tilde{H} = i\hbar\frac{\partial}{\partial t}$; the discrete time gradients of bounded quantities such as $(H/N)$ (and those of other measurable quantities $Q$) as well as the standard deviation of the time discretized version of $\tilde{H}$ cannot exceed $\mathcal{O}(1/\Delta t)$. However, the expectation values of gauge non-invariant quantities can be made arbitrarily large or small. For instance, in non-relativistic systems such as the ones that we study, the energy can be shifted by a constant without altering the expectation values of all measurable observables $Q$ so long as these observables are not explicitly time dependent ($\frac{\partial Q}{\partial t} = 0$). A shift of the energy by a constant is tantamount to a trivial gauge transformation- a multiplication of the wavefunction by a linear in time phase factor. We emphasize that the measurable frequencies of these oscillations and their standard deviation are bounded in time discretized systems. The standard deviation of the energy density is a physical gauge invariant quantity (as are general relative energy differences) and all quantities evaluated with the density matrix $\tilde{\rho}(t)$ of Eq. (36).

[88] Assumption (3) and Assumption (3') do not hold for the factorizable states of Section V when $M = \mathcal{O}(N)$ with a Hamiltonian that acts disjointly on each of the $M$ local regions. If a product state description exists for the *closed isolated* hybrid system $\mathcal{I} = \mathcal{S} \cup \mathcal{E}$ (i.e., if the density matrix on $\mathcal{I}$ factorizes into density matrices on disjoint spatial regions) then we may focus on the primitive disjoint subhybrid system $\mathcal{I}_{\max} = \mathcal{S}_{\max} \cup \mathcal{E}_{\max}$ of $\mathcal{I}$ (wherein $\mathcal{I}_{\max} \subset \mathcal{I}$, $\mathcal{S}_{\max} \subseteq \mathcal{S}$, and $\mathcal{E}_{\max} \subseteq \mathcal{E}$). For systems with short range bounded strength interactions, the subsystem $\mathcal{S}_{\max}$ spans a volume of size $\mathcal{O}(N)$ such that all sites in $\mathcal{S}_{\max}$ only couple to those in $\mathcal{E}_{\max}$. For such systems, in order for the energy density of $\mathcal{S}$ to vary at an extensive rate, $\mathcal{S}_{\max}$ (and $\mathcal{E}_{\max}$) can have a volume of size $\mathcal{O}(N)$ (and, respectively, of a volume that is at least $\mathcal{O}(N)$); if all volumes $\mathcal{I}$ were uncorrelated and decoupled from each other the rate of change of the energy density may scale as $\sqrt{N}$ (i.e., not as $\mathcal{O}(N)$). The density matrix in $\mathcal{I}_{\max}$ cannot be further factorized into density matrices on disjoint spatial regions. It is for this density matrix on $\mathcal{I}_{\max}$ that we can then aim to apply Assumptions (1-3) (or Assumptions (1,2,3')) and the uncertainty inequalities that follow leading to Eq. (41).

[89] Although Assumptions (1-3) of Section IX A are not always satisfied (e.g., [84]), rather general systems do obey them and further exhibit a finite rate of change of the energy density $|\frac{d\epsilon}{dt}|$. To make contact with our earlier examples of Section VI, as a case in point, we may take $\tilde{H}$ to be $H_{spin}$ of Eq. (8) having a short range $H_{symm}$ and in which the magnetic field $B_z$ is along the $z$ direction over the entire volume $\mathcal{I}$; in a similar spirit, we consider the Hamiltonian $H$ in the region $\mathcal{S}$ to be given by Eq. (8) with an internal applied field $B_y$ parallel to the $y$ direction. Thus, in this example (apart from "surface terms" arising from the short range interactions in $H_{symm}$), the interaction between the system $\mathcal{S}$ and its external environment $\mathcal{E}$ is dominated by an external field ($\vec{B} = (0, -B_y, B_z)$) applied from sites exterior to $\mathcal{S}$. This external field couples to all spins in $\mathcal{S}$; when added together with $H$, this field will reproduce the terms of $\tilde{H}$ appearing in $\mathcal{S}$. In this example, the density matrix $\rho = |\psi\rangle\langle\psi|$, where we will choose $|\psi\rangle$ to correspond to an equal ampli-

tude superposition of two eigenstates of maximal total spin $S_{tot} = \mathcal{O}(N)$ that differ by one quantum of $\hbar$ in their $S_z^{tot}$ eigenvalues. That is, in the convention of Section VI, $|\psi\rangle = \frac{1}{\sqrt{2}}(|v_\alpha; S_{tot}, S_{tot}^z\rangle + |v_\alpha; S_{tot}, S_{tot}^z - \hbar\rangle)$. It is readily verified that Assumptions (1-3) are satisfied. The state $|\psi\rangle$ displays a standard deviation $\sigma_{\tilde{H}} = \frac{\hbar|B_z|}{2} = \mathcal{O}(1)$ (consistent with the finite standard deviation of the total energy characterizing the microcanonical ensemble on the full volume $\mathcal{I}$). However, within the smaller subvolume $\mathcal{S}$, the standard deviation of the energy density (i.e., the standard deviation of $H/N$) is $\sigma_\epsilon = \frac{|B_y|\hbar S_{tot}}{2N}\sqrt{2 + \frac{2}{S_{tot}} - (w^2 + (w - \frac{1}{S_{tot}})^2)} = \mathcal{O}(1)$. That is, the standard deviation of the total energy of $\mathcal{S}$ is generally extensive, $\sigma_H = \mathcal{O}(N)$. In the above, as in Section VI, we employ the shorthand $w \equiv S_{tot}^z/(\hbar S_{tot})$. The rate of change of the energy density is $|\frac{d\epsilon}{dt}| = \frac{|B_y B_z|\hbar S_{tot}}{2N}\sqrt{1 + \frac{1}{S_{tot}} - w(w - \frac{1}{S_{tot}})}$. Thus, $|\frac{d\epsilon}{dt}| = \mathcal{O}(1)$ for finite $B_y$ and $B_z$ when $S_{tot} = \mathcal{O}(N)$ and $|w| < 1$.

[90] In general, the sites exterior to $\mathcal{I}$ will give rise to an evolution of a canonical ensemble probability density matrix as given by Eq. (51) applied to $\mathcal{I}$ (and $\mathcal{S}$). In systems that exhibit non-equilibrium behavior, the time evolved $\tilde{H}^H(t)$(and $H^H(t)$)) may not be similar to the initial $\tilde{H}$ (and $H$). Consequently, the resulting probability distribution $\tilde{\rho}(t)$ will give rise to averages that differ from those in equilibrium. That is, Eq. (31) will always be satisfied. However, the time evolved $H^H(t)$ and $\tilde{H}^H(t)$ will, when Eq. (42) holds, deviate from the initial Hamiltonians $H$ and $\tilde{H}$ and thus have expectation values and fluctuations different from those in equilibrium.

[91] S. Goldstein, T. Hara, and H. Tasaki, "Extremely quick thermalization in a macroscopic quantum system for a typical nonequilibrium subspace", New J. of Physics **17**, 045002 (2015).

[92] M. B. Mensky, "Quantum restrictions for continuous observation of an oscillator", Phys. Rev. D. **20**, 384 (1979).

[93] Y. Aharonov, D. Z. Albert, and L. Vaidman, "How the result of a measurement of a component of the spin of a spin-1/2 particle can turn out to be 100", Phys. Rev. Lett. **60**, 1351 (1988).

[94] V.P. Belavkin, "Quantum continual measurements and a posteriori collapse on CCR", `https://arxiv.org/pdf/math-ph/0512070.pdf`, Commun. Math. Phys. **146**, 611 (1992).

[95] N. Foroozani, M. Naghiloo, D. Tan, K. Molmer, and K. W. Murch, "Correlations of the Time Dependent Signal and the State of a Continuously Monitored Quantum System", `https://arxiv.org/pdf/1508.01185.pdf`, Phys. Rev. Lett. **116**, 110401 (2016).

[96] B. Misra and E. C. G. Sudarshan, "The Zeno's paradox in quantum theory", J. of Mathematical Physics **18**, 756 (1977).

[97] F. A. Wolf, "A Watched Pot Never Boils" (November 27, 2000) `https://www.forbes.com/asap/2000/1127/093.html`.

[98] A well defined temperature does not always imply a sharp energy density. For instance, a finite interval of the internal energy densities corresponds to the single melting (and other coexistence) temperature(s). This width of this interval is set by the latent heat of fusion at the melting temperature. At other temperatures in which no phase coexistence appears, there is a unique

internal energy density (a sharp thermodynamic state variable) $\epsilon(T)$ associated with every temperature $T$.

[99] If the spectrum of $H$ exhibits degeneracies then we will explicitly choose the states $\{|\phi(\{q'\}; \mathcal{W})\rangle\}$ to be common eigenstates of both $H$ and $\mathcal{O}$.

[100] For completeness, we note that our generally derived broadening of the distribution $P(\epsilon')$ in driven systems would imply relations similar to Eq. (48) also for systems that do not exhibit equilibrium properties, e.g.,[54–62]; averages of observables $\mathcal{O}$ with the broadened distribution $P$ differ from those with a delta function type distribution.

[101] Eq. (51) implies that if the initial Hamiltonian satisfies the Eigenstate Thermalization Hypothesis (ETH) [45–53] (according to which single eigenstate expectation values are identical to those computed with the microcanonical or canonical ensemble) then the same will hold at all times with $H$ replaced by $H^H(t)$. This may be proven by setting, in Eq. (51), $f(H^H) = \delta(H^H - E_n)(\delta(\mathcal{W}^H - \mathcal{W}_n))$ with $E_n$ the energy of the eigenstate (and $\mathcal{W}^H$ the Heisenberg picture operator with eigenvalues $\mathcal{W}_n$ further specifying the eigenstate if a degeneracy exists) for which the eigenstate expectation value and the equilibrium average are identical. Thus, Eq. (51) may rationalize the results of a recent preprint: S. Moudgalya, T. Devakul, D. P. Arovas, and S. L. Sondhi, "An Extension of ETH to Non-Equilibrium Steady States", arXiv 1811.03114v1 (2018)

[102] T. B. Liverpool, " Non-equilibrium systems have steady-state distributions and non-steady dynamics", `https://arxiv.org/pdf/1810.10980.pdf`

[103] Louk Rademaker and Jan Zaanen, "Quantum Thermalization and the Expansion of Atomic Clouds", Scientific Reports **7**, 6118 (2017), `https://arxiv.org/pdf/1703.02489.pdf`.

[104] For completeness, we remark that in this case in which the environment is not explicitly included (and, hence, the system is closed), the system evolution is unitary, the spectrum of the density matrix cannot change. Thus, it will not be possible for a system displaying a Boltzmann distribution at an initial temperature to re-equilibrate to a Boltzmann distribution at a different final temperature (since the set of eigenvalues of equilibrium density matrices at different temperatures is not the same (yet under a unitary evolution the spectrum cannot change)).

[105] S. Goldstein, T. Hara, and H. Tasaki, "On the time scales in the approach to equilibrium of macroscopic quantum systems", `https://arxiv.org/pdf/1307.0572.pdf` (2013), Phys. Rev. Lett. **111**, 140401 (2013).

[106] S. Sachdev, Quantum Phase Transitions, second Edition, Cambridge University Press (2011).

[107] J. A. N. Bruin, H. Sakai, R. S. Perry, and A. P. Mackenzie, "Similarity of Scattering Rates in Metals Showing T-Linear Resistivity", Science **339**, 804 (2013).

[108] J. Zaanen, "Superconductivity: Why the temperature is high", Nature **430**, 512 (2004).

[109] S. Nussinov, T. Madziwa-Nussinov, and Z. Nussinov, "Decoherence due to thermal effects in two quintessential quantum systems", `https://arxiv.org/pdf/1407.0362.pdf`, Quantum Studies: Mathematics and Foundations **1**, 155 (2014).

[110] Sean A. Hartnoll, "Theory of universal incoherent

metallic transport", `https://arxiv.org/pdf/1405.3651.pdf`(2014), Nature Physics **11**, 54 (2015).

[111] Z. Nussinov, F. Nogueira, M. Blodgett, and K. F. Kelton, "Thermalization and possible quantum relaxation times in "classical" fluids: theory and experiment", `https://arxiv.org/pdf/1409.1915.pdf` (2014).

[112] H. Eyring, "The Activated Complex in Chemical Reactions", J. Chem. Phys. **3**, 107 (1935).

[113] F. Leyvraz and S. Ruffo, "Ensemble inequivalence in systems with long-range interactions" `https://arxiv.org/pdf/cond-mat/0112124.pdf`, J. of Physics A - Mathematical and General **35**, 285 (2002).

[114] J. Barre, D. Mukamel, and S. Ruffo, "Inequivalence of ensembles in a system with long-range interactions", `https://arxiv.org/pdf/cond-mat/0102036.pdf`, Phys. Rev. Lett. **87**, 030601 (2001).

[115] A. Campa, T. Dauxois, and S. Ruffo, "Statistical mechanics and dynamics of solvable models with long-range interactions", `https://arxiv.org/pdf/0907.0323.pdf`, Physics Reports **480**, 57 (2009).

[116] Y. Murata and H. Nishimori, "Ensemble Inequivalence in the Spherical Spin Glass Model with Nonlinear Interactions", `https://arxiv.org/pdf/1207.4887.pdf`, J. of the Physical Society of Japan **81**, 114008 (2012).

[117] J. Berges, Sz. Borsanyi, and C. Wetterich, "Prethermalization", `https://arxiv.org/pdf/1712.08533.pdf`, Phys. Rev. Lett. **93**, 142002 (2004).

[118] M. Gring, M. Kuhnert, T. Langen, T. Kitagawa, B. Rauer, M. Schreitl, I. Mazets, D. Adu Smith, E. Demler, and J. Schmiedmayer, "Relaxation and Pre-thermalization in an Isolated Quantum System", `https://arxiv.org/pdf/1112.0013.pdf`, Science **337**, 1318 (2012).

[119] F. H. L. Essler, S. Kehrein, S. R. Manmana, and N. J. Robinson, "Quench Dynamics in a Model with Tuneable Integrability Breaking", `https://arxiv.org/pdf/1311.4557.pdf`, Phys. Rev. B **89**, 165104 (2014).

[120] T. Kitagawa, A. Imambekov, J. Schmiedmayer, and E. Demler, "The dynamics and prethermalization of one dimensional quantum systems probed through the full distributions of quantum noise", `https://arxiv.org/pdf/1104.5631.pdf`, New J. Phys. **13**, 073018 (2011).

[121] M. Badabi, E. Demler, and M. Knap, "Far-from-equilibrium field theory of many-body quantum spin systems: Prethermalization and relaxation of spin spiral states in three dimensions", `https://arxiv.org/pdf/1504.05956.pdf`, Phys. Rev. X **5**, 041005 (2015)

[122] E. D. Zanotto and J. C. Mauro, "The glassy state of matter: Its definition and ultimate fate", J. of Non-Crystalline Solids **471**, 490-495 (2017).

[123] N. B. Weingartner, C. Pueblo, F. S. Nogueira, K. F. Kelton, and Z. Nussinov, "A Quantum Theory of the Glass Transition Suggests Universality Amongst Glass Formers", `https://arxiv.org/pdf/1512.04565.pdf` (2015); Nicholas B. Weingartner, Chris Pueblo, Flavio Nogueira, Kenneth F. Kelton, and Zohar Nussinov, "A phase space approach to supercooled liquids and a universal collapse of their viscosity", `https://arxiv.org/pdf/1611.03018.pdf` (2016), Frontiers in Materials **3**, 50, doi: 10.3389/fmats.2016.00050 (2016).

[124] Nicholas B. Weingartner, Chris Pueblo, K. F. Kelton, and Zohar Nussinov "Critical assessment of the equilibrium melting-based, energy distribution theory of supercooled liquids and application to jammed systems",

`https://arxiv.org/pdf/1512.04565.pdf` (2015).

[125] Since the energy density is bounded from below by its ground state value $\epsilon_{g.s.}$, the probability distribution $P(\epsilon')$ vanishes for $\epsilon' < \epsilon_{g.s.}$. The exact distribution $P(\epsilon')$ (whether a Gaussian or of another approximate form) is, of course, cut off at such low $\epsilon'$. The same holds for the Gaussian distribution describing the energy density of a large finite size system within the canonical ensemble.

[126] While it is natural to expect a continuous Gaussian distribution for supercooled fluids and glasses, the distribution $P(\epsilon')$ for plastically deformed crystals might be somewhat different. Since different spatial regions of an equilibrium crystal that has been cracked, etc., are locally similar to small finite volume patches of an equilibrium solid, the spatial overlap integral between the wavefunction describing the plastically deformed solid and the state of the equilibrium solid is anticipated to be finite. Thus, $P(\epsilon')$ may be a sum of delta-functions characterizing the different finite volume patches that emulate the finite equilibrium crystal.

[127] One may work backwards to extract an effective $\sigma_\epsilon$ needed to fit the experimental viscosity data when using the first equality in Eq. (56). One then finds that $\sigma_T \equiv \frac{T_{melt}-T}{\epsilon_{melt}-\epsilon}\sigma_\epsilon$ (which according to Eq. (55) is equal to $\overline{A}T$) exhibits larger deviations from a linear in $T$ near $T_{melt}$ than at temperatures far below $T_{melt}$ where a nearly perfect linear behavior appears. Such deviations from a nearly perfect linear increase of the effective $\sigma_T$ at lower temperatures are seen in, e.g., Figure 3 in [123] and Figures 16, 17, and S1 in [124]. In Figure 7 of the current work, the logarithm of the scaled dimensionless viscosities of all liquids must collapse onto the single ordinate $\log(\eta(T)/\eta(T_{melt})) = 0$ at $T = T_{melt}$. The smaller deviation from linear in $T$ behavior of the effective extracted $\sigma_T$ at lower temperatures is consistent with the better collapse at lower temperatures seen in Figure 7.

[128] L. Berthier and G. Biroli, "Theoretical perspective on the glass transition and amorphous materials", `https://arxiv.org/pdf/1011.2578.pdf`, Reviews of Modern Physics **83**, 587 (2011).

[129] Sans any approximations, the exact transition rates from specific initial $|\psi_i\rangle = \sum_s c_s(i)|\phi_s\rangle$ to final $|\psi_f\rangle = \sum_{s'} c_{s'}(f)|\phi_{s'}\rangle$ states are, rather trivially, given by $\mathcal{R}_{i\to f} = \frac{d}{dt}|\langle\psi_i|U_{pert}(t)|\psi_f\rangle|^2 = \sum_{\ell,\ell',s,s'}(c_\ell(i)c_{s'}(f)c_s^*(i)c_{\ell'}^*(f)\frac{d}{dt}(\langle\ell|U_{pert}(t)|\ell'\rangle^* \times \langle s|U_{pert}(t)|s'\rangle))$. The matrix elements of the evolution operator $U_{pert}$ containing the effects of general (arbitrary magnitude) perturbations are temperature-independent and are only determined by the Hamiltonian governing the system (describing a particular supercooled fluid) and the perturbation in question. By contrast, the amplitudes $\{c_s(i)\}$ and $\{c_{s'}(f)\}$ (that determine the distributions $P$ associated with the initial and final states) will generally vary with the energy densities of the initial and final states (or their associated temperatures). Indeed, from Eq. (55), the width of $P$ increases with the temperature (if, as assumed, $\overline{A}$ is kept constant) thus smearing the equilibrium result more significantly at elevated temperatures. The sum of transition rates to all possible final states orthogonal to the initial state can, similar to Eq. (48), be expressed as a weighted average of the equilibrium transition rates at

different energy densities [21].

[130] P. K. Dixon, "Specific-heat spectroscopy and dielectric susceptibility measurements of salol at the glass transition". Phys. Rev. B **42**, 8179 (1990).

[131] C. A. Angell, "Glass formation and glass transition in supercooled liquids, with insights from study of related phenomena in crystals", `https://arxiv.org/pdf/0712.4233.pdf`, J. of Non-Crystalline Solids **354**, 4703 (2008).

[132] Z. Nussinov, N. B. Weingartner, and F. S. Nogueira, "The 'glass transition' as a topological defect driven transition in a distribution of crystals and a prediction of a universal viscosity collapse" in "Topological Phase Transitions and New Developments", pp. 61-79 (online Sept. 2018), World Scientific.

[133] A comparison to experimental and earlier theoretical fits (see, e.g., Fig. 4 of [182]) for the heat capacity data yields a crossover temperature window $\Delta T_h$ (the width along the temperature axis of the "hysteresis" where the heating and cooling heat capacity curves differ) for different systems). Empirically, different heating (and cooling) rates yield different heat capacity curves yet these, too, tend to almost exactly coincide outside of the crossover temperature window of width $\Delta T_h$. For OTP, Salol (only experimental results for the heat capacity on heating are given in [182]), TNB (ditto), and Glycerol, the corresponding ratios $\frac{\Delta T_h}{T_g} \sim 0.08, 0.087$, and 0.13 (corresponding to $\Delta T_h \sim 20\text{K}, 20\text{K}$, and 25K). By comparison, the values of the scale factor $\overline{A}\sqrt{2}$ appearing in Eq. (56) for the fit of Figure 7 are, respectively, 0.070, 0.087, and 0.109 [123]. We caution that the general numerical coincidence of the scales is only suggestive ($\overline{A}$ does not vary dramatically across different fluids). Furthermore, here a comparison was only made for the three liquids (fluids examined in [182] that have, unambiguously, an identical composition to those whose viscosity was studied in [123]). The experimental heat capacity traces in [182] are from [183] (for OTP), [184] (for glycerol), and [185] (for Salol).

[134] E. B. Jones and V. Stevanovic, "The Glassy Solid as a Statistical Ensemble of Crystalline Microstates", `https://arxiv.org/pdf/1902.05939.pdf` (2019).

[135] Z. Nussinov, "Macroscopic Correlations in Non-Equilibrium Systems and their possible realizations", `https://arxiv.org/pdf/1710.06710.pdf` (2017).

[136] H. Sillescu, "Heterogeneity at the glass transition: a review", J. of Non-Crystalline. Solids **243**, 81 (1999).

[137] M. D. Ediger, "Spatially heterogeneous dynamics in supercooled liquids", Annual Review of Physical Chemistry **51**, 99 (2000).

[138] R. Richert, "Heterogeneous dynamics in liquids: fluctuations in space and time", J. of Physics: Condensed Matter **14**, R 703 (2002).

[139] W. Kob, C. Donati, S. J. Plimpton, P. H. Poole, and S. C. Glotzer, "Dynamical heterogeneities in a supercooled Lennard-Jones liquid", Phys. Rev. Lett. **79**, 2827 (1997).

[140] C. Donati, J. F. Douglas, W. Kob, S. J. Plimpton, P. H. Poole, and S. C. Glotzer, "Stringlike cooperative motion in a supercooled liquid", Phys. Rev. Lett. **80**, 2338 (1998).

[141] In this brief comment, we consider an initial state $|\psi\rangle = \sum_n c_n|\phi_n\rangle$ where $|\phi_n\rangle$ are eigenstates of a local

Hamiltonian $H = \sum_{i=1}^{N'} \mathcal{H}_i$ (where $H|\phi_n\rangle = E_n|\phi_n\rangle$). The system evolves with the time independent Hamiltonian $H$. We define the long time average of the local energy density fluctuations by

$$\overline{\sigma}_i \equiv \lim_{\tilde{T}\to\infty} \int_0^{\tilde{T}} dt (\langle \mathcal{H}_i(t)\rangle - \overline{\mathcal{H}}_i)^2, \qquad \text{(K10)}$$

where

$$\overline{\mathcal{H}}_i \equiv \frac{1}{\tilde{T}} \int_0^{\tilde{T}} dt\langle \overline{\mathcal{H}}_i(t)\rangle = \sum_n |c_n|^2 \langle\phi_n|\mathcal{H}_i|\phi_n\rangle \quad \text{(K11)}$$

is the long time average of $\mathcal{H}_i$. Here, $\langle\mathcal{H}_i(t)\rangle$ denotes the Heisenberg picture expectation value of the local energy term $\mathcal{H}_i(t)$ in the initial state $|\psi\rangle$. In the following, we consider the situation when level spacings of $H$ are relatively incommensurate (any equation of the type $(E_n - E_m) = (E_{m'} - E_{n'})$ may only be satisfied if $m' = n$ and $n' = m$). A simple calculation (along the lines of that performed in Ref. [21] for the long time average of general observables) then yields

$$\overline{\sigma}_i^2 = \sum_{n\neq m} |c_n|^2|c_m|^2|\langle\phi_n|\mathcal{H}_i|\phi_m\rangle|^2. \qquad \text{(K12)}$$

The fluctuations of the global energy density are

$$\sigma_\epsilon^2 = \frac{1}{N^2}\Big(\sum_n |c_n|^2 E_n^2 - \sum_{n,m} |c_n|^2|c_m|^2 E_n E_m\Big). \text{ (K13)}$$

Since, in any time independent Hamiltonian system, all eigenstates are stationary, Eqs. (K12, K13) must vanish when $|\psi\rangle$ is an eigenstate (as they indeed do). Trivially, a long time heterogeneity in the local energy (a finite $\overline{\sigma}_i$) can only appear in non-stationary states $|\psi\rangle$. This tautological relation may extend to an approximate general trend. Qualitatively, a larger spread in the global energy density $\sigma_\epsilon$ may appear hand in hand with a larger standard deviation of the local energy density $\overline{\sigma}_i$. This is natural from various viewpoints. The Schrodinger equation $H\psi(\{\vec{x}\}) = E\psi(\{\vec{x}\})$ holds for all (many body) spatial coordinates $\{\vec{x}\}$ with a global value of $E$. This, of course, holds true even if the Hamiltonian $H$ is a sum of local operators that do not connect the wave function at $\{\vec{x}\}$ to its values at spatially distant regions- i.e., operators that do not have off diagonal matrix elements connecting spatially far degrees of freedom. Thus, within any eigenstate, the expectation value of any projection of the Hamiltonian onto a local volume (i.e., the local energy density in systems with local interactions) must also be spatially uniform (i.e., independent of the spatial positioning of this local volume). Viewed from this perspective, this obvious single eigenstate relation suggests that larger deviations $\overline{\sigma}_i$ measuring spatial fluctuations of the local energy density may naturally appear in unison with a larger spread of the global energies in the spectral decomposition of $|\psi\rangle$ in the eigenbasis of $H$ (i.e., a larger standard deviation $\sigma_\epsilon$).

[142] Our approach suggests other possible consequences. For instance, since smaller systems may generally exhibit (already in equilibrium) larger fluctuations of their energy density, a corollary of our approach is that supercooling of small fluid droplets might be more easy to achieve than that of a macroscopic system.

[143] We allude to the standard rule of thumb in which quantization afforded by Planck's constant must be effectively introduced when counting microstates in classical integrals in phase space in order to obtain a dimensionless number. In basic classical statistical mechanics, the sum over microstates of a system of distinguishable

$N$ particles in $d$ spatial dimensions leads to the phase space integral contain a prefactor which involves a constant that (when compared to the exact quantum descriptions of ideal gases and other systems) is Planck's constant $h$,

$$\sum_n \cdots \rightarrow \frac{1}{h^{dN}} \int d^{dN}x \; d^{dN}p \cdots . \qquad \text{(K14)}$$

Similarly, for identical particles, there is an additional prefactor of $\frac{1}{N!}$ that needs to also be introduced by hand in the classical description (to avoid Gibbs' paradox and related behaviors). This celebrated fudge factor emulates, once again, quantum states of identical particles (reflecting their invariance under permutations).

[144] C. M. Varma, Z. Nussinov, W. van Saarloos, "Singular or Non-Fermi Liquids", https://arxiv.org/pdf/cond-mat/0103393.pdf, Physics Reports **361**, 267 (2002).

[145] A. A. Abrikosov, L. P. Gor'kov, and I. E. Dzyaloshinshki, "On the Application of Quantum-Field-Theory Methods to Problems of Quantum Statistics at Finite Temperatures", Soviet Physics JETP **36**, 636 (1959); A. A. Abrikosov, L. P. Gor'kov, and I. E. Dzyaloshinshki, "Methods of Quantum Field Theory in Statistical Physics" (Dover, 1975).

[146] E. M. Lifshitz and L. P. Pitaevskii, "Statistical Physics, Part 2" (Pergamon Press, 1980).

[147] P. Coleman, "Many Body Physics", Cambridge University Press (2015).

[148] J. T. Park, D. S. Inosov, Ch. Niedermayer, G. L. Sun, D. Haug, N. B. Christensen, R. Dinnebier, A. V. Boris, A. J. Drew, L. Schulz, T. Shapoval, U. Wolff, V. Neu, Xiaoping Yang, C. T. Lin, B. Keimer, and V. Hinkov, "Electronic Phase Separation in the Slightly Underdoped Iron Pnictide Superconductor $Ba_{1-x}K_xFe_2As_2$", https://arxiv.org/pdf/0811.2224.pdf, Phys. Rev. Lett. **102**, 117006 (2009).

[149] J. Zaanen and O. Gunnarson, "Charged magnetic domain lines and the magnetism of high-$T_c$ oxides", Phys. Rev. B **40**, 7391 (1989).

[150] K. Machida, "Magnetism in $La_2CuO_4$ based compounds", Physica C **158**, 192 (1989).

[151] H. J. Schulz, "Incommensurate antiferromagnetism in the two-dimensional Hubbard model", Phys. Rev. Lett. **64**, 1445 (1990).

[152] U. Low, V. J. Emery, K. Fabricius, and S. A. Kivelson, "Study of an Ising model with competing long- and short-range interactions", Phys. Rev. Lett. **72**, 1918 (1994).

[153] J. M. Tranquada, B. J. Sternlieb, J. D. Axe, Y. Nakamura, and S. Uchida, "Evidence for stripe correlations of spins and holes in copper oxide superconductors", Nature **375**, 561 (1995).

[154] K. H. Kim, N. Harrison, M. Jaime, G. S. Boebinger, and J. A. Mydosh, "Magnetic-Field-Induced Quantum Critical Point and Competing Order Parameters in $URu_2Si_2$", Phys. Rev. Lett. **91**, 256401 (2003).

[155] K. Izawa, Y. Nakajima, J. Goryo, Y. Matsuda, S. Osaki, H. Sugawara, H. Sato, P. Thalmeier, and K. Maki, "Multiple superconducting phases in new heavy fermion superconductor $PrOs_4Sb_{12}$", https://arxiv.org/pdf/cond-mat/0209553.pdf, Phys. Rev. Lett. **90**, 117001 (2003).

[156] M. B. Salamon and M. Jamie, "The physics of manganites: Structure and transport", Rev. Mod. Phys. **76**, 583 (2001).

[157] B. Simovic, M. Nicklas, P. C Hammel, M. Hucker, B. Buchner, and J. D. Thompson, "Interplay between freezing and superconductivity in the optimally doped under hydrostatic pressure", https://arxiv.org/pdf/cond-mat/0309193.pdf, Europhys. Lett. **66**, 722 (2004).

[158] T. Park, Z. Nussinov, K. R. Hazzard, V. A. Sidorov, A. V. Balatsky, J. L. Sarrao, S.-W. Cheong, M. F. Hundley, Jang-Sik Lee, Q. X. Jia, and J. D. Thompson, "Novel Dielectric Anomaly in the Hole-Doped $La_2Cu_{1-x}Li_xO_4$ and $La_{2-x}Sr_xNiO_4$ Insulators: Signature of an Electronic Glassy State", https://arxiv.org/pdf/cond-mat/0404446.pdf, Phys. Rev. Lett. **94**, 017002 (2005).

[159] C. Panagopoulos and V. Dobrosavljevic, "Self-generated electronic heterogeneity and quantum glassiness in the high-temperature superconductors", https://arxiv.org/pdf/cond-mat/0410111.pdf, Phys. Rev. B **72**, 014536 (2005).

[160] E. Dagotto, "Complexity in strongly correlated electronic systems", https://arxiv.org/pdf/cond-mat/0509041.pdf, Science **309**, 257 (2005).

[161] V. F. Mitrovic, M.-H. Julien, C. de Vaulx, M. Horvatic, C. Berthier, T. Suzuki, and K. Yamada, "Similar glassy features in the $^{139}$La NMR response of pure and disordered $La_{1.88}Sr_{0.12}CuO_4$, https://arxiv.org/pdf/0806.0207.pdf, Phys. Rev. B **78**, 014504 (2008).

[162] J. Schmalian and P. G. Wolynes, "Stripe glasses: self-generated randomness in a uniformly frustrated system", https://arxiv.org/pdf/cond-mat/0003267.pdf, Phys. Rev. Lett. **85**, 836 (2000).

[163] H. Westfahl, Jr., J. Schmalian, and P. G. Wolynes, "Self-generated randomness, defect wandering, and viscous flow in stripe glasses", https://arxiv.org/pdf/cond-mat/0102285.pdf, Phys. Rev. B **64**, 174203 (2001).

[164] G. C. Milward, M. J. Calderon, P. B. Littlewood, "Electronically soft phases in manganites", https://arxiv.org/pdf/cond-mat/0407727.pdf, Nature (London) **433**, 607 (2005).

[165] Z. Nussinov, I. Vekhter, A. V. Balatsky, "Nonuniform glassy electronic phases from competing local orders", https://arxiv.org/pdf/cond-mat/0409474.pdf, Phys. Rev. B **79**, 165122 (2009).

[166] J. C. Seamus Davis and Dung-Hai Lee, "Concepts relating magnetic interactions, intertwined electronic orders, and strongly correlated superconductivity", https://arxiv.org/pdf/1309.2719.pdf, Proceedings of the National Academy of Sciences **110**, 17623 (2013).

[167] E. Fradkin, S. A. Kivelson, and J. M. Tranquada, "Theory of Intertwined Orders in High Temperature Superconductors", https://arxiv.org/pdf/1407.4480.pdf, Reviews of Modern Physics **87**, 457 (2015).

[168] Qimiao Si, S. Rabello, K. Ingersent, and J. L. Smith, "Locally critical quantum phase transitions in strongly correlated metals", https://arxiv.org/pdf/cond-mat/0011477.pdf, Nature **413**, 804 (2001).

[169] S. I. Mirzaei, D. Stricker, J. N. Hancock, C. Berthod, A. Georges, E. van Heumen, M. K. Chan, Xudong Zhao, Yuan Li, M. Greven, N. Barisic, and D. van der Marel, "Spectroscopic evidence for Fermi liquid-like energy and temperature dependence of the relaxation

rate in the pseudogap phase of the cuprates", `https://arxiv.org/pdf/1207.6704.pdf`, Proceedings of the National Academy of Science (USA) **110**, 5774 (2013).

[170] Similarly, even for Fermi systems that have not experienced a change in their carrier density or other intensive parameters, general interactions my be replaced by Fermi bilinear by introducing auxiliary variables $\mathcal{Q}'$ and writing general averages in any subsystem as a weighted sum $\langle \mathcal{O} \rangle = \sum_{\mathcal{Q}'} P_{\mathcal{Q}'} \langle \mathcal{O} \rangle_{\mathcal{Q}'}$ (with, in the general case. weights $P_{\mathcal{Q}'}$ that are of non-uniform sign (a property related to the NP hard [172] "minus sign problem" of Quantum Monte Carlo)). Here, $\langle \mathcal{O} \rangle_{\mathcal{Q}'}$ denotes the average of any quantity $\mathcal{O}$ (including the pair correlator $G$) in a sector of fixed $\mathcal{Q}'$ for a *free fermionic system* [173–175]. If the auxiliary variables $\mathcal{Q}'$ are not translationally invariant (so that the resulting free Fermi system for a particular $\mathcal{Q}'$ realization is not diagonal in $k$-space), any discontinuity in $G_{coh}$ may be further diminished.

[171] P. Corboz, T. M. Rice, and M. Troyer, "Competing states in the t-J model: uniform d-wave state versus stripe state", `https://arxiv.org/pdf/1402.2859.pdf`, Phys. Rev. Lett. **113**, 046402 (2014).

[172] M. Troyer and U -J. Wiese, "Computational Complexity and Fundamental Limitations to Fermionic Quantum Monte Carlo Simulations", `https://arxiv.org/pdf/cond-mat/0408370.pdf`, Phys. Rev. Lett. **94**, 170201 (2005).

[173] R. Blankenbecler, D. J. Scalapio, and R. L. Sugar, "Monte Carlo calculations of coupled boson-fermion systems. I", Phys. Rev. D **24**, 2278 (1981).

[174] S. R. White, D. J. Scalapino, R. L. Sugar, E. Y. Loh, J. E. Gubernatis, and R. T. Scalettar, "Numerical study of the two-dimensional Hubbard model", Phys. Rev. B **40**, 506 (1989).

[175] T. Grover, "Entanglement of Interacting Fermions in Quantum Monte Carlo Calculations", `https://arxiv.org/pdf/1307.1486.pdf`, Phys. Rev. Lett **111**, 130402 (2013).

[176] G. Rigolin and G. Ortiz, "Degenerate Adiabatic Perturbation Theory: Foundations and Applications", `https://arxiv.org/pdf/1403.6132.pdf`, Phys. Rev. A **90**, 022104 (2014).

[177] C. M. Caves, C. A. Fuchs, and R. Schack, "Quantum probabilities as Bayesian probabilities", Phys. Rev. A **65**, 022305 (2002).

[178] H. D. Zeh, "On the interpretation of measurement in quantum theory", Foundations of Physics **1**, 69 (1970).

[179] W. H. Zurek, "Probabilities from Entanglement, Born's Rule $p_k = |\psi_k|^2$ from Envariance", `https://arxiv.org/pdf/quant-ph/0405161.pdf`, Phys. Rev. A **71**, 052105 (2005).

[180] Aharon Casher, Shmuel Nussinov, and Jeffrey Tollaksen, "Collapses and Avoiding Wave Function Spreading", `https://arxiv.org/pdf/1408.3097.pdf` (2014).

[181] Since the value of $r_L = \pm 1$ does not appear in $H_I$, there is an additive $\ln 2$ contribution to the entanglement entropy vis a vis the application of the formulas if these are applied for the partition of $(L_A + L_B - 1)$ decoupled Ising spins $\{r_i\}_{i=1}^{L-1}$ of energy $E = -J \sum_{i=1}^{L-1} r_i$ into two groups of $L_A$ and $(L_B - 1)$ spins whose energies are given by the spatially uniform Hamiltonians $H_{AI}$ and $H_{BI}$ respectively. This does not, of course, impact the asymptotic scaling of Eq. (F10).

[182] Aaron S. Keys, Juan P. Garrahan, and David Chandler, "Calorimetric glass transition explained by hierarchical dynamic facilitation", Proceedings of the National Academy of Sciences **110**, 4482 (2013).

[183] V. Velikov S. Borick, and C. A. Angel (2001) "The glass transition of water, based on hyperquenching experiments" Science **294**, 2335 (2001).

[184] L. M. Wang, V. Velikov, and C. A. Angell, "Direct determination of kinetic fragility indices of glassforming liquids by differential scanning calorimetry: Kinetic versus thermodynamic fragilities", J Chem Phys **117**, 10184 (2002).

[185] W. T. Laughlin and D. R. Uhlmann, "Viscous flow in simple organic liquids", J Phys Chem **76**, 2317 (1972).