# Peer review of "Infinite Range Correlations in Non-Equilibrium Systems and their possible realizations"

_SciPost Physics_

## Round 10 · Referee Report · Anonymous · 2018-12-23

Strengths

- A paper with an extraordinary claim: infinite-range correlations develop in any generic driven system

Weaknesses

- The paper is difficult to read. It covers many topics and various arguments are not only given in the main text and the appendices but also in several pages (!) of footnotes.
- The main claim of the paper is not justified and I believe that it is not correct

Report

The author has substantially modified the paper in response to my previous report.
Nevertheless, I recommend that the paper is rejected because I believe that its main result (the generation of infinite-ranged correlations for generic time-dependent Hamiltonians) is not sufficiently justified and likely to be wrong. Here I have to emphasize that I was not able to follow all the argument of the author. Below I will comment only on a selected issues:

1) The author now includes several discussions of the relation of his results to Lieb-Robinson bounds (which impose rigorous limits on the possibility to build up non-local correlations). I find these discussions highly contradictory. On the bottom of page 4 (and on page 10, subsection 3) the author correctly argues that a time t_min proportional to the system size is needed to build up any infinite-range correlations. This statement contradicts most of the paper, which finds infinite range correlations even for short times. As far as I can follow, the author argues that this is not a contradiction, as for times smaller t_min the average energy density cannot change. This is obviously not true: both by a uniform coupling to an (extensive) thermal bath and by using a translationally invariant time-dependent Hamiltonian one can easily change local operators and therefore also the energy density of a system in a finite time (this is obviously also possible if one considers many identical, but decoupled systems with vanishing Lieb-Robinson velocity). It is, however, not possible to built up infinite-range correlations! The topic is also discussed on p. 15, subsection 2, and the footnote [78], which seems to state again that the Lieb-Robinson light cone somehow (?) extents over the full system when long-range correlations develop and whenever the energy density changes with a finite rate.

2) As I have argued in my previous report, the example discussed by the author in Sec VI is a model which includes infinite-range correlations in its initial state and can therefore not be used to argue that "generically" infinite-range correlations build up for time-dependent systems. The reply seems to confirm my conclusion in this respect.

3) The author argues in Sec IX that his result follows for a system-bath version of his setup from the exact uncertainty relation, Eq (29). For his conclusions, the author needs to assume that Eq (37) holds, sigma_{\tilde H} = O(1).
He argues that this holds because the system somehow approaches a microcanonical ensemble due to equilibration which has less fluctuations than, e.g., a canonical ensemble where sigma_{\tilde H} = O(sqrt[N]). This is incorrect as is obvious from the fact that sigma_{\tilde H} is time-independent for a time-independent Hamiltonian (thus the system never approaches a microcanonical ensemble, only local observables equilibrate, not the global energy distribution). It also seems to contradict the main result of the paper. If one considers \tilde H has a time-dependent closed system the results of the paper suggest that sigma_{\tilde H} can even become of order N, rather than sqrt[N].

  • validity: poor
  • significance: low
  • originality: top
  • clarity: low
  • formatting: below threshold
  • grammar: good

Author Zohar Nussinov on 2019-03-24
(in reply to Report 1 on 2018-12-23)
Category:
remark

Dear Editor,

The items in this second referee report were mentioned in his/her first report in October 2018. These were addressed (in some detail) in my reply in December 2018 and in the resubmitted manuscript. Below I am answering anew these remarks. To avoid repetition, I am further referring to the earlier referee report and my original reply to it. In that earlier reply in December 2018, many of these aspects were discussed in some detail.

(1) The first comment in the current referee report was the first comment in his/her original report last fall. Specifically, it appeared in the second paragraph of that referee report from October 2018.

By Heisenberg's equation of motion, in order to have an energy density that varies at a finite rate, the average of the commutator [H(t),\tilde{H}(t)] (with H(t) the Heisenberg picture Hamiltonian of the system at time t and \tilde{H}(t) the Heisenberg picture Hamiltonian of the system augmented by its environment) must be extensive (i.e., be of order O(N) with N the number of particles in the system). Imagine (as the referee asks about) that the Schrodinger picture Hamiltonian H(t=0) is local, i.e., H = \sum_{R} H_{R} with the sum extending over O(N) local regions R. In order to have an energy density varying at a finite rate, it follows from Heisenberg's equation that the trace Tr(\tilde{\rho} [H(t),\tilde{H}(t)]) = O(N) with \tilde{\rho} the global density matrix. In other words, the average \frac{1}{N} \sum_{R} Tr(\tilde{\rho} [H_{R}(t), \tilde{H}(t)]) =O(1). If I understand the referee's comment, his/her claim is that the Lieb-Robinson bound does not allow this average commutator to be finite since all non-local commutators must vanish. That claim of the referee is somewhat puzzling. In any system having an energy density (or any other intensive quantity) changing at a finite rate, average commutators such as those above cannot vanish. The Lieb-Robinson light cone (i.e., the region in which the commutators are finite) must cover part of the system if the rate of change of the intensive quantities is non-vanishing. The Lieb-Robinson bound does not forbid finite commutators at long times and indeed what I discuss in the paper is at times long enough to allow the energy density to change at a finite rate.

See my reply to comment (1) from December 2018 for further details.

(2) This second comment in the current report is a compressed version of the second comment from his/her report five months ago. This comment constituted the third and fourth paragraphs in that report.

If the system has finite rates of change in related intensive quantities then the correlations are mandatory. The toy high dimensional many body example (which was constructed in order to be exactly solvable and thus, of course, special) indeed demonstrates how finite rates of change of the energy density appear in unison with long range correlations. I thought that readers might like to see exact results in a solvable model where they may intuit their way and check calculations very readily and see how the average energy density changes in tandem with the distribution of the energy density. The long range correlations in this toy model are unavoidable as I try to explain in great detail in the footnote and in the Appendix D in this example and are generic to driven systems. This system is a toy model. It does not preclude the rigorous bounds that I derive in general. Rather, it is consistent with them. This exactly solvable model in high dimensions is, of course, special precisely since it is exactly solvable.

For details concerning this example see comments (2-4) in my reply from December 2018 in which various aspects were addressed in some depth.

(3) This last comment in the current report of the referee mirrors and expands the last comment in his/her earlier report from October 2018. It appeared in the second half of the fifth paragraph in that referee report. I addressed that remark in December 2018 (item (6) of my reply last December).

A closed system that approaches equilibrium at long times, i.e., has long time averages that coincide with equilibrium averages may indeed have a long time average density matrix described by the microcanonical ensemble. The latter equilibrium average (the equilibrium average of a closed system with fixed energy, volume, and particle number) constitutes the textbook definition of the microcanonical ensemble. Semiclassically, the energy of any closed system has no uncertainty. Along similar lines, one can consider a closed system with a fixed number of particles (instead of energy) that is composed of a smaller subsystem and an environment. For a generic closed system, the fluctuation in any its components may be large but the fluctuations in the global conserved quantities may not be large. In the context of what my paper is about, if the full closed system (composed of the smaller subsystem and the environment) conserves the total energy, then the energy fluctuations in the small subsystem (H) can be large while those of the full system (described by \tilde{H}) can be small. This is also true in the quantum arena in various situations. One of these is when the closed system equilibrates at long time and is described by the microcanonical ensemble (assumption (3) in Section IX A). The referee questioned the derivation and wrote “... as is obvious from the fact that \sigma_{\tilde{H}} is time independent for a time independent Hamiltonian”. However, this statement of the referee is exactly what I already wrote in the paper!! Indeed, it is precisely for this reason that I used assumption (3) of Section IX A. The time independent \sigma_{\tildeH}} of a closed equilibrated system for which assumption (3) holds is of order unity and leads to one of the derivations of the result.

In the revised paper, I also already derived new inequalities for open systems (the systems that have \sqrt{N} fluctuations that the referee mentioned). This discussion of systems having \sqrt{N} fluctuations spans two pages and appeared in subsection IX B of the manuscript. The inequalities of subsection IX B are new and rigorously formalize time scales earlier motivated by the chaos bounds of Maldacena, Shenker, and Stanford.

Sincerely,
Zohar Nussinov

---

## Round 10 · Referee Report · Anonymous · 2019-5-12

Strengths

The topic of energy uncertainty in nonequilibrium dynamics is interesting and relevant to many experiments. The research on uncertainty relation in open quantum system is also very interesting.

Weaknesses

While the specific calculations in the paper may be technically sound, the paper seems to claim as the major result that infinite-range correlations can build up from a finite temperature equilibrium state in a finite time, which is very misleading.

Report

I was asked to resolve the impasse between the author and the second referee regarding the correctness of the main result of the paper.

I agree with the second referee that infinite-range (connected) correlations cannot develop in a short-range interacting lattice system at a finite time. This statement is of course assuming that the initial state of the system does not have infinite-range correlations. The proof of this statement follows straightforwardly from the Lieb-Robinson bound.

The calculations done by the author, however, are not necessarily wrong by this statement. Although it is impossible for me to check all the calculations, I tend to believe they are technically correct, particularly for the exact calculations in Eqs.(5-8).

The real problem is that the author made very vague and confusing statements of the calculations in the abstract and introduction that can be easily regarded as wrong. The calculations are all about the evolution of energy density uncertainty. It is indeed correct that the energy density uncertainty, initially vanishing in the thermodynamic limit for a finite temperature equilibrium state, can become finite at a finite time. But this statement is very different from the second referee’s statement I agreed above. Let me use a simple example to explain why:

Suppose at t=0, the Hamiltonian of the system is a 2D nearest-neighbor Ising model with no field (i.e. \$H=\sum_{i,j}S_i^z*S_j^z\$). The system is at a finite temperature below the well-known finite critical temperature. In the thermodynamic limit, the standard deviation (uncertainty) of the energy uncertainty vanishes. Now consider adding a magnetic field along z at any finite rate g=O(1), so \$H(t)=H(0)+g*t*\sum_{i}*S_i^z\$. The state of the system will remain invariant, but for any finite t=O(1), the energy density uncertainty is now finite. This is because the energy density uncertainty is now the energy density uncertainty at t=0 plus g*t times the connected correlation density \$\sum_{i,j}<S_i^z*S_j^z>/N^2\$, which we know is finite as the state of the system is ordered below the critical temperature. Note that I’m assuming no spontaneously symmetry breaking here so \$<S_i^z>=0\$ for any spin, which is fine for a finite system that will be taken to the thermodynamic limit later.

So what do we learn from the example? It is true that the energy uncertainty is made of connected correlations of local observables (for a local Hamiltonian), but when the system is put into non-equilibrium, the energy uncertainty is no longer made of the connected correlations of the SAME local observables. The only way to make the energy density changes from 0 to O(1) at a finite time is to have H(t>0) including local operators whose connected correlations are already infinitely ranged at t=0. In this case, the author’s finding is not surprising at all.

The author is trying to defend the validity of his results in wrong ways. Yes the Lieb-Robinson bound do not constrain signals beyond t=O(L) but the examples the author used to support the main argument are clearly about t=O(1).

The second referee also made a statement that energy density itself can be changed by O(1) in t=O(1). I believe this is correct (an obviously example is \$H(t)=(1+g*t)\sum_i S_i^z)\$. The author seemingly tried to disprove this statement in a comment which I don’t understand.

To summarize, while the calculations in the paper may well be technically correct, the main message the paper is trying to deliver is at best misleading, if not wrong. To make connected correlations in a system infinite-ranged at finite time, those connected correlations must be infinite-range at t=0 and there is simply no other way around. The author used the very vague term “intermediate times” in the abstract which makes the abstract not technically wrong but boring if intermediate time includes t=O(L).

The physical picture in Fig.1 is however correct, but as I explained, it cannot be interpreted as the buildup of infinite-range correlations (for the same local operators) over a rapid process.

In conclusion, the paper is worth of publication in some form but the author need to significantly revise the paper with a new theme based on my suggestions below to avoid a misleading message to the community.

Requested changes

My suggestion to the author is to give up the misleading story of creating infinite-range correlations over a rapid nonequilibrium process, but build upon the hefty calculations a rigorous story around Fig.1. The author has also done a lot of calculations for open quantum systems, but those do not contribute well to the current story. I believe many results in this paper (especially the part regarding open quantum systems) are useful for the community, and I’m sympathetic about the huge amount of effort the author has put into the writing and revision of the paper. However the author need to think about a different way to sell these results together in a coherent and non-misleading manner.

I do recommend a more careful discussion about the relation between finite energy density uncertainty and infinite-range correlations with reference to Lieb-Robinson bounds. Such discussion can be broadly interesting. For example, Hastings proved in PhysRevLett.93.126402 that for any fermionic systems, connected correlations of local fermionic operators will decay exponentially at any finite temperature. This means if the Hamiltonian at any time is made of local fermionic operators, then the energy density uncertainty has to vanish at any finite time. And this statement even hold for bosonic or spin Hamiltonians if the initial state temperature is above some value, as shown in PhysRevX.4.031019.

  • validity: good
  • significance: good
  • originality: good
  • clarity: low
  • formatting: good
  • grammar: excellent

Author Zohar Nussinov on 2019-06-02
(in reply to Report 2 on 2019-05-12)
Category:
question

I am grateful to the last referee (#3) for his/her comments and help. At this end, any changes that may further help the paper become more complete will be greatly welcomed. I hold very high regard for this new referee and genuinely appreciate his/her aim to help. However, I was not sure what to do upon resubmission. This is the reason for the below query. The current paper is long and its contents cannot be reduced to a simple extension of known results making it extremely easy to miss certain points. It should be stressed that this query should not be misconstrued as a critique of any sort but rather as a list of items that were not clear to me. At this end, significant time has been spent on this problem and a critical investigation of many aspects related to it. After all of these efforts, I do not wish to leave anything imprecise nor unclear. Prior to resubmitting a revised draft encompassing the possible changes highlighted in bullet form below, I want to make certain that I correctly understand the specific comments/requests of the new referee and his/her desired modifications to the manuscript- especially since some items suggest several simple unintentional natural oversights. This paper has already suffered a very long delay by now (since 2017) and I do not wish to risk yet another delay and a negative outcome due to a simple misunderstanding. Before listing the viable changes prompted by the last referee's comments, I must ask central questions concerning examples/comments that formed the backbone of his/her report.

Question:

(1) The new referee provided a specific example that aimed, with very good intentions (whose spirit was well appreciated at this end), to cogently illustrate that the thesis of my article is "misleading". I read this example quite a few times by now and, frankly, it was not very clear to me what this example illustrates and what I am expected to do as a result. As explained below, this example of the referee seems, on its own, to be somewhat wrong (see aspects (i) and (ii) below). It should be stressed that I do not, at all, fault the referee for small mistakes on his/her part if any occurred (nor do I wish to appear as personally criticizing the second referee for a simple error (see (3) below)) and other omissions. Rather, I need to be certain that nothing was overlooked at my end regarding all items that were mentioned in any referee report thus far. As an author, I want this work (on which much time was spent) to be as precise as possible. If desired, in the paper draft, I will try to note my own reading of this example and far more general variations of it- all of which lead to conclusions that are diametrically opposite to those reached by the referee.

$\bullet$ First let me focus on the specific example of the new referee which was intended to pedagogically illustrate his/her central point.

(i) For a 2D Ising model, the sum of the connected pair correlation functions over all site pairs

$\sum_{i,j} \langle (S_i^z - m) (S_j^z - m) \rangle$

at all temperature apart from the critical temperature, $T<T_{c}$) scales with the number of sites $N=L \times L$- not as $N^2$. More precisely, for the large finite $N$ rendition that the referee had in mind, the above sum asymptotically scales as

$N \xi^{2}$

with $\xi$ being the correlation length. In the above, $m$ is the average magnetization $m \equiv\langle S_i^z \rangle$ which, as the referee noted, vanishes for a finite size system. Even at criticality ($T=T_{c}$), the ratio of the above double sum to $N^2$ will decrease as $N$ increases tending to zero for macroscopic systems as

$L^{-\eta}=N^{-\eta/2}$

with

$\eta = 1/4 $ being the anomalous exponent of the 2D Ising model. Thus, the energy density variance as the referee seems to define it (see item (ii) for the way it is defined in the paper) will indeed vanish for large $N$ decaying to zero (as $1/N$ for general temperatures and at criticality as $N^{-1/8}$). **This result is at odds with my understanding of what the referee states for this example- namely that the energy density uncertainty (as he/she defines it) will be finite.**

(ii) Moreover, in the first place, under the conditions specified in the paper, examples such as the above (including another provided by the last referee) need not lead to macroscopic connected correlations nor, equivalently, to fluctuations of the energy density.

Specifically, the statement being proven in the paper for a closed system and bath hybrid is the following:

If the expectation value of the original system Hamiltonian (i.e., that at time $t=0$) changes in the time evolved state then the energy density fluctuations associated with that initial Hamiltonian are finite. Similar results apply to all other intensive quantities.

This result was proven in the paper (Section IX) for a time independent Hamiltonian for the combined system plus bath; an integration over the bath leads to a time independent Hamiltonian for the system alone. This is what transpires in real physical systems wherein the form of all fundamental interaction terms is time independent; time dependent Hamiltonians may arise when tracing the latter exact independent Hamiltonians over a spatial subvolume or bath. (Effective time dependent Hamiltonians were examined in the paper in many other Sections.)

** For the 2D Ising example, and others of the same general type with commuting terms (e.g., the referee's remark about "$H(t)=(1+g*t)\sum_i S_i^z$"), an equilibrated system (i.e., one having its initial density matrix depend only on the initial Hamiltonian) **does not** change its $t=0$ state as a consequence of adding, at times $t>0$, a term that commutes with the initial Hamiltonian**. In other words, the uncertainty of the energy density as evaluated with the initial Hamiltonian in the time evolved state is ***identically zero*** (since the state of the system does not change).

Stated equivalently and more formally, if the the probability density matrix $\rho$ at time $t=0$ is a function of the initial Hamiltonian, $\rho = f(H)$ (e.g., a Boltzmann distribution or any other that depends on the initial Hamiltonian $H$ alone) then a consequent evolution with any Hamiltonian $H(t)$ that commutes with the initial Hamiltonian will leave the probability density matrix invariant,

$\rho(t) = {\cal{U}}(t) \rho {\cal{U}}^{\dagger}(t) = \rho$,

with the evolution operator

${\cal{U}}(t) \equiv {\cal{T}} e^{-\frac{i}{\hbar} \int_{0}^{t} H(t') dt'}$,

where ${\cal{T}}$ denotes time ordering and $[H(t'), H]=0$.

In particular, all expectation values of general quantities $Q$ (including those associated with variances) will not change in time,

$Tr( \rho(t) Q) = Tr (\rho Q)$.

Specifically, if the standard deviation of the energy density in the initial equilibrium state vanishes then it will remain zero at all times under such an evolution. (This may be proven by applying the above equality for both $Q=\frac{H}{N}$ and $Q=\frac{H^2}{N^2}$.)

Thus, the 2D Ising model example of the last referee and others like it do not satisfy the conditions discussed in my work for the extensive macroscopic correlations (item (ii)). Accordingly, it indeed does need to display these extensive fluctuations (item (i)) as is consistent with what this specific example yields and more generally the rigorous result proven in the paper.

The inequalities of Section IX proven for a closed system plus bath (nor the other inequalities derived in that Section along nearly identical lines for open system and bath hybrids that the referee was positive about) follow from the non commutativity of the system and bath Hamiltonians (this non commutativity enables the system to change its energy density with time).

Question:

(2) The principal message of the paper is centered around the physics summarized in Figure 1 that the new referee suggests to emphasize. Thus, that advice (made, once again, with kind helpful intentions in mind) was similarly quite confusing to me. If possible, I prefer not to rewrite the nearly forty page paper on this same core principle.

Question:

(3) I now turn to one of the simple possible mistakes of referee #2 (concerning the Lieb-Robinson (LR) bounds) which seems to have triggered the exceptionally long impasse. This viable error of the second referee was already noted in a remark that I added earlier (as well as in the paper itself [78]) yet may have been easily overlooked. To hopefully avoid any misunderstanding and/or oversights of mine and the others, I am repeating it anew here (with minor variations so as to be consistent with the notation/abbreviations employed in the current query). **I will be grateful if the third referee can note if there are any omissions on my part regarding LR bounds in the below.** (Employing the word "obvious" in either sense (positive or negative) is fine- I just want to be certain that all stated in the paper is correct or needs a correction.) This concern of referee #2 seems to have been his/her central more detailed objection.

"The Schrodinger picture Hamiltonian $\tilde{H}(t=0)$ (i.e., Heisenberg picture Hamiltonian at time $t=0$) of the combined system (${\cal S}$) + environment (${\cal E}$) hybrid may be expressed (similar to the paper) as $\tilde{H} = H + H_{{\cal S}-{\cal{E}}} + H_{{\cal{E}}}$ where $H$ is the system Hamiltonian, $H_{{\cal S}-{\cal E}}$ denotes the coupling of the system to its environment, and $H_{{\cal{E}}}$ is the Hamiltonian of the environment. The commutator

$\frac{1}{N} \sum_{R} Tr(\tilde{\rho} [H_{R}(t), \tilde{H}(t)]) = \frac{1}{N} \sum_{R} Tr(\tilde{\rho} [H_{R}(t), H(t) + H_{{\cal S}-{\cal{E}}}(t) + H_{{\cal{E}}}(t)]) = \frac{1}{N} \sum_{R} Tr(\tilde{\rho} [H_{R}(t), H_{{\cal S}-{\cal{E}}}(t) + H_{{\cal{E}}}(t)])$

(since $H(t) = \sum_{R} H_{R}(t)$ trivially commutes with itself). Here, $\tilde{\rho}$ denotes the density matrix of the system-environment hybrid. For all times $t>0$, the norm of the above commutator average

$\frac{1}{N} | \sum_{R} Tr(\tilde{\rho} [H_{R}(t), H_{{\cal S}-{\cal{E}}}(t) + H_{{\cal{E}}}(t)]) | = \frac{1}{N} | \sum_{R,R'} Tr(\tilde{\rho} [H_{R}(t), H_{R'}(t)])| \le \frac{1}{N} \sum_{R,R'} c' \exp(-(a|R-R'|-v_{LR}t)).$

The decomposition of the system Hamiltonian $H = \sum_{R} H_{R}$ into a sum over local regions spans no more than $N$ terms- the number of sites in the system. In the above, $c'$ is a constant, and $a$ and $v_{LR}$ denote the LR decay constant and speed respectively. (Rather explicitly, to arrive at the last inequality, we invoked $ |Tr(\tilde{\rho} [H_{R}(t), H_{R'}(t)])| \le || [H_{R}(t), H_{R'}(t)]||$, where $||~\cdot~||$ denotes an operator norm, along with the LR bounds $ || [H_{R}(t), H_{R'}(t)]|| \le c' \exp(-(a|R-R'|-v_{LR}t))$.)

For each $R \in {\cal S}$, there is a minimum distance $D(R)$ between $R$ and the surrounding environment. For any such $R$, we may bound (from above) the sum over all $R' \in {\cal{E}}$ of the exponential $e^{-(a|R-R'|-v_{LR} t)}$ by a sum of this exponential over the larger domain external to a sphere of radius $D(R)$ around $R$ (such a volume contains ${\cal{E}}$ as a subset). For sufficiently short times $t$, the sum

$\frac{c'}{N} \sum_{R,R'} e^{-(a|R-R'|-v_{LR} t)}$

vanishes as $N \to \infty$ (since the minimal distance $D(R)$ of a typical $R \in S$ to its surrounding environment is of the order of the system length (thus diverging in the thermodynamic limit)); for vanishingly small times, the sum of $ e^{-(a|R-R'|-v_{LR} t)}$ over such a larger domain of $R'$ values with $|R'-R| \ge D(R)$ decays exponentially in $D(R)$. Putting all of the pieces together, we see that the LR bounds imply that at vanishingly short times, $\frac{1}{N} |\sum_{R} Tr(\tilde{\rho} [H_{R}(t), \tilde{H}(t)]|$ is bounded from above by a function that is exponentially small in the length of the system, tending to zero in the thermodynamic limit. In other words, the energy density cannot change at a finite rate at sufficiently small times.

Contrary to the above LR bounds, the second referee argued that a finite minimal time cannot possibly be required in order to change the energy density of the system at a finite rate. Specifically, the referee wrote a sentence on several inter-related scenarios: ``This is obviously not true: both by a uniform coupling to an (extensive) thermal bath and by using a translationally invariant time-dependent Hamiltonian one can easily change local operators and therefore also the energy density of a system in a finite time (this is obviously also possible if one considers many identical, but decoupled systems with vanishing Lieb-Robinson velocity)."

Suggestion (i)- that of ``uniform coupling to an extensive bath" precisely involves bulk or "infinite" range interactions (the external bath couples to local operators in the system bulk- a distance (similar to the above $D(R)$) that diverges in the thermodynamic limit). This is exactly what we elaborated on in the above bounds.

Suggestion (ii) violates an assumption of a time independent $\tilde{H}$. The fundamental interaction terms in $\tilde{H}$ for the system-environment hybrid are (per our assumption) time independent. Any statements made about a time dependent system Hamiltonian $H(t)$ formed by tracing over the environment cannot violate those underlying $\tilde{H}$.

Suggestion (iii) is that of many uncorrelated systems having vanishing Lieb Robinson speeds $v_{LR}$. If such a scenario materializes then the density matrix will admit a tensorial product structure with no long range correlations. Indeed, precisely such a situation was already discussed in detail in an entire Section of the paper (Section V).

(The ensuing results are consistent with the two exact bounds derived in Sections VII and IX of the manuscript. In the context of Section IX, when the density matrix is a tensorial product of $O(N)$ decoupled systems, the uncertainty of the system energy ($\sigma_{H}$) will scale as $N^{1/2}$. Such a scaling will also hold (as a lower bound) for the uncertainty of the Hamiltonian of the combined system-environment hybrid if each of the $O(N)$ subsystems couples to its own separate bath (i.e., $\sigma_{\tilde{H}} \ge O(N^{1/2}))$.)

Regardless of what scenario one wishes to examine, a finite average commutator $\frac{1}{N} \sum_{R} Tr(\tilde{\rho} [H_{R}(t), \tilde{H}(t)]) ={\cal{O}}(1)$ for a bath separated by large distance from a local region in the system bulk can (as explained earlier and above, by the LR bounds) only arise at sufficiently long times if all of the interactions are local."

The next listed items (4-10) ***do not concern possible omissions/inconsistencies (more precisely, things that were not clear to me) in the report(s)*** but rather suggested changes to the text that I wish to be certain will suffice for publishing the paper.

(4) Rereading all of the referee reports in sequence with the added benefit of hindsight, I get the sense that what triggered the long delay after the first referee's recommendation in the Spring of 2018 may have simply been a rather been a quick reflex on the part of the second referee. I suspect that, sometime last year or before, when the second referee saw the word "infinite" in the title, he/she was immediately certain that whatever is written thereafter must be, in his/her own words, “not possible!”. With minor variations, this line of deep conviction of the second referee reappears in his/her reports. The calculations were essentially deemed correct but that was then effectively said to be irrelevant since infinite range correlations are “not possible”. The non-optimal choice of the title might have cost this paper a year long delay. The second referee’s reaction and repeated questions also prompted me to introduce text reviewing LR bounds in order to explain that no problems arise. I do not blame the second referee for his initial reaction last year due to my poor choice of the title and took his/her initial first report to heart.

The new (third) referee was concerned that (although, similar to the second referee, judged all calculations to seem technically correct) a general reader might be "misled”. Prompted by this somewhat similar reaction concerning the opening sections of the paper, I will indeed aim to change the beginning of the paper so as to avoid a misconception from its very start about what is proven. Towards that end,

$\bullet$ The word "infinite" in the title can thus be changed to "long" or perhaps a more appropriate specific adjective such as "macroscopic". Related changes can be made in the main text of the paper. What was meant by "infinite range" was already discussed earlier in some detail in the main text but can be sharpened in a revised version.

(5) To hopefully avoid any further confusion of this sort, several detailed explanations can be explicitly repeated throughout the text. Specifically,

$\bullet$ I can furthermore explicitly write in the Abstract and Introduction that the paper demonstrates that In realistic driven physical systems of linear dimensions $L$, after a time scale of order $t_{\min}$, correlations can exist everywhere in the system when the system starts from equilibrium having *no connected long range correlations* at $t=0$. Here, $t_{\min}$ is the time below which the energy density cannot change at a finite rate. (As explained in the earlier version of the paper and explicitly highlighted in the earlier reply to the second referee, a trivial calculation illustrates that the LR bounds imply that $t_{\min} \ge L/v_{LR}$ with $v_{LR}$ being a relevant LR type speed.)

(6) In the paper text, I should perhaps reiterate the above once again. Namely, I should state that my results indeed do not violate the LR bounds on commutators nor their powerful implications for correlations (which invoke and lucidly build on the commutator LR bounds and thus automatically are constrained by the same time scales). In fact, as just repeated above (3), both in the revised version that was resubmitted last year and in the reply/further comments to the second referee on his/her report several months ago, I used the LR bounds to illustrate that indeed the energy density cannot change at a finite rate on times shorter than $L/v_{LR}$ (i.e., $t \ge t_{\min}$). It is on this time (one larger than $t \ge t_{\min}\ge L/v_{LR}$) that the paper demonstrated, via a variety of calculations both very general rigorous results and dual specific yet exactly solvable toy models (that for the spin model was also solved in two different ways that led to the same result) that long range correlations may appear in unison with a finite rate of the variation of the energy density. I tried to be very cautious in all equations and statements.

$\bullet$ In the text of the paper, I can further repeat this yet again at the very beginning of the paper. I can underscore in the Introduction that the dual toy examples and the far more general proofs provided in the paper were for times larger than $t_{\min} \ge L/v$ with $v$ the relevant cutoff speed (an LR like speed $v_{LR}$ for non-relativistic systems).

The above was already noted in the main paper and also belabored in an appendix (Appendix D).

$\bullet$ The simple toy models of Section VI allow for a finite rate of variation of the energy density already at $t=0$. Here, the clock was set such that any finite time $t>0$ already corresponds to time after a vanishingly small $t_{\min}$. To avoid any confusion, I can make explicit in the paper (complementing and re-enforcing the already existing text in subsection VI3) that a finite, spatially uniform, coupling of the form of Eq. (7) is possible for times $t \ge t_{\min}$. The calculations provided in Section VI3 are for $t\gg t_{\min}$.

(7) The focus of my work does not concern LR bounds but rather on all time scales $t\ge t_{\min}$ that are larger than these (time scales in which a finite rate of variation of the energy density or other intensive quantities occurs); my results are, of course, trivially consistent with the LR bounds (and thus would indeed be obviously correct or "boring" if examined only through these lens alone). Since $t_{\min}$ can be finite and very short, the physics at times $t \ge t_{\min}$ is not irrelevant to experiment. Thus to summarize,

$\bullet$ In the revised Abstract and Introduction, I can further stress that $t_{\min}$ as well as the LR time $L/v$ are not irrelevant divergent time scales but can be rather short in actual experiments.

(8) To make the above lucid, I can provide in the main text of the paper ***simple physical orders of magnitude estimates***. If, e.g., $L$ is the order of 1cm for a macroscopic sample and, the relevant velocity $v=c$ is a typical radiation speed (as in, radiative cooling or heating) then the time scale $L/v$ will be exceedingly short

$\sim 3 \times 10^{-11}$ seconds

for an index of refraction $\sim 1 $). For supercooled liquids, the experiments themselves are done on liquids that are cooled at a rapid finite rate (an occurrence only possible for times $t \ge t_{\min}$). Thus, one does not need to worry about the LR bounds the experimental time scale $t\ge t_{\min}$ automatically exceeds these the original LR bound. The cooling rates of common supercooled silicate fluids and metallic glass formers differ by many decades. However, the cooling rate in all of these fluids cannot exceed the system size divided by the speed of light or relevant non-relativistic LR type speed. In metallic liquids (that form glasses when supercooled), often in experiments one uses (radiative) laser beam heating. In typical metals, both the heat and charge effectively travel at a finite fraction (typically of the order of $10^{-2}$) of the speed of light $c$ the effective speed of electrons in a metal;. both effective heat and charge transport velocities are possibly the same in conventional metals obeying the Wiedemann-Franz law). The speed/rate of heat transfer is bounded from below by the speed/rates of any of the individual (radiative/conduction/convection) processes that contribute to it. Thus, if either the typical radiative or conductive processes occur at speeds associated with a finite fraction of the speed of light $c$ then so, too, is the total heat transfer. In metals as well as in systems where the radiative penetration depth is larger than or of the scale of the linear dimension of the material, the speed associated with heat transfer is rather large and, correspondingly, the LR like time scale can become very short. In numerous systems (including many glass formers), the empirical required minimal time $t_{\min}$ for the system change its energy density at a finite rate can be large.

These considerations may also be formally and rather trivially rationalized from the continuity equation applied to the energy density,

$\partial_{t} \epsilon + \vec{\nabla} \cdot \vec{j}_{q} =0$.

This equation implies that for an incompressible system a finite rate of change of the energy density is possible at times

$t \ge t_{\min}\ge \ell/v_{q}$.

Here, $\ell$ is the distance over which $\vec{j}_{q}$ has finite gradients and $v_{q}$ is the speed associated with the energy (heat) flux (current). Consistent with item (3), the above equation suggests that a finite minimal time $t_{\min}$ must elapse before the system can change its energy density. In the simplest settings (e.g., a spherical geometry with radial currents similar to that employed in several experiments, $\ell$ is of the scale of the linear dimension or radius of the system). The point is that in realistic macroscopic systems, the "LR" like time scale $\ell/v_{q}$ may be finite (and in many systems the minimal time can become exceedingly short).

One of my prime interests when I started working on this problem years ago was that of supercooled liquids where these time scales are very small on experimental time scales and the cooling was performed at a finite rate (for which, automatically, $t \ge t_{\min}$). These systems start from equilibrium at $t=0$ with no long range connected correlations and then are supercooled to temperature below equilibrium melting. A principal aim of the paper is to illustrate that these liquids may develop an energy density uncertainty in an extremely short time that leads to a prediction for the viscosity that is **consistent with viscosity measurements over 16 decades for all known supercooled liquids**. This result is not, at all, that "boring". In fact, until my work there was no universal expression for the viscosity on all time scales for the viscosity of all supercooled liquids that did not introduce additional ad hoc special non equilibrium temperature scales. (The approach described in Section XII relies solely on the equilibrium transition temperature and does not invoke the existence of any other putative theorized temperatures.)

Thus, with the above in mind,

$\bullet$ I will briefly note the above simple numerical example concerning order of magnitude estimate in a brief sentence in the main text or short appendix.

(9) Judging from the new referee's remark, my earlier reply to the second referee regarding the absence of general time dependence (that seems to have not been written well; I will try to write it anew in the main text). Indeed,

$\bullet$ To avoid any misinterpretation regarding the latter, I can emphasize in the main text the systems that I discuss are those that can be described by time independent Hamiltonians when combined with their baths.

Lastly,

(10)

$\bullet$ I will indeed note the two additional references to the LR bounds in finite temperature equilibrium systems. Perhaps no less important to resolving the misunderstanding of Referee #2 is item (3) above.

Once again, I wish to thank this last referee for his/her suggestions! His/her remarks and advice were written with the best intentions in mind. I will aim to heed these and will update the paper draft. However, as I wrote above, prior to resubmitting the paper to Scipost, I wanted to make certain that indeed invoking the above changes is what he/she had in mind. Some items had me confused prompting this query letter.

This paper has been delayed since 2017. I do not fault anyone but myself for a bad choice of words for the title for that. Nonetheless, I do wish to get this work out if all of its calculations and the description of these computations are correct. (Furthermore, in the past few months, there were preprints that reported on broad distributions similar to those in Figure 1- far after my submission). I will start to incorporate the above changes to the manuscript and will heed the referee's reply and comments. In the manuscript, I aimed to be extremely careful in the calculations and text describing them and their viable extensions. I am very grateful to the last referee for his/her helpful comments and must, once again, genuinely apologize in advance for any oversights on my part if I misunderstood what he/she wrote and/or overlooked anything else.

Thank you very much,
Zohar Nussinov

Author Zohar Nussinov on 2019-06-07
(in reply to Report 2 on 2019-05-12)
Category:
correction

---

## Round 10 · Author Response

Dear Scipost Editor,

I wish to thank you for the handling of my manuscript and for the new referee for his/her thoughtful comments. In the revised manuscript, these are addressed in great detail. In what follows, I list all comments of the referee followed by a brief response.

Sincerely,
Zohar Nussinov
* * *
1) Referee "The paper makes a surprising claim: the author argues that rapid changes of a local, bouded Hamiltonian generically induce infinite-range correlations in a finite time interval. This seems to contradict numerous other studies on driven system. Most notably, it seems to contradict the famous Lieb-Robinson bounds which set rigorous limits on the speed with which information can spread. The relation to known results on the spread of correlations is not discussed in the paper."

Author: I wish to thank the referee for encouraging me to emphasize once again that, in order to induce a finite rate of change of the energy density, the coupling between the external drive and the system must be extensive. This was already noted in the earlier version. However, not much emphasis was placed on it nor did I explicitly cite the Lieb-Robison paper (although I did already cite in the earlier version that the referee reviewed the works by Hastings, Koma, Eisert, and Kastoryano that related the Lieb-Robinson bounds to the decay or correlations). In the revised version, we now discuss this once again and in far more detail in the second half of Section IV (starting from the paragraph before Eq. (3)), and in subsections VI A3, and IXA2. These comments concern both the original Lieb-Robinson bound (a bound on commutators not on correlation functions) by explaining that the commutators associated with the terms in the Hamiltonian cannot be local in order to ensure a change of the energy density as well as the implications of the Lieb-Robinson bounds on correlation functions as investigated by the earlier cited works Hastings and others. I now further cite additional works. In the new added text, an emphasis is placed in explaining that there is no violation of the causality implied by the Lieb-Robinson bound.
* * *
2) Referee: "The strongest argument of the author is an exact calculation where the author shows that a time-dependent uniform magnetic field induces infinite-ranged correlations in the energy of a spin system."

Author: The toy model of Section VI A is an exactly solvable toy example; it is not the strongest result of the paper. In the revised paper, this is now further emphasized by explicitly highlighting the central results of Eqs. (21, 29, 38, 40, 41). These general results (including the two corollaries of Eqs. (21) and (41)) are now made more lucid. We further discuss other extensions in the new Sections VIII and XI. I also explain how the results of the spin model further imply similar results elsewhere (e.g., excitations in anharmonic solids (item (c) of Section VIB).
* * *
3) Referee: "As far as I can tell, the calculation presented by the author in Sec. VI is technically correct."

Author: As an author, it is crucial for me to make certain that all of the results are correct. I appreciate the judgement of the referee regarding the calculation. To verify this once again for myself, apart from the original derivation, I now also rederive the result using a semi-classical calculation (Section VIA2). The final result of Eq. (14) coincides with the earlier result of Eq. (11). This approach also underscores a geometrical interpretation of the effect and its presence in semi-classical systems.
* * *
4) Referee: "The paper fails, however, to discuss that the system under consideration had already infinite-ranged correlation in its initial state (see below). If the initial state has infinite range correlation it is not surprising that also the finial state (after some time evolution) has such correlations (possibly in a different variable). Obviously, one cannot use such an example to argue that a time evolution generically generates infinite-ranged correlations ..."

Author: These correlations are not terribly surprising and, in fact, are rather generic. As the referee notes, for the specific model example of Section VI, these correlations were already implicit in the earlier version although I indeed have not explicitly noted them. Far more generally, in a real physical system, a bulk coupling induces long range correlations (see Eq. (3) of the revised manuscript). With regard to the specific example of Section VIA, in footnote [66] of the revised manuscript, I explain that if not only the rate of the expectation value of the rate of change of the energy density is finite (all the systems that we study in the current work have such a finite rate) but also the expectation values of the higher moments of the rate of energy density change are finite then the correlations that the referee notes are mandatory. In a new Appendix (D), I outline a gedanken experiment in which the spin model will be realized.
* * *
5) Referee: "Besides the example, the paper gives more arguments. In Sec. VII a Magnus expansion is used and the author claims that "higher order terms do not cancel identically" but I was not able to finite a proof/plausible argument for that."

Author: I am thankful to this particular comment of the referee. In the revised manuscript, the calculations are streamlined and detailed anew to explain (see Eq. (21)) that generally a finite standard deviation is rather generic. I explain, once again, that for product states, the standard deviation of the energy density vanishes. However, for general states, a finite standard deviation arises.
* * *
6) Referee: "In Sec. VIII uses an uncertainty relation in combination with Eq. (20) which claims that the fluctuations of the macroscopic total energy of the system are of order 1 because "the microscopical ensemble may be invoked". I do not understand the argument and it is certainly not sufficient to substantiate the very strong claim of the paper."

Author: In the revised manuscript, I took this referee's comment to heart and expanded in very elaborate detail on the rigorous consequences of the new uncertainty relation inequalities. In the earlier version (as well as the current version), I invoked the "microcanonical" ensemble when discussing closed energy conserving systems. Contrary to what the referee wrote, I did not "claim" anything concerning a "microscopical ensemble" (there is no such ensemble and I assume that the referee simply unintentionally mistyped). In the revised manuscript, I added references to the microcanonical ensemble. I also explain that even without such an assumption, divergent long distance correlations follow as a rigorous consequence of the uncertainty relations (Eq. (32)). In the revised manuscript, I further elaborate and explain how, the order of magnitude, of my uncertainty bounds for open systems (Eq. (40)) coincide with suggested by Maldacena, Shenker, and Stanford for Lyupanov exponents in open thermal systems and those of others.
* * *
7) Referee: "In conclusion, I recommend that the paper is rejected."

Author: In the revised manuscript, I expound on the items that the referee noted and related others to make certain that all earlier and alternate derivations may be readily reproduced and checked by readers.

In conclusion, the paper greatly profited from the comments of both referees. I must indeed thank this last referee for his/her time and critical comments. This new set of remarks forced me to expand on additional aspects that significantly strengthened the work and further underscored the generality of its results. I hope that these new (and rather detailed) results and ideas will not be "rejected".

---

## Round 10 · List of Changes

(1) More references concerning correlations and in particular those relating to the Lieb-Robinson bounds are cited.

(2) A new (second half) of Section IV is added to explain the relation between my results and causality (including the Lieb-Robinson bounds) with a more general account of how effective long range interactions are induced.

(3) A new subsection (Section VIA2) was inserted to explain how the quantum calculation (in the earlier version) coincides with a semi-classical one. This subsection contains two new figures.

(4) A new subsection (Section VIA3) was appended to discuss the relation between the spin model of Section VIA and causality.

(5) At the very end of Section VIB, a new paragraph (c) was added to explain that the spin model results further allow for analogous results for excitations in anharmonic solids.

(6) In Section VII, Eq. (20) is now rewritten (the earlier version had a typo). I now further elaborate, in a new second half of Section VII, how Eq. (20) generally allows for a finite standard deviation of the energy density (while for product states the expansion leads to a vanishing standard deviation of the energy density). This additional discussion replaces Appendix D of the previous version of the manuscript.

(7) In a new Section VIII, I explain how a finite standard deviation of the energy density appears trivially for a finite time average and how this allows the construction of further instantaneous states with a finite standard deviation.

(8) In Section IX, I expound on how the uncertainty relations lead to a bounds on the standard deviation of the energy density. The results for the closed and open systems that were highlighted in the earlier version are now further discussed in more depth. I further cite general references concerning the microcanonical ensemble and discuss, once again, causality (and the Lieb-Robinson bounds).

(9) In a new brief Section XI, I explain how driven systems such as the ones analyzed in the current work may display an equilibrium behavior with an effective Hamiltonian.

(10) A new Appendix (D) outlines how a purely gedanken experiment may be carried out to create an initial state with |w|<1 for the spin model of Section VIA.

(11) Other references were added (including, e.g., those relating to work in driven quantum systems).

(12) Specifically, and thanks to the last referee, In a long footnote (Ref. [66]), I highlight that the |w|<1 states exhibit long range correlations. I explain that these correlations appear whenever the second moment (i.e., the variance) of the time derivative of the energy density is finite. A finite first order moment (i.e., the simple average) of this derivative is the requisite for all systems that we study in this work. I further connect a finite variance to the considerations of Eq. (3).

Throughout, the paper was further streamlined for consistency and clarity. Several previous inline equations were converted into explicitly written equations to ease reading. Some equations were further highlighted so that the reader will recognize what the author considers to be the more central results.

---

## Editorial Decision

resubmitted